# A checkpoint function for Nup98 in nuclear pore formation suggested by novel inhibitory nanobodies

Mireia Solà Colom[1,2], Zhenglin Fu [ID][1], Philip Gunkel [ID][1], Thomas Güttler [ID][1,3], Sergei Trakhanov[1], Vasundara Srinivasan [ID][1,4], Kathrin Gregor [ID][1], Tino Pleiner [ID][1,5] & Dirk Görlich [ID][1✉]

## Abstract

**Nuclear pore complex (NPC) biogenesis is a still enigmatic example of protein self-assembly. We now introduce several cross-reacting anti-Nup nanobodies for imaging intact nuclear pore complexes from frog to human. We also report a simplified assay that directly tracks postmitotic NPC assembly with added fluorophore-labeled anti-Nup nanobodies. During interphase, NPCs are inserted into a pre-existing nuclear envelope. Monitoring this process is challenging because newly assembled NPCs are indistinguishable from pre-existing ones. We overcame this problem by inserting *Xenopus*-derived NPCs into human nuclear envelopes and using frog-specific anti-Nup nanobodies for detection. We further asked whether anti-Nup nanobodies could serve as NPC assembly inhibitors. Using a selection strategy against conserved epitopes, we obtained anti-Nup93, Nup98, and Nup155 nanobodies that block Nup–Nup interfaces and arrest NPC assembly. We solved structures of nanobody-target complexes and identified roles for the Nup93 α-solenoid domain in recruiting Nup358 and the Nup214·88·62 complex, as well as for Nup155 and the Nup98 autoproteolytic domain in NPC scaffold assembly. The latter suggests a checkpoint linking pore formation to the assembly of the Nup98-dominated permeability barrier.**

**Keywords** FG Phase; Nuclear Envelope; Nuclear Pores; Nuclear Transport; VHH Antibodies
**Subject Categories** Membranes & Trafficking; Methods & Resources; Organelles

## Introduction

The nucleus and cytoplasm of eukaryotic cells are separated by the nuclear envelope (NE), which comprises two concentric lipid bilayers: an outer nuclear membrane (ONM) and an inner nuclear membrane (INM). This separation protects the genome, allows

proper handling of introns, and provides sophisticated control over gene expression. However, it also necessitates a dedicated infrastructure for communication and fine-tuned exchange between the two compartments. Nucleocytoplasmic transport proceeds through NE-embedded nuclear pore complexes (NPCs; reviewed in Görlich and Kutay, 1999; Wing et al, 2022). With a molecular mass of around 120 MDa and an outer diameter of ~120 nm in higher eukaryotes, NPCs are among the cells' largest protein complexes. They are composed of about 30 different proteins called nucleoporins, or Nups for short (Fig. 1A; reviewed in Knockenhauer and Schwartz, 2016).

The NPC scaffold has octagonal symmetry and is organized in three coaxial, stacked rings. The inner ring (IR) encloses the central transport channel, and it is flanked by two outer rings: the cytoplasmic ring (CR) and the nuclear ring (NR; Fig. 1A). Several Nups pre-assemble into subcomplexes. The Y-complex, named for its Y-shaped structure, is the largest Nup complex and the most prominent building block of the outer rings (Siniossoglou et al, 2000; Harel et al, 2003b; Walther et al, 2003a; Beck et al, 2004, 2007). In vertebrates, it consists of nine proteins, namely Nup160, Nup133, Nup107, Nup96, Nup75, Nup43, Nup37, Sec13, and Seh1.

The Nup93·Nup188 complex (Miller et al, 2000) is a constituent of the inner ring, while recent NPC structures have shown that the (paralogous) Nup93·Nup205 complex is present in all three rings (Bley et al, 2022; Mosalaganti et al, 2022; Zhu et al, 2022). Nup35 (also called Nup53 or MP44) and Nup155 are further structural elements of the inner ring, connecting it through Aladin and the membrane-integral nucleoporin Ndc1 to the pore membrane (Eisenhardt et al, 2014; Mosalaganti et al, 2022).

So-called FG Nups equip NPCs with a permeability barrier. Their intrinsically disordered FG repeat domains can engage in cohesive interactions and condense into a selective FG phase. The FG phase is highly permeable to shuttling nuclear transport receptors (NTRs) and NTR·cargo complexes, while rejecting inert macromolecules that are not recognized as valid cargoes (Schmidt and Görlich, 2015; Labokha et al, 2013; Frey and Görlich, 2007).

Vertebrate NPCs contain 11 FG Nups, each anchored in a specific way. Nup98 provides the most cohesive and barrier-critical FG domain (Hülsmann et al, 2012). It is anchored by its

[1]Department of Cellular Logistics, Max Planck Institute for Multidisciplinary Sciences, Göttingen, Germany. [2]Present address: AI Proteins, 20 Overland St., Boston, MA, USA. [3]Present address: Octapharma Biopharmaceuticals, Im Neuenheimer Feld 590, 69120 Heidelberg, Germany. [4]Present address: Department of Chemistry, Institute of Biochemistry and Molecular Biology, Universität Hamburg, Hamburg, Germany. [5]Present address: Department of Molecular and Cellular Physiology, Stanford University School of Medicine, CA, USA. ✉E-mail: goerlich@mpinat.mpg.de

autoproteolytic domain (APD) either to the flexible N-terminus of Nup96 in the Y-complex (Hodel et al, 2002) or to the β-propeller of Nup88 (Griffis et al, 2003). The N-terminus of Nup93 provides binding sites for two other FG Nup complexes, namely the Nup62·58·54 coiled-coil trimer complex (Chug et al, 2015; Stuwe et al, 2015) and the Nup214·Nup88·Nup62 complex (Griffis et al, 2003; Bley et al, 2022). The Nup62·Nup214·Nup88 complex also appears to contact Nup85 (Fontana et al, 2022).

Y-complexes of the cytoplasmic ring anchor Nup358/ RanBP2 (Beck et al, 2004, 2007), a giant FG Nup that forms the cytoplasmic filaments (Walther et al, 2002; Wu et al, 1995; Yokohama et al, 1995). In addition to providing several FG domain segments, Nup358 pentamerizes and functions as an architectural element of the cytoplasmic ring that stabilizes interactions between Y-complexes (Von Appen et al, 2015; Zhu et al, 2022; Bley et al, 2022; Mosalaganti et al, 2022).

In animals, NPCs form de novo from individual Nup complexes at two distinct stages of the cell cycle, namely upon exit from mitosis and during interphase. In mitosis, NPCs and the NE disassemble prior to chromosome segregation, resulting in soluble Nups and Nup complexes as well as in the dispersal of membrane-integral Nups into vesicles/NE remnants (reviewed by Kutay et al, 2021). After cell division, NPCs re-assemble "postmitotically" from these building blocks in a synchronized wave, while fusion events between membrane sheets and vesicles re-form a closed NE and establish the special topology of the pore membrane.

Extracts from activated *Xenopus* eggs have been widely used to reconstitute postmitotic NPC assembly in the test tube (Newport, 1987; Finlay and Forbes, 1990; Marshall and Wilson, 1997; Bernis and Forbes, 2015). *Xenopus* eggs are laid in metaphase of meiosis II. Their nuclei are thus disassembled, and they contain large stockpiles of all NPC and NE components. Upon activation, they return to an interphase state and become assembly-competent. In a typical nuclear assembly experiment, egg extract is mixed with demembranated sperm chromatin that serves as an assembly template. Within an hour, nuclei form that contain functional NPCs, exclude inert macromolecules, and actively accumulate supplied import cargoes. This in vitro system thus recapitulates NPC and NE formation at the end of cell division and allows the analysis of phenotypes that would be lethal in a cellular setting (Fig. 1B).

A powerful approach has been to deplete specific Nup components from *Xenopus* egg extracts and analyze the consequences for the assembly process (reviewed in Antonin et al, 2008; Schellhaus et al, 2016). This approach revealed essential functions of the Y-complex (Walther et al, 2003a; Harel et al, 2003b; Franz et al, 2007; Rasala et al, 2008), Nup35 (Vollmer et al, 2012), Nup93 (Sachdev et al, 2012), and Nup155 (Franz et al, 2005) in NPC scaffold assembly. This way, it has also been shown that a functional permeability barrier requires cohesive Nup98 FG repeat interactions (Hülsmann et al, 2012).

However, Nup depletions are tedious; they may remain incomplete, may damage the extracts in a non-specific manner, or may not be feasible for all components. Furthermore, removing an entire protein or Nup complex abolishes multiple protein–protein interactions and introduces a rather dramatic change in the system. Therefore, it is not straightforward to define which assembly step(s) are specifically affected or whether the observed phenotypes correspond to actual intermediates.

In contrast to the postmitotic assembly mode described above, NPCs are inserted into an intact NE during interphase. This requires a pore-forming fusion event between the inner and outer nuclear membranes (reviewed in Otsuka and Ellenberg, 2018) and is also more fundamental since it is the only NPC biogenesis pathway in organisms with closed mitosis (such as fungi).

Interphase NPC assembly has been followed by examining the increase in NPC numbers after the completion of postmitotic NPC assembly from *Xenopus* egg extracts (D'Angelo et al, 2006; Vollmer et al, 2015) and also visualized by live-cell and electron microscopy in other model systems (Maul et al, 1972; Goldberg et al, 1997; Dultz and Ellenberg, 2010; Otsuka et al, 2016). The earliest assembly event has been proposed to be the direct binding of Nup153 to the INM. Nup153, in turn, is thought to recruit the Y-complex to new assembly sites (Vollmer et al, 2015). Subsequently, additional Nups appear to be recruited, perhaps exerting mechanical force to deform the NE. The local fusion between the INM and ONM seems to require Torsins (Laudermilch et al, 2016; Rampello et al, 2020; Prophet et al, 2022), but is otherwise only poorly understood.

It is generally accepted that interphase and postmitotic NPC assembly proceed through a stepwise sequence of structurally defined intermediates, and fundamental mechanistic differences between the two pathways have been described (Antonin et al, 2008; Dultz and Ellenberg, 2010; Otsuka et al, 2016, 2018). Nevertheless, either pathway is still poorly understood, mainly because it has been very difficult to identify assembly intermediates, arrange them in temporal order, and characterize them both biochemically and structurally.

In this study, we introduce the use of Nup-specific nanobodies to investigate the mechanisms of both NPC assembly modes. Nanobodies (Nbs) are the isolated variable domains of heavy chain-only IgGs (Casterman et al, 1993). They are small in size (~12–15 kDa) and can be selected for very high affinity (with dissociation constants down to the picomolar range) and robust folding. In addition, functional nanobodies can be produced in microorganisms at high yields and easily labeled with fluorescent dyes (reviewed in Cheloha et al, 2020; Helma et al, 2015; Ingram et al, 2018; Muyldermans, 2013). We (Pleiner et al, 2015; Chug et al, 2015) and others (Nordeen et al, 2020) have previously employed nanobodies to aid in the biochemical and structural characterization of NPCs. Here, we extend the anti-Nup-nanobody toolbox to investigate the mechanisms of NPC biogenesis and overcome limitations of the currently available assembly assays.

Our first set of nanobodies, termed tracking Nbs, bind soluble Nup complexes as well as intact NPCs. These nanobodies allow robust Nup tracking along the course of postmitotic NPC assembly by simply adding them in a labeled form to an assembly reaction and detecting them by direct fluorescence microscopy. We also present a novel approach to study the insertion of NPCs into an intact interphase NE. It relies on human cell nuclei as NPC insertion templates, a sfGFP-Nup107 fusion to mark pre-existing NPCs, interphase *Xenopus* egg extract to supply nucleoporins that assemble new NPCs into human NEs, and tracking nanobodies that selectively stain and thus identify newly inserted *Xenopus* NPCs. The system is flexible in terms of which Nups are being tracked, and it allows for biochemical manipulation.

A second set of anti-Nup nanobodies, termed inhibitory Nbs, block essential Nup–Nup interactions and arrest the assembly of

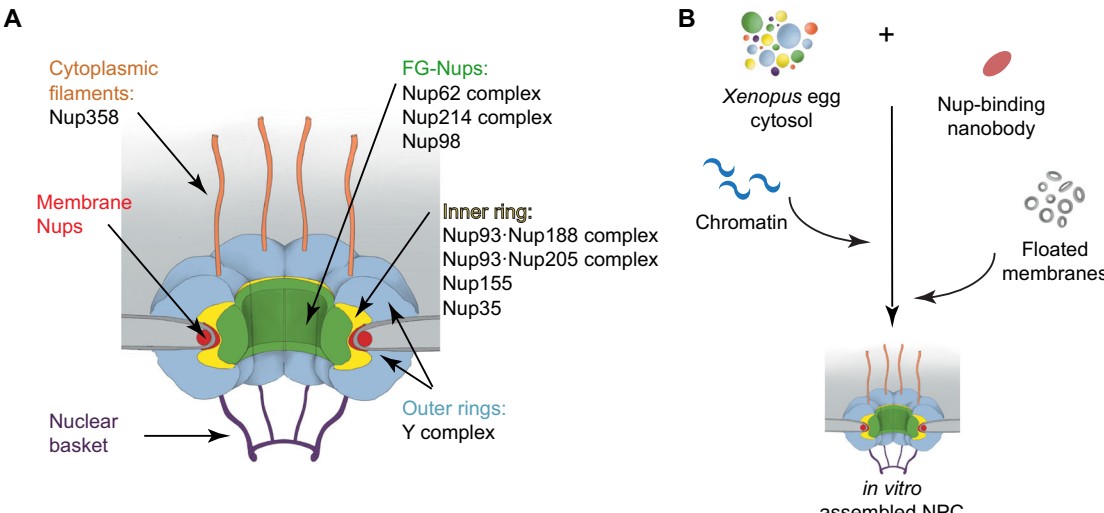

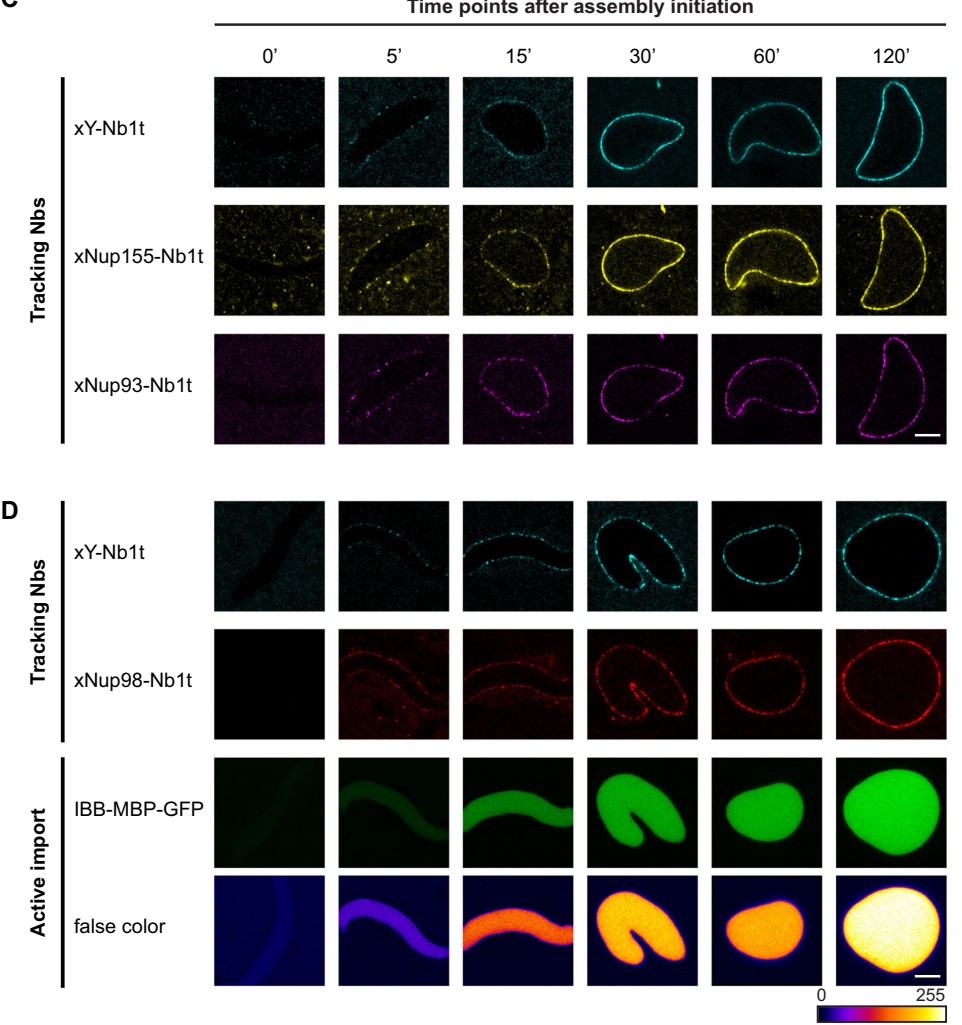

◄ **Figure 1.   Anti-Nup nanobodies for tracking postmitotic NPC assembly from *Xenopus* egg extracts.**

(A) Schematic representation of the NPC showing the locations of the major Nup complexes (B) Illustration of the nuclear assembly assay from *Xenopus* egg extracts in the presence of anti-Nup nanobodies (see Fig. 2A). (C, D) For tracking, the soluble fraction of *Xenopus* egg extract was pre-incubated with indicated fluorescent nanobodies at a 50 nM concentration, which is roughly stoichiometric to endogenous Nups (Wühr et al, 2014). Energy mix and sperm chromatin were added, and the chromatin was allowed to decondense for 15 min. Then a purified *Xenopus* membrane fraction was added and the reactions allowed to proceed at 20 °C. Images of live nuclei were acquired at the indicated time points using a LSM880 FAST Airyscan confocal microscope. (C) The Y-complex, Nup155, and Nup93 are recruited to the NE as nuclear assembly proceeds and nuclei grow in size. (D) Reconstituted nuclei become transport-competent as the Y-complex and Nup98 are recruited to assembly sites. 1 μM IBB-MBP-GFP was added as an active nuclear import reporter. This is a triple fusion of the Importin β-binding domain of importin α (a very strong nuclear import signal; Görlich et al, 1996), the maltose-binding protein (MBP) and a green fluorescent protein (GFP). For the "false color" representation, the "Fire" LUT from "FiJi" was used. Scale bar, 5 μm. Source data are available online for this figure.

functional NPCs by targeting conserved Nup epitopes that get buried during NPCs assembly. All inhibitory nanobodies cross-react between *Xenopus* and human Nups, whereas tracking nanobodies can be either cross-reactive or *Xenopus*-specific. We describe an anti-Nup93 inhibitory nanobody that prevents the incorporation of Nup358 and the Nup214·88·62 complex, indicating a specific arrest in the assembly of the cytoplasmic ring. In addition, inhibitory nanobodies targeting distinct domains of Nup155 and the APD of Nup98 block very early assembly steps. The requirement of the Nup98-APD for the assembly of the NPC scaffold suggests that the formation of NE-perforating pores is closely linked to the establishment of the permeability barrier. This can be seen as a checkpoint to avoid assembly intermediates with non-selectively open pores.

## Results

### Anti-Nup nanobodies for tracking postmitotic NPC assembly from *Xenopus* egg extracts

We have previously generated a toolbox of *Xenopus*-specific anti-Nup nanobodies that recognize fully assembled NPCs (Pleiner et al, 2015). These nanobodies were directly coupled to fluorophores. They yielded bright and specific fluorescent signals at the NE of *Xenopus* cells, without having to use secondary detection reagents. We now use these nanobodies to simplify NPC assembly assays.

To study the postmitotic mode, we mixed various anti-Nup nanobodies (carrying compatible fluorophores) with *Xenopus* egg extract, an energy-regenerating system, sperm chromatin, and egg membranes (Fig. 1B). The nanobodies bound their Nup targets without interfering with the formation of functional NPCs, thus allowing a multiplexed Nup tracking during the assembly process by confocal laser scanning microscopy—live and without additional sample preparation steps (Fig. 1C).

In some experiments, we also added an IBB-MBP-GFP fusion as a fluorescent import cargo. The nuclear accumulation of this IBB fusion allows correlating Nup recruitments with the formation of functional NPCs (Fig. 1D), because cargo retention in the nuclear compartment and the import-driving RanGTP gradient both require a properly installed permeability barrier (reviewed in Schmidt and Görlich, 2015; Knockenhauer and Schwartz, 2016).

For this study, we added 13 new anti-Nup nanobodies to our repertoire (see below). This set (listed in Fig. 2A) now includes nanobodies for tracking the Y-complex, Nup155, Nup93, Nup35, Nup98, Nup358, and the Nup62·Nup58·Nup54 and Nup214·-Nup88·Nup62 complexes during NPC assembly. The nanobody

names (Fig. 2A) include the recognized Nup, the recognized species as a prefix (x, *Xenopus*-specific; xh, *Xenopus*-human cross-reacting), a running number, and the suffix "t" (for "tracking").

### Tracking interphase NPC assembly using semi-permeabilized HeLa cells and *Xenopus* egg extracts

Compared to postmitotic NPC assembly, interphase NPC formation is more challenging to track because only a few pores get sporadically inserted into the growing NE over a relatively long period of time. Moreover, the newly inserted NPCs are indistinguishable from the pre-existing ones; they look identical (reviewed in Otsuka and Ellenberg, 2018; Weberruss and Antonin, 2016). Nevertheless, we wanted to establish a readily accessible experimental assay for interphase NPC assembly that allows direct biochemical manipulation. This requires not only that the process occurs in a test tube but also that newly inserted NPCs can be identified and distinguished from pre-existing ones. To this end, we considered assembling NPCs from a *Xenopus* egg extract into human NEs. Newly inserted NPCs would then be of *Xenopus* origin and thus chemically distinct from the pre-existing human NPCs. *Xenopus*-specific tracking nanobodies should then label only newly assembled NPCs (see Fig. 3A for a scheme). However, given the great evolutionary distance between frogs and humans, it was initially unclear whether the NPC components and assembly machineries would still be compatible enough for such an interspecies experiment to work.

To mark pre-existing human NPCs, we generated a CRISPR/Cas9-edited HeLa cell line whose Nup107 carries an N-terminal sfGFP tag. We grew these cells on coverslips, selectively permeabilized their plasma membranes with a low concentration (30 μg/ml) digitonin, and incubated them with a membrane-free ("high-speed") extract of activated *Xenopus* eggs, which is a rich source of assembly-ready Nups and Nup complexes. To track the assembly of new nuclear pores, we added fluorophore-labeled nanobodies recognizing the *Xenopus* Y-complex, Nup155, or other frog Nups.

In equatorial confocal sections of the incubated nuclei, the *Xenopus*-specific tracking nanobodies produced a bright fluorescent signal that coincided (at low resolution) with the GFP signal (Fig. 3B, top panels). High-resolution images not only confirmed that the *Xenopus* Y-complex had accumulated at the human NE but also resolved the *Xenopus* signal as being located between old NPCs (Appendix Fig. S1). In nuclear surface scans, the tracking nanobodies produced a fluorescent spot pattern that was similar to, but offset from, the GFP signal, which is also consistent with the formation of new NPCs (Fig. 3B, bottom panels). Importantly, the

## A

**Overview of the nanobodies used in this study**

| Nb ID | Nup target | Specificity | Epitope | $K_D$ *Xenopus* Nup | $K_D$ human Nup | Use | Reference |
|---|---|---|---|---|---|---|---|
| xY-Nb1t | Y complex | *Xenopus* | Not determined (*) | n.d. | No binding | Tracking | Present study |
| xhNup133-Nb2t | Y complex | *Xenopus* + human | Nup133 β-propeller | ≤ 10 pM | 2 nM | Tracking | Present study |
| xNup155-Nb1t | Nup155 | *Xenopus* | Orth. to xh-Nup155-Nb2i and xh-Nup155-Nb3i | ≤ 10 pM | No binding | Tracking | Pleiner et al. 2015 |
| xhNup155-Nb2i | Nup155 | *Xenopus* + human | NTD β-propeller (**) | 3 nM | 400 pM | Inhibitory | Present study |
| xhNup155-Nb3i | Nup155 | *Xenopus* + human | CTD α-solenoid (**) | ≤ 10 pM | 500 pM | Inhibitory | Present study |
| xNup93-Nb1t (TP179) | Nup93 | *Xenopus* | α-solenoid (residues 168-820) | 100 pM | No binding | Tracking | Pleiner et al. 2015 |
| xNup93-Nb2t (TP317) | Nup93 | *Xenopus* | α-solenoid (residues 168-820) (**) | 4 nM | No binding | Cryo-EM reference | Pleiner et al. 2015 |
| xhNup93-Nb3t | Nup93 | *Xenopus* + human | α-solenoid (residues 168-820) | ≤ 10 pM | 1 nM | Tracking | Present study |
| xhNup93-Nb4i | Nup93 | *Xenopus* + human | α-solenoid (residues 168-820) (**) | ≤ 10 pM | 3 nM | Inhibitory | Present study |
| xhNup35-Nb1t | Nup35 | *Xenopus* + human | RRM domain (**) | 500 pM | 800 pM | Tracking | Present study |
| xNup98-Nb1t (TP377) | Nup98 | *Xenopus* | APD (residues 715-866) (**) | 1.7 nM | > 100 nM (***) | Tracking | Pleiner et al. 2015 |
| xhNup98-Nb2i | Nup98 | *Xenopus* + human | APD (residues 715-866) (**) | ≤ 10 pM | 20 pM | Inhibitory | Present study |
| xhNup98-Nb3i | Nup98 | *Xenopus* + human | APD (residues 715-866) (**) | ≤ 10 pM | ≤ 10 pM | Inhibitory | Present study |
| xNup62-Nb1t (Nb15) | Nup62 complex | *Xenopus* | Nup62·Nup58·Nup54 trimeric coiled coil | 1 nM | No binding | Tracking | Chug et al. 2015 |
| xhNup214-Nb1t | Nup214 complex | *Xenopus* + human | Nup214·Nup62·Nup88 trimeric coiled coil | ≤ 10 pM | ≤ 10 pM | Tracking | Present study |
| xNup358-Nb1t | Nup358 | *Xenopus* | NTD (residues 1-145) | ≤ 10 pM | No binding | Tracking | Present study |
| xhNup358-Nb2t | Nup358 | *Xenopus* + human | NTD (residues 1-145) | 60 pM | 20 pM | Tracking | Present study |

## B

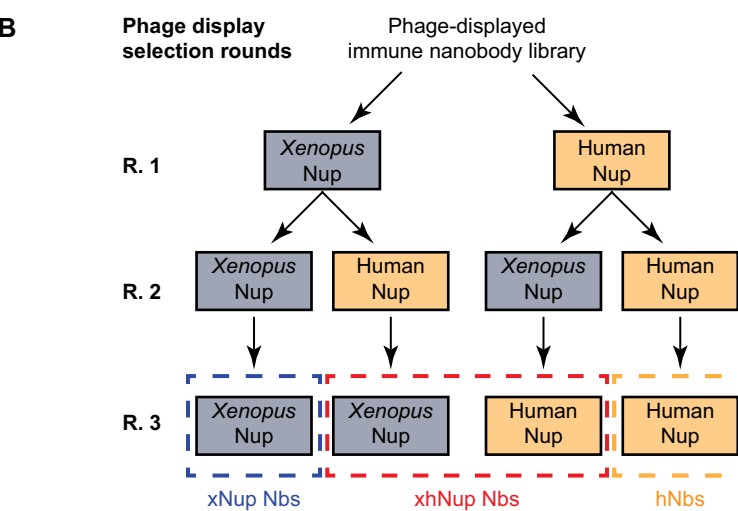

**Figure 2. Overview of anti-Nup nanobodies and phage display strategy used in this study.**

(A) List of anti-Nup nanobodies. The targeted Nup, specificity, affinities estimated by biolayer interferometry (Appendix Fig. S2), epitope, and use are indicated. NTD N-terminal domain, CTD C-terminal domain, APD autoproteolytic domain of Nup98, orth. orthogonal, n.d. not determined. (*) Not determined because this nanobody recognizes a still unidentified domain within the ~800 kDa-sized frog Y-complex that is not yet available in recombinant form. (**) for epitope determination, see below Figs. 10, 11 and EV4; Appendix Fig. S4. (***) Affinities with a $K_D$ worse than 100 nM dissociate too rapidly for contributing detectable signals in NPC stainings. (B) Scheme of the used phage display strategy to identify nanobodies with tailored specificities. xNup Nbs *Xenopus*-specific anti-Nup nanobodies, xhNup *Xenopus*-human cross-reactive anti-Nup nanobodies, hNup Nbs human-specific anti-Nup nanobodies.

nanobody signal was strictly dependent on the addition of egg extract, confirming the species specificity of the tracking nanobodies and the *Xenopus* origin of the labeled NPCs.

During extract incubation, the initially flat HeLa nuclei became rounder (Fig. EV1A). They grew ~twofold in volume (Fig. EV1B), suggesting that a massive import of *Xenopus* nuclear proteins had occurred, inflating nuclei by their osmotic-colloidal pressure. This indicates that the newly assembled nuclei contained a functional permeability barrier that kept the imported material inside the nuclei.

To assess whether different *Xenopus* Nups co-localize with each other on the detected spots, we stained the samples with various combinations of tracking nanobodies and imaged them at higher resolution. Figure 4A shows that newly assembled NPCs can be clearly identified as sfGFP-Nup107-free but *Xenopus* Y-complex, Nup155, and Nup93-co-localizing spots. *Xenopus* Nup358 and the Nup62 complex also co-localized with the *Xenopus* Y-complex, indicating that barrier-forming FG Nups and the central organizer of the cytoplasmic NPC ring were also recruited to the newly inserted pores.

The *Xenopus*-specific xNup98-Nb1t and a *Xenopus*-specific anti-Nup153 antibody co-localized with *Xenopus*-derived NPCs—as expected from a faithful assembly process—but also gave bright signals co-localizing with GFP-positive old human NPCs (Fig. 4B). Since these reagents do not recognize the human proteins, as confirmed by the lack of staining in the absence of egg extract, this suggests either that the pre-existing NPCs still had vacant binding sites for Nup98/Nup153, or that some exchange between NPC-bound and free populations occurred over time. This would be consistent with the reported mobility of these Nups (Griffis et al, 2002a, 2004; Rabut et al, 2004). It should be noted, however, that the huge excess of *Xenopus* FG Nups in the added extract is likely to increase the off-rates of the human proteins—as their anchoring occurs through multiple weak interactions that are easily out-competed by the excess molecules in a stepwise fashion.

The presented approach relies on a high species specificity of the used tracking nanobodies (Fig. 2A), which can indeed be explained by the corresponding nanobody-target structures. In the case of xNup98-Nb1t, an arginine residue at CDR3 of the nanobody interacts with an asparagine residue at the APD of *Xenopus* Nup98, but would clash with a lysine at the same position in its human counterpart, preventing its binding (Pleiner et al, 2015; Fig. EV2A). Similarly, our newly solved cryo-EM structure shows that xNup93-Nb2t fits well into a pocket at the C-terminus of *Xenopus* Nup93, but clashes with the backbone of human Nup93 at the same protein region (Fig. EV2B; Appendix Table S1).

As expected, cross-specific anti-Nup35 and anti-Nup214 complex nanobodies (see below) stained both pre-existing (GFP-positive) human and newly assembled (GFP-free) *Xenopus* NPCs as spots of similar brightness (Fig. 4C). This is a key control, documenting the architectural equivalence of old and new NPCs.

To further confirm the insertion of properly assembled NPCs, we acquired higher-resolution images of the HeLa NE containing newly assembled pores, using two-color STED (stimulation emission depletion) microscopy (Klar et al, 2000; Göttfert et al, 2013). The anti-GFP nanobodies (labeling Nup107 in the pre-existing human NPCs) allowed to resolve ring-like structures of ~100–120 nm in diameter (Fig. 5A,C), which is consistent with the dimensions of the NPC outer rings. As expected, the *Xenopus*-specific xY-Nb1t-stained rings of a similar shape and diameter, which did not co-localize with the GFP-labeled old NPCs (Fig. 5B,C).

These STED images (in particular the higher resolved channel 1) appear to visualize not only fully assembled NPCs. Possibly they also captured smaller *Xenopus* intermediates that might not yet have reached the ring-like assembly stage (Fig. 5B). This is particularly obvious in comparison to the imaged old human NPCs that are surrounded by only very few intermediates (Fig. 5A, left).

New NPCs detected in this assay did not cluster in specific membrane regions but maintained an even and homogenous density on the human NE (Figs. 4 and 5A,B), which was also observed by Otsuka et al (2016) in live-cell microscopy studies. Furthermore, the fact that essentially no GFP signal was detected in the *Xenopus* Nup structures (Figs. 4 and 5A,B) indicates that the new pores assembled de novo and did not originate from pore-splitting events, which is also consistent with previous conclusions (D'Angelo et al, 2006; Dultz and Ellenberg, 2010).

In these assays, we included membrane-depleted ("high-speed") egg extracts. Thus, if the new NPCs had incorporated any integral membrane Nups, such as Ndc1 or Gp210, they should be of human origin. To test this, we detected human Ndc1 and Gp210 with antibodies and observed that both indeed co-localized with the new pores stained with the *Xenopus*-specific xY-Nb1t (Fig. EV3). Thus, human membrane Nups present in the HeLa cells support the assembly of the new *Xenopus* NPCs.

Two hours after the addition of the *Xenopus* extract, *Xenopus* NPCs represented ~30% (range 15–60%) of the pre-existing human NPCs (Fig. EV1C). Longer incubations did not increase the number of assembled *Xenopus* NPCs. Perhaps this plateau was reached because membrane Nups (in particular, the essential transmembrane Nup Ndc1) had become limiting. Nevertheless, this initial rate of NPC insertion is higher than in cultured HeLa cells (Maul et al, 1972), where the count of NPCs doubles over a period of ~20 h (G1 + S + G2). We would explain this high rate by the embryonic origin of the added extract and the high concentration of available Nups.

The insertion of frog NPCs into a human NE is not a trivial result. Instead, it implies that the ~30 soluble different frog Nups can cooperate smoothly with the still elusive human membrane fusion machinery and the at least three human membrane-integral

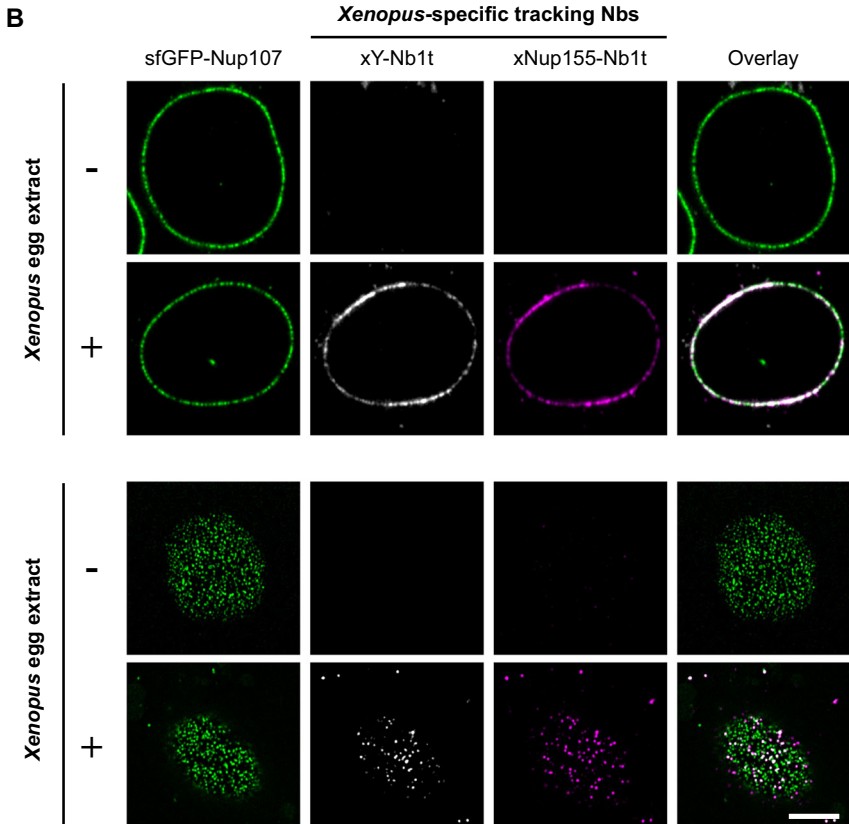

**Figure 3. Interphase insertion of *Xenopus* NPCs into human NEs.**

(**A**) Scheme of the assay. To initiate NPC assembly into an intact human NE, *Xenopus* egg extract (as a source of soluble xNups), fluorophore-labeled tracking nanobodies, and energy mix are added to digitonin-permeabilized HeLa cells expressing a genomically sfGFP-tagged hNup107. In this setup, the newly inserted NPCs are of *Xenopus* origin and can be specifically stained by the *Xenopus*-specific nanobodies. (**B**) Indicated *Xenopus*-specific tracking nanobodies were conjugated at two or three engineered cysteines (Pleiner et al, 2015) to Alexa Fluor 568 maleimide (xY-Nb1t) or Alexa Fluor 647 maleimide (xNup155-Nb1t). They were then mixed with either buffer (−) or a high-speed *Xenopus* egg extract (+), supplemented with energy, and incubated with the semi-permeabilized sfGFP-hNup107 cells for 2 h at room temperature (RT), before confocal images were acquired. The *Xenopus*-specific tracking nanobodies gave a bright, specific signal at the midplane (top panels) and a punctate signal at the NE surface (lower panels), which is consistent with the insertion of new NPCs. Scale bar, 5 μm. Source data are available online for this figure.

Nups. Indeed, the fact that human Nup98 and human Nup153 can be exchanged by their *Xenopus* counterparts (Fig. 4C) shows that even NPCs containing Nups from multiple species are operational. Given that the lineages leading to mammals and frogs diverged ~350 million years ago (Benton et al, 2015), such interspecies compatibility is remarkable and suggests that functionally essential Nup–Nup interfaces are well conserved through evolution.

## A cross-species phage display strategy for targeting functional Nup regions

Capturing and characterizing intermediate stages of NPC assembly has remained the main bottleneck in the field. As NPCs are essential structures, such capture can be complicated by lethal phenotypes and is thus particularly challenging in living cells. Biochemical alterations are easier to introduce in *Xenopus* cell-free extracts, where all potential assembly phenotypes are, in principle, accessible. In this system, only a few inhibitors of nuclear pore formation have been described, including wheat germ agglutinin (WGA) (Wiese et al, 1997), BAPTA (1,2-bis(O-aminophenoxy) ethane-N, N, N', N'-tetraacetic acid) (Macaulay and Forbes, 1996; Wiese et al, 1997; Bernis and Forbes, 2015), and an excess of importin β (Harel et al, 2003a; Walther et al, 2003b). These treatments result in pore-free membranes and thus block NPC assembly at very early stages, providing only limited information on the mechanisms of this process.

We reasoned that nanobodies directed against functional Nup interfaces could prevent relevant Nup–Nup interactions and thus arrest the NPC assembly process at discrete intermediate steps. Such nanobodies would act through the smallest possible change in the system without having to deplete any protein. They would provide "epitope resolution" and not just reveal requirements for protein components or entire protein complexes. In addition, nanobodies could be added to the soluble NPC components in nonlimiting amounts, which would ensure that all targeted Nup molecules become trapped in Nup·nanobody complexes when NPC assembly is initiated.

To test this idea, we performed nuclear assembly reactions in the presence of various anti-Nup nanobodies selected by Pleiner et al (2015) (Fig. 1C,D); however, none of them affected the formation of functional nuclear pores (see below). While initially disappointing, this outcome became plausible when considering (i) that all of the initially tested nanobodies had been selected for their bright stain of fully assembled NPCs (Pleiner et al, 2015), and thus for targeting well-accessible epitopes, and (ii) that all of them recognize non-conserved, *Xenopus*-specific epitopes. In contrast, functional protein regions, whose block can arrest NPC assembly, should get masked by protein–protein interactions and be well conserved through evolution. With this in mind, we set out to select nanobodies recognizing conserved Nup epitopes.

Since Pleiner et al (2015) had immunized alpacas exclusively with *Xenopus* Nups, the antibody response was likely biased towards variable *Xenopus*-specific surfaces that the immune system did not recognize as "self". To boost an immune response directed against conserved epitopes, we re-immunized the same animals simultaneously with *Xenopus* and human Nup homologs. We chose the same Nup domains that had already been administered by Pleiner et al (2015), namely the autoproteolytic domain (APD)

of Nup98, the Nup93 α-solenoid domain, and the full-length Nup155. In addition, we immunized the N-terminal domain (NTD) of Nup358 (Kassube et al, 2012), the RRM domain of Nup35 (Handa et al, 2006), the β-propeller of the Y-complex component Nup133 (Berke et al, 2004), and the trimeric (Δ FG) Nup62·Nup214·Nup88 complex (Figs. 1A and 2A). This focus on well-folded immunogens considered that nanobodies only rarely bind linear epitopes.

We employed three parallel phage display strategies to select binders of a given Nup from the obtained immune libraries: (i) a *Xenopus*-specific (multi-round) selection using the *Xenopus* Nup ortholog as a bait, (ii) a selection using the human Nup ortholog, and (iii) a cross-specific selection alternating between the *Xenopus* and the human Nup orthologs in successive selection rounds, in order to exclusively enrich nanobodies recognizing Nup epitopes that are conserved among the two species (Fig. 2B). The bait concentrations were gradually reduced in each selection round to a final concentration below 1 nM to only enrich high-affinity binders. Not surprisingly, we re-discovered the same nanobody classes as Pleiner et al (2015) in the *Xenopus*-specific selections against Nup98, Nup93, and Nup155. However, all cross-specific pannings selected new nanobody classes—consistent with an enrichment of nanobodies against conserved Nup epitopes. Figure 2A provides an overview of all selected nanobodies used in this study, including estimates for their affinity as measured by biolayer interferometry (Appendix Fig. S2).

## Cross-specific tracking nanobodies that stain human NPCs

We expressed and purified representative members of all nanobody classes obtained from the cross-species selections, conjugated them to fluorophores, and screened them for NPC staining in HeLa cells. A subset, including xhNup133-Nb2t, xhNup93-Nb3t, xhNup35-Nb1t, and xhNup358-Nb2t yield bright and specific NPC signals in both fixed, Triton X-100-permeabilized (Fig. 6A) and non-fixed, digitonin-semi-permeabilized (Appendix Fig. S3) HeLa cells with still intact nuclear membranes, suggesting that their epitopes are well-exposed in fully assembled NPCs.

In addition, these four tracking nanobodies perform well in super-resolution microscopy and allow to resolve NPC rings on HeLa cell NEs by STED (Fig. 6B). As expected, nanobodies recognizing Nups of the NPC outer rings (e.g., Nup358, Nup133) labeled rings with larger diameters than nanobodies against inner ring components (e.g., Nup93, Nup35; Fig. 6C).

Since these nanobodies originate from cross-specific phage display selections, they can also track NPC assembly from *Xenopus* egg extracts (Fig. 4A,C, see also below Figs. 8 and EV6). Probably, they stain NPCs from other vertebrate organisms as well. To understand such broad cross-reaction, we solved the crystal structure of xhNup35-Nb1t bound to its target. This revealed that the nanobody docks in two copies to a highly conserved epitope of the homodimeric RRM domain of Nup35 (Fig. EV4), contacting identical residues in the human and frog Nup. Consistent with this, xhNup35-Nb1t stains fully assembled human and frog NPCs (Figs. 4C, 6A,B, 8, and EV6) and is compatible with the assembly of functional nuclear pores.

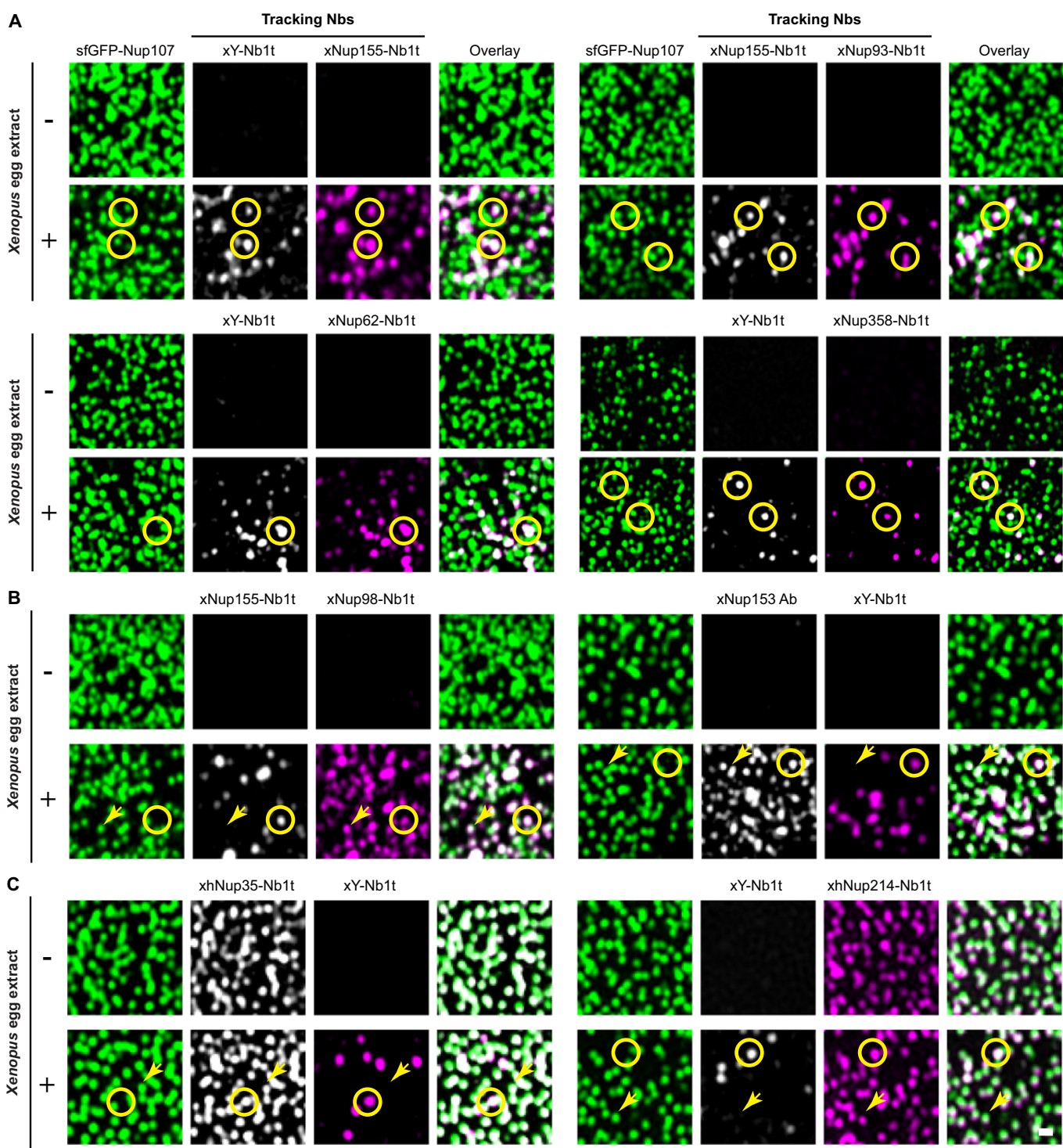

**Figure 4. Tracking nanobodies to image pre-existing human NPCs, new frog NPCs, and mobile Nups.**

Interphase NPC insertion was initiated as described in Fig. 3, in the presence of the indicated tracking nanobodies and either buffer (−) or *Xenopus* egg extract (+). (**A**) *Xenopus* Y-complex co-localizes with (*Xenopus*) xNup155, xNup62, and xNup358 in newly assembled frog NPCs. xNup155 also co-localizes with xNup93 in all cases. (**B**) xNup98 and xNup153 are detected in both pre-existing and newly assembled NPCs. The buffer controls (no egg extract added) indicate that this is not due to cross-reaction but to either complementing vacant binding sites in old pores or to subunit exchange. (**C**) Cross-specific xhNup35-Nb1t and xhNup214-Nb1t co-localize with both pre-existing and newly assembled NPCs. To resolve single pores, the acquired confocal images were deconvolved by Airyscan Processing (Huff, 2015). Circles mark newly inserted pores; arrows point to pre-existing NPCs. Prefixes indicating species specificity: *x*, *Xenopus*-specific; *xh*, cross-reaction between *Xenopus* and human. Scale bar, 0.25 μm. Source data are available online for this figure.

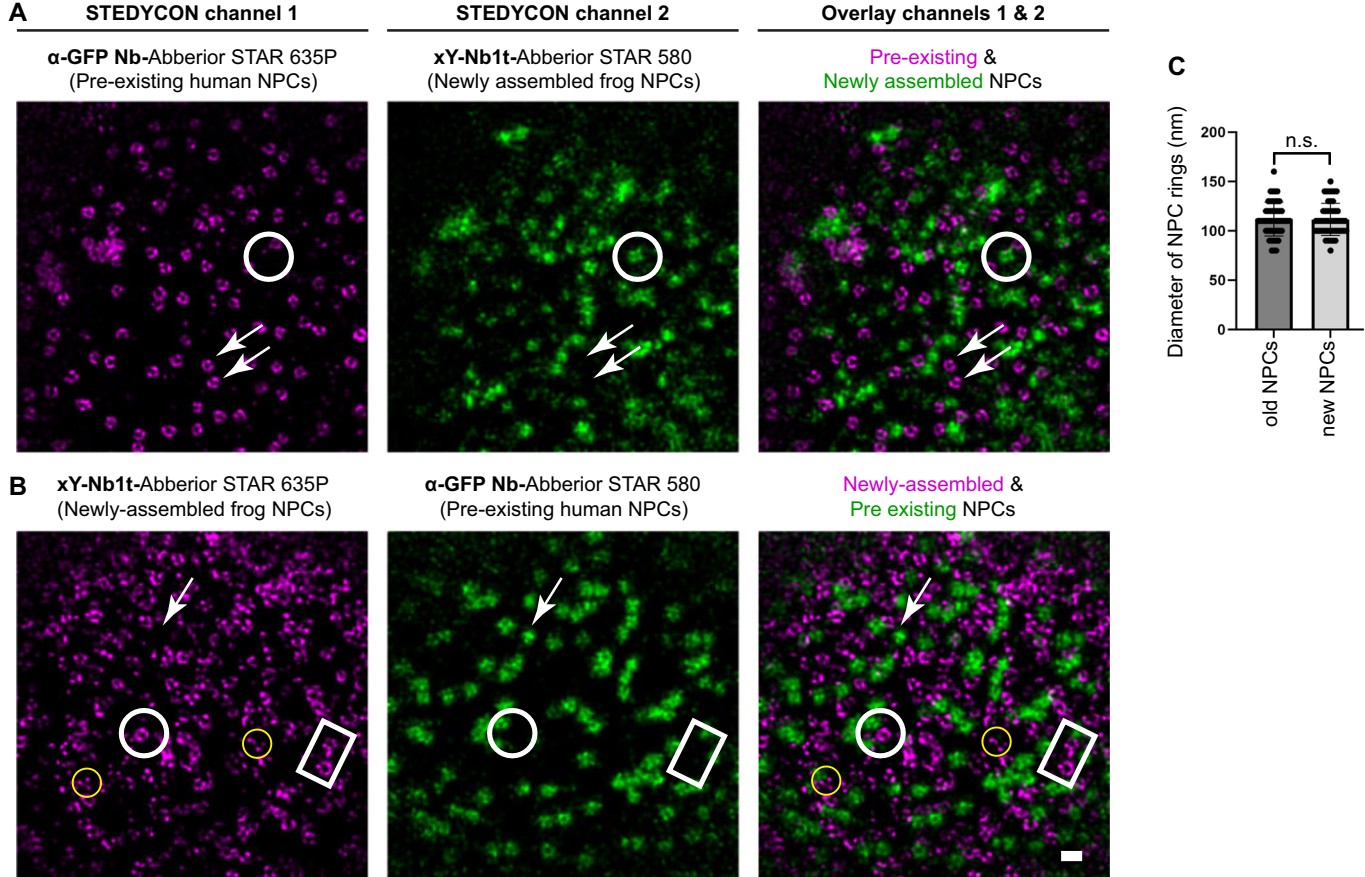

**Figure 5.   Ring-like architecture of *Xenopus* NPCs newly inserted into human NEs.**

New NPC insertion was initiated as described in Fig. 3, in the presence of an anti-GFP nanobody (Kirchhofer et al, 2010) targeting the human sfGFP-Nup107 fusion of pre-existing NPCs and xY-Nb1t specific for newly inserted frog NPCs. The two nanobodies had been coupled to compatible STED dyes using maleimide chemistry (Pleiner et al, 2015). Two-channel images of the NE plane were acquired using a STEDYCON microscope. The dyes were swapped between (**A**) and (**B**), as channel 1 provides higher resolution. Newly inserted frog NPCs are marked by white circles or squares, pre-existing pores by white arrows. Yellow circles indicate putative *Xenopus* intermediates that have not yet reached the state of a ring-like assembly. Scale bar, 200 nm. (**A**) The anti-GFP nanobody revealed ring-like human NPC structures of ~100–120 nm in diameter, which did not co-localize with the spots labeled by the *Xenopus*-specific xY-Nb1t. (**B**) The *Xenopus*-specific xY-Nb1t yielded ring-like structures of similar size that did not co-localize with the GFP-labeled old human NPCs. (**C**) NPC ring diameters from at least 50 different pores and two different nuclei were measured with FiJi. n.s. difference not significant with $P = 0.9$ (unpaired $t$ test). Source data are available online for this figure.

## Identification of cross-specific nanobodies that target buried NPC epitopes

The NPC staining by the human-*Xenopus* cross-reacting nanobodies xhNup93-Nb4i, xhNup98-Nb2i, xhNup155-Nb2i, and xhNup155-Nb3i was remarkably dim (Fig. 6A; Appendix Fig. S3). This was particularly striking when compared to the internal sfGFP-Nup107 reference and to the tracking nanobodies tested in parallel. These very weak signals could be explained by (i) a poor affinity to their target Nups or (ii) by their epitopes being buried by protein–protein interactions in fully assembled NPCs.

To distinguish between these scenarios, we measured affinities for their recombinant *Xenopus* and human Nup targets and found low nanomolar to low picomolar $K_D$s (Fig. 2A; Appendix Fig. S2). These numbers indicate a tight or even very tight binding. As a complementary experiment, we used the weakly-staining nanobodies as baits to purify soluble (unassembled) Nups from *Xenopus* egg extracts and HeLa cell lysates. This resulted in a clean one-step

purification of Nup93 (with co-purifying Nup188 and Nup205), Nup98 (with co-purifying Gle2), and Nup155 (Fig. 6D). HeLa and *Xenopus* egg extracts gave essentially identical results, confirming the cross-species recognition. Thus, the xhNup93-Nb4i, xhNup98-Nb2i, xhNup155-Nb2i, and xhNup155-Nb3i nanobodies recognize their targets specifically, which in turn is consistent with their low to sub-nanomolar affinities (Fig. 2A). Their weak staining of intact NPCs is therefore best explained by a burial of their epitopes.

## Blocks in NPC assembly by nanobodies that target conserved, buried epitopes

Next, we tested all cross-specific nanobodies for their effects on postmitotic NPC assembly from *Xenopus* egg extracts. All nanobodies that brightly stained NPCs of HeLa cells also allowed the formation of import-competent nuclei and, thus, of functional nuclear pores (Fig. 7A), which is not surprising given that their epitopes are obviously exposed on the NPC surface.

**A**

NPCs of Triton-permeabilized HeLa cells stained with:

Tracking nanobodies targeting accessible epitopes

xhNup358-Nb2t    xhNup35-Nb1t    xhNup93-Nb3t    xhNup133-Nb1t

Inhibitory nanobodies targeting buried epitopes

xhNup93-Nb4i    xhNup98-Nb2i    xhNup155-Nb2i    xhNup155-Nb3i

Nanobody-AF647

0    255

sfGFP-Nup107

**B**    STED imaging of human NPCs on HeLa cells with tracking nanobodies

Outer ring Nups

xhNup133-Nb1t    xhNup358-Nb2t

Inner ring Nups

xhNup35-Nb1t    xhNup93-Nb3t

**C**

Diameter of NPC rings (nm)

n.s.    ****    ****    n.s.

xhNup133-Nb1t
xhNup358-Nb2t
xhNup35-Nb1t
xhNup93-Nb3t

**D**    Purification of endogenous Nups with inhibitory nanobodies targeting buried epitopes

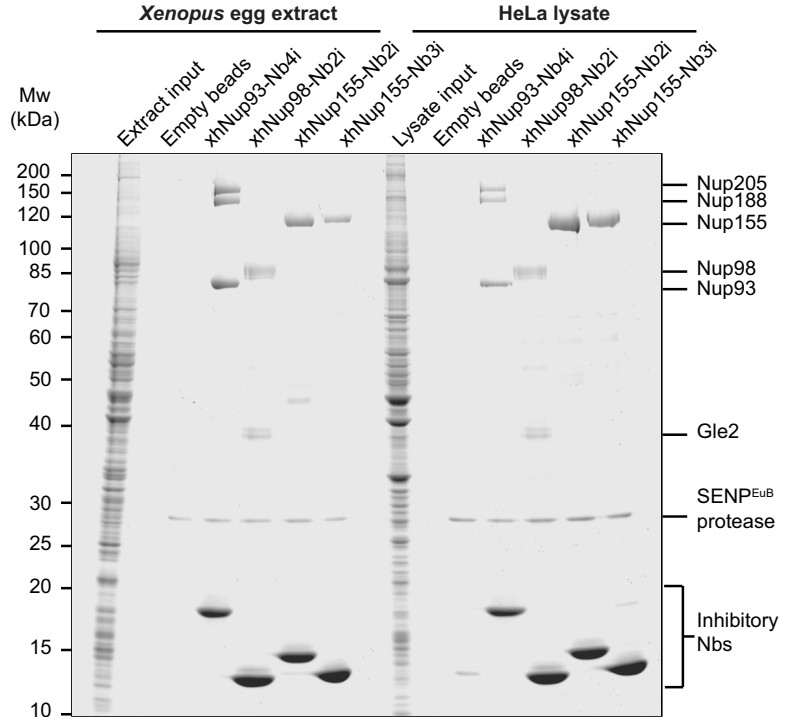

**Figure 6.  Cross-specific (frog-human) anti-Nup nanobodies for imaging NPCs and purifying Nup complexes.**

(A) Paraformaldehyde (PFA)-fixed and Triton X-100-permeabilized sfGFP-Nup107 HeLa cells were stained with 35 nM of the indicated nanobodies coupled to Alexa Fluor 647. Confocal sections across nuclei at their equatorial plane were acquired using a Leica SP8 confocal microscope with identical settings. Four of the eight nanobodies gave bright NPC signals, indicating that they recognize exposed Nup epitopes. For a clearer comparison between the different nanobodies, a false-color representation is shown (along with the applied color look-up table). The GFP signal is included as an internal reference. Scale bar, 10 μm. (B) Tracking nanobodies allow the ring-like structure of human NPCs to be to resolved by STED microscopy. HeLa cells were stained with 35 nM of indicated Abberior STAR 635P-labeled nanobodies and imaged using a STEDYCON system. Representative complete NPC rings are marked in white circles or rectangles. Scale bar, 250 nm. (C) Diameters of nanobody-decorated NPCs rings were measured with FiJi as described in Fig. 5C. n.s., no significant difference; **** significant difference with *P* < 0.0001 (unpaired *t* test). For each tracking nanobody, the diameters of 45 NPC rings from three different images were measured. (D) Nanobodies that failed to stain HeLa NPCs do purify their human and *Xenopus* Nup targets in a specific manner. Indicated nanobodies were produced with a cleavable N-terminal (biotin)-Avi-SUMO[Eu1]-tag, immobilized on streptavidin-agarose beads, and incubated with either HeLa lysate or *Xenopus* egg extract to capture the targeted Nups. Nup-nanobody complexes were then eluted by tag-cleavage with SENP[EuB] protease (Vera-Rodriguez et al, 2019) and analyzed by SDS-PAGE/Coomassie staining. Protein identities were confirmed by mass spectrometry. Source data are available online for this figure.

In contrast, the xhNup93-Nb4i nanobody caused a quite diagnostic phenotype. Import of IBB-MBP-GFP into assembled nuclei was ≥tenfold reduced. Likewise, nuclear exclusion of MBP-Cherry and an NES-GFP fusion failed; instead, the two reporters equilibrated between nucleus and cytoplasm. This is consistent with defects in the (passive) permeability barrier and in Xpo1-mediated nuclear export (Fig. 7B,D). Nevertheless, it appears that at least rudimentary NPCs can still assemble in the presence of this nanobody (see also below).

In contrast, the phenotype was very severe when xhNup98-Nb2i, xhNup98-Nb3i, xhNup155-Nb2i, or xhNup155-Nb3i were present during the nuclear assembly reaction. Nuclear import of IBB-MBP-GFP was drastically reduced—100-fold or even to the extent of nuclear exclusion (Figs. 7B,D and EV5A, left). In addition, the nuclei remained very small, probably because there was no nuclear import to drive nuclear growth. Indeed, the "nuclei" trapped by these nanobodies resemble those assembled in the presence of WGA, an excess of importin β, or BAPTA (Fig. 7C), which are known to inhibit nuclear pore formation completely and to result in a pore-free nuclear envelope around the added chromatin (Wiese et al, 1997; Harel et al, 2003a; Bernis and Forbes, 2015).

Strikingly, these nanobodies had no inhibitory effect when added 30 min after membrane addition and thus beginning of the assembly reaction (Fig. EV5B). Therefore, they do not block nuclear import directly, supporting the interpretation that they arrest NPC assembly by targeting Nup epitopes that are only transiently accessible. Furthermore, this control is another strong argument against off-target effects (besides their exquisite binding specificities, as documented in Fig. 6).

## Composition of nanobody-arrested (pseudo) NPC assemblies

To characterize the assembly-arrested phenotypes at the level of NPCs, we repeated the postmitotic assembly reactions with inhibitory nanobodies, stained the resulting (pseudo) nuclei with the full set of tracking nanobodies, and analyzed the samples in several ways.

Figure 8A shows scans through the equator of the whole (pseudo) nuclei, Fig. 8B shows overview scans of the nuclear surface, while Fig. 8C is a quantification of the respective Nup-positive structures. This revealed that the inhibitory anti-Nup98 (xhNup98-Nb2i) and anti-Nup155 nanobodies (xhNup155-Nb2i and -Nb3i) caused severe defects, namely a dramatic loss of inner ring components (Nup35, Nup93, Nup155) and FG Nups (Nup98, Nup358, Nup62·58·54 and Nup214·62·88 subcomplexes) from the NE.

Nuclear surface scans at single-pore resolution (Fig. EV6) not only confirmed the severe reduction of Nup-positive structures in the arrested nuclei, but also revealed that the residual structures had largely lost the NPC-typical colocalization between Nups. This is consistent with inhibitory anti-Nup98 and anti-Nup155 nanobodies causing defects early in NPC assembly.

A quantification of these effects is shown in Fig. 9. It comprises 25 scatter plots, each analyzing (for hundreds of Y-complex-positive structures) how a given inhibitory nanobody affected the colocalization of a tracked Nup with the Y-complex. The figure is laid out like a table, where each column represents one inhibitory and each row one tracking nanobody. These analyses confirm that the inhibitory anti-Nup98 and anti-Nup155 nanobodies disrupt the ordered recruitment of inner ring and FG Nups to the Y-complex. They also document that the inhibitory xhNup93-Nb4i was still compatible with the assembly of a basic NPC scaffold and the recruitment of Nup98 and the Nup62·Nup58·Nup54 complex, but caused a selective loss of Nup358 and the Nup214·62·88 complex from nuclear pores (see xhNup93-Nb4i panels in Figs. 8A,B, 9, and EV6). In the following, we will delve deeper into the assembly phenotypes and their structural basis.

## Inhibitory nanobodies targeting distinct Nup155 domains block NPC scaffold assembly

xhNup155-Nb2i recognizes the N-terminal β-propeller of Nup155, whereas xhNup155-Nb3i recognizes its α-solenoid domain (Fig. 2A; Appendix Fig. S4A–C). These two inhibitory nanobodies are not only orthogonal to each other but also orthogonal to the anti-Nup155 tracking nanobody (xNup155-Nb1t; Appendix Fig. S4D). Thus, xNup155-Nb1t can track Nup155 molecules also in (nano-body-inhibited) pseudo-NPCs.

The two inhibitory nanobodies blocked the postmitotic incorporation of Nup155 into Y-complex-positive structures (Figs. 8, 9, and EV6) and prevented the assembly of transport-competent NPCs (Fig. 7B, D). Nup155 interacts with Nup160 of the Y-complex, with Nup205, and with unstructured regions of Nup98 (R3) and Nup35. In addition, it forms an interaction hub with Ndc1 and Aladin that anchors the inner ring to the nuclear membrane (Lin et al, 2016; Fontana et al, 2022; Mosalaganti et al, 2022). So far, we cannot tell which of these interactions are directly blocked by the nanobodies. However, their strong phenotypes reaffirm the central role of Nup155 in scaffold assembly and demonstrate that both its β-propeller and its α-solenoid domain are required.

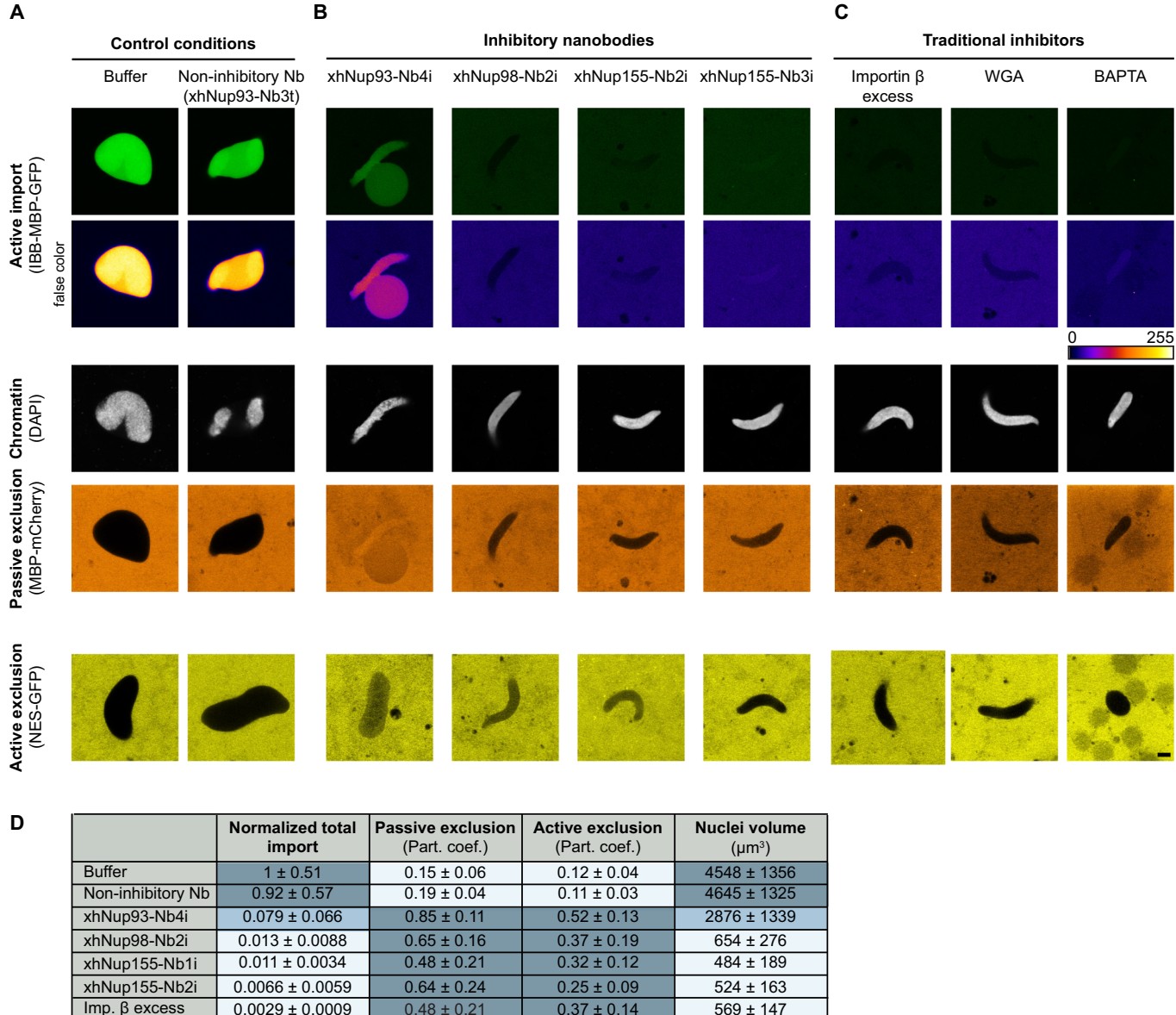

**Figure 7. Inhibitory anti-Nup nanobodies disrupt the transport competence of newly assembled nuclei.**

*Xenopus* egg extract was supplemented with either 2 µM of the indicated anti-Nup nanobodies, 2 µM importin β, 2.5 µM WGA, or 5 mM BAPTA. Postmitotic nuclear assembly was then initiated by adding energy mix, sperm chromatin, egg membranes and allowed to proceed for 1 h at 20 °C. DAPI, a fluorescent import cargo (IBB-MBP-GFP), and either a passive nuclear exclusion reporter (MBP-mCherry) or an active (Xpo1-dependent) export cargo (NES-GFP) were added. Confocal scans were taken 30 min later. Scale bar, 5 µm. (A) None of the *Xenopus*-specific or cross-specific nanobodies that brightly stain HeLa NPCs (see Fig. 6A) caused transport defects, as exemplified by the xhNup93-Nb3t control. (B) The four (inhibitory) cross-specific Nbs that failed to stain intact (fully assembled) NPCs (Fig. 6A; Appendix Fig. S3) and (C) the traditional inhibitors of NPC assembly resulted in nuclei that failed to import the IBB-MBP-GFP fusion protein. (D) Total import was quantified as the partition coefficient (Part. coef.) between the mean intensity inside and outside the nuclei multiplied by the nuclear volumes. The obtained values were then normalized to the control values (i.e., nuclei assembled without inhibitor). Passive and active exclusion were quantified as the partition coefficient (Part. coef.) between the mean intensity outside and inside the nuclei. The nuclear volume was quantified from 3D reconstructions of acquired z-stacks. In all cases, the mean and SD of 10–50 nuclei from at least four independent experiments are shown. Source data are available online for this figure.

Even though the assembly of functional NPCs was fully blocked, it was evident that some larger NPC subcomplexes of varying composition still assembled in the presence of the anti-Nup155 inhibitory nanobodies (Figs. 9 and EV6). For example, we observed objects that stained bright for Nup358 as well as for the Y- and Nup214 complexes but lacked Nup155. Others stained positive for Nup35, Nup93, and the Y-complex. These rudimentary structures alter the correlation between the intensity of the Y-complex and later-recruited Nups in the imaged pore-like structures (see Fig. 9 for an extensive analysis). They likely represent off-pathway intermediates that are heterogeneously assembled. Their occurrence illustrates that NPC building blocks have an intrinsic affinity for each other and suggests that proper NPC assembly is kinetically controlled—probably by cooperative binding events.

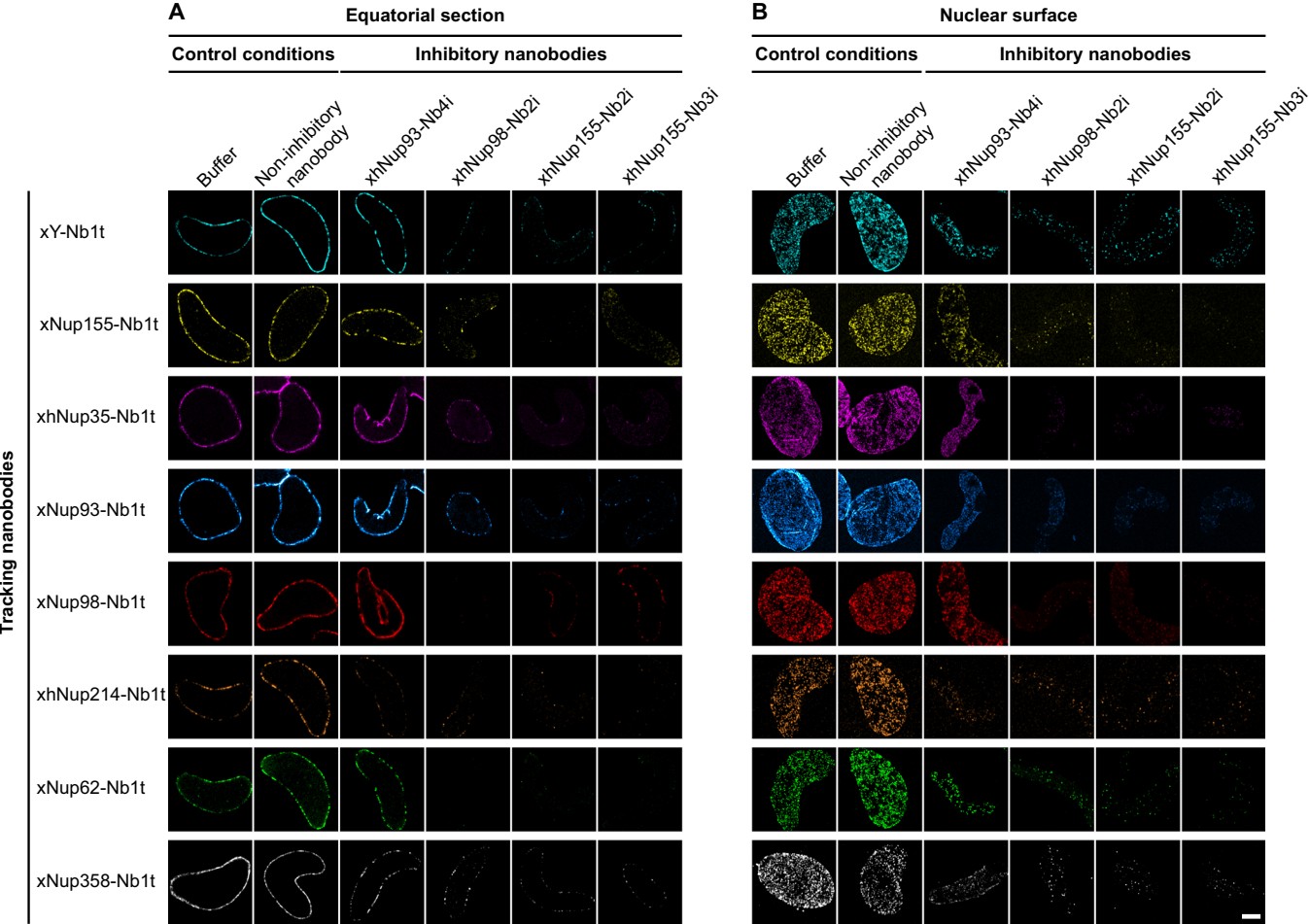

**C** **Number of Nup-positive spots (NPCs+arrested structures) detected by tracking nanobodies**
(normalized to the buffer control)

|  |  | Control conditions | | Inhibitory Nbs | | | |
|---|---|---|---|---|---|---|---|
|  |  | Buffer | Non-inhibitory Nb | xhNup93-Nb4i | xhNup98-Nb2i | xhNup155-Nb2i | xhNup155-Nb3i |
| **Tracking Nbs** | xY-Nb1t | 1 ± 0.29 | 0.90 ± 0.33 | 0.32 ± 0.18 | 0.26 ± 0.11 | 0.26 ± 0.12 | 0.18 ± 0.11 |
|  | xhNup35-Nb1t | 1 ± 0.33 | 0.90 ± 0.24 | 0.42 ± 0.25 | 0.13 ± 0.11 | 0.15 ± 0.08 | 0.05 ± 0.04 |
|  | xNup155-Nb1t | 1 ± 0.23 | 0.84 ± 0.20 | 0.27 ± 0.15 | 0.15 ± 0.10 | 0.09 ± 0.05 | 0.03 ± 0.02 |
|  | xNup93-Nb1t | 1 ± 0.16 | 0.94 ± 0.20 | 0.46 ± 0.21 | 0.15 ± 0.11 | 0.1 ± 0.05 | 0.09 ± 0.06 |
|  | xNup98-Nb1t | 1 ± 0.15 | 0.88 ± 0.11 | 0.37 ± 0.11 | 0.14 ± 0.09 | 0.35 ± 0.15 | 0.26 ± 0.16 |
|  | xNup62-Nb1t | 1 ± 0.29 | 1.02 ± 0.25 | 0.33 ± 0.21 | 0.07 ± 0.04 | 0.06 ± 0.04 | 0.04 ± 0.03 |
|  | xhNup214-Nb1t | 1 ± 0.19 | 1.04 ± 0.23 | 0.15 ± 0.09 | 0.10 ± 0.04 | 0.07 ± 0.03 | 0.06 ± 0.02 |
|  | xNup358-Nb1t | 1 ± 0.24 | 0.81 ± 0.21 | 0.12 ± 0.07 | 0.07 ± 0.04 | 0.05 ± 0.03 | 0.04 ± 0.03 |

**Figure 8.  NPC assembly arrests by inhibitory nanobodies.**

Postmitotic nuclear assembly with inhibitory nanobodies was as described in Fig. 7. (Pseudo) nuclei were PFA-fixed, Triton X-100-permeabilized, stained with fluorescent tracking Nbs targeting various NPC subcomplexes, and imaged on a LSM880 FAST Airyscan confocal microscope. Equatorial planes (**A**) and nuclear surface planes (**B**) of the nuclei are shown. Note that the fixation conditions can alter the morphology and apparent volume of the nuclei; the volume phenotype is thus better appreciated in live samples (i.e., Fig. 7). Scale bar, 5 µm. (**C**) Quantification of panel B but based on a larger dataset. Nup-positive structures were counted from acquired images of the nuclear surface. Note that all inhibitory nanobodies disrupted the recruitment of Nups to the NE. The non-inhibitory nanobody controls in the different rows included: xhNup93-Nb3t, xNup93-Nb1t, xhNup35-Nb1t, xNup155-Nb1t, xNup358-Nb1t, and xNup62-Nb1t. They were chosen so that they could not interfere with the labeled tracking nanobodies. Source data are available online for this figure.

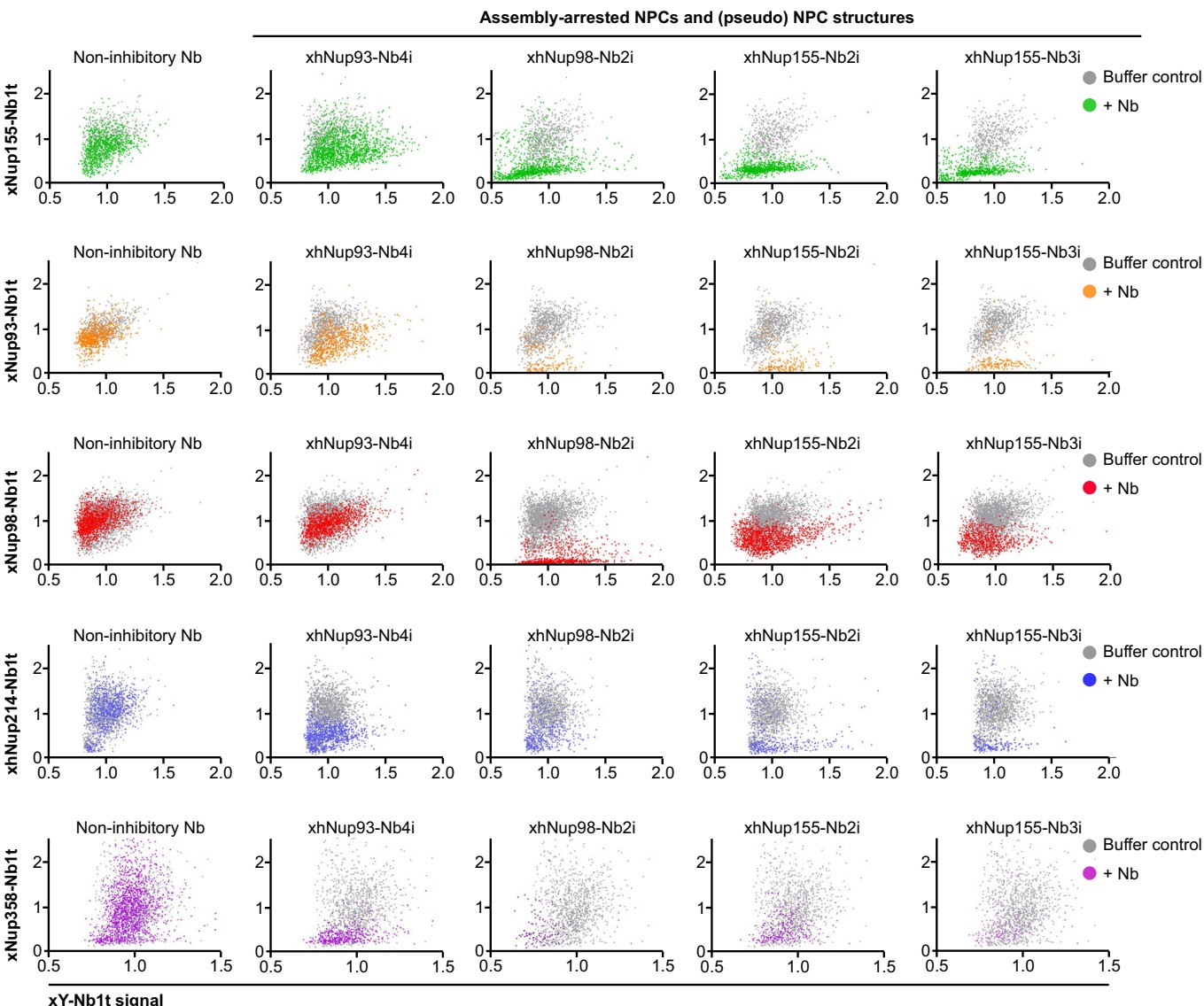

**Figure 9. Quantification of Nup colocalization in residual assembly-arrested (pseudo) NPC structures.**

Postmitotic NPC assembly was arrested by inhibitory nanobodies as in Figs. 7 and 8. Individual (pseudo) NPC structures were detected from 3-channel images. Selected images are shown in Fig. EV6. The Y-complex channel served as a reference. Signal intensities were quantified by ImageJ/ FiJi scripts. Figure shows 25 scatter plots for 25 combinations of inhibitory and tracking nanobodies. Each point in a plot represents a detected (Y-complex-positive) structure, whereby the point coordinates are the normalized signal intensity of the Y-complex and the signal intensity of a second Nup. The point clouds for nanobody-arrested structures are plotted in color and are overlaid with point clouds for unperturbed NPCs (buffer control) in gray. xhNup93-Nb4i selectively blocked the recruitment of Nup358 and the Nup214·88·62 complex to NPCs. xhNup98-Nb2i, xhNup155-Nb2i and 3i had more severe effects, disrupting already the recruitment of inner ring Nups (Nup93, Nup155) as well as of Nup98, Nup214 and Nup358 to Y-complex-positive structures. Note that these plots still underestimate the effects because they only show pore-like structures with detectable Y-complex. Structures with a low Y-complex signal and a high signal for any of the other Nups, as well as the general reduction in Y-complex-positive structures, were not considered. Data from two independent experiments with each 3–7 nuclei per sample are shown. The non-inhibitory nanobodies used in the experiments shown as different rows were xhNup93-Nb3t, xNup93-Nb1t, xhNup35-Nb1t, xNup155-Nb1t, xNup358-Nb1t, or xNup62-Nb1t. A total of 28610 NPC/pseudo-NPC structures were analyzed. Source data are available online for this figure.

## The inhibitory anti-Nup93 nanobody disrupts the assembly of the NPC cytoplasmic ring

The presence of xhNup93-Nb4i during postmitotic assembly altered the composition of NPCs. The Y-complex, Nup35, Nup93, Nup155, Nup98, and the Nup62·Nup58·Nup54 complex still co-localized in approximately wild-type ratios. In contrast, the Nup214·Nup88·Nup62 complex and Nup358 were detected only in few pore-like structures and with clearly reduced intensities (Figs. 8, 9, and EV6). Since the Nup214·Nup88·Nup62 complex and Nup358 reside exclusively on the cytoplasmic side of the NPC, this points to defects in the assembly of the cytoplasmic ring.

Nup358 is an architectural element of the cytoplasmic ring that comprises two Y-complex rings, and the outer Y-complex ring is

lost in the absence of Nup358 (Von Appen et al, 2015). We therefore assume that xhNup93-Nb4i phenocopies this effect. The epitope of xhNup93-Nb4i is poorly accessible in properly assembled NPCs (Fig. 6A; Appendix Fig. S3), but becomes accessible in NPCs upon Nup358-depletion (Fig. 10A). Thus, there is indeed an antagonism between this nanobody and Nup358 incorporation.

To map the functional epitope of xhNup93-Nb4i, we solved its structure in complex with the α-solenoid domain of xNup93 (residues 168–820) and xNup93-Nb2t by X-ray crystallography and cryo-EM (Fig. 10B–D; Appendix Tables S2 and S3). The tracking nanobody xNup93-Nb2t binds to a C-terminal region of the *Xenopus* Nup93 α-solenoid, it is frog-specific (Fig. 2A) and did not inhibit NPC assembly.

In contrast, xhNup93-Nb4i recognizes an epitope in the middle of the Nup93 α-solenoid (i.e., residues 450–536). This nanobody cannot interfere with the direct Nup93-Nup188 and Nup93-Nup205 interactions, since it efficiently purifies the two complexes from HeLa and *Xenopus* egg extracts (Fig. 6D). However, the alignment of our structure with recent cryo-EM maps of the NPC revealed that xhNup93-Nb4i clashes with a Nup205 molecule from a neighboring Nup93·Nup205 heterodimer (Fig. 10D), which would plausibly explain a failed assembly of the cytoplasmic ring (Lin et al, 2016; Bley et al, 2022; Mosalaganti et al, 2022; Zhu et al, 2022). This consideration further suggests that the occasional Nup214 and Nup358 signals seen in some of the xhNup93-Nb4i distorted NPCs represent off-pathway intermediates (Figs. 9 and EV6).

The failure to recruit Nup358 and the Nup214·Nup88·Nup62 complex is consistent with the observed functional deficits (Figs. 7–9 and EV6). The Nup214·Nup88·Nup62 complex directly contributes two FG domains (from Nup214 itself and from Nup62) to fully assembled NPCs. In addition, its Nup88 β-propeller anchors a subset of Nup98 molecules, whose FG domains would also be unavailable. Nup358 is normally present in 40 copies, each also contributing multiple FG sub-domains (Kosinski et al, 2016; Bley et al, 2022; Mosalaganti et al, 2022). The loss of the outer Y-complex ring results in the loss of anchoring sites (through Nup96) for another eight Nup98 molecules. Therefore, anti-Nup93-nanobody-distorted NPCs suffer a cumulative loss of ~6 megadaltons of FG mass. This explains the increased passive leakage through the FG-based permeability barrier and the defect in Xpo1/NES-mediated nuclear export (Fig. 7B), which heavily relies on Nup214 and Nup358 (Hutten and Kehlenbach, 2006; Labokha et al, 2013).

## Nup98 is not just anchored to the NPC scaffold but required for scaffold assembly

Nup98 has a conserved domain structure. The unusually conserved N-terminal FG domain is interrupted by a Gle2-binding (GLEBS) domain and makes a key contribution to the NPC permeability barrier (Bailer et al, 1998; Laurell et al, 2011; Hülsmann et al, 2012). It is followed by an intervening disordered domain with conserved linear motifs (R1, R2, and R3) and the aforementioned C-terminal autoproteolytic domain (APD) (Hodel et al, 2002; Bley et al, 2022). The catalytic activity of the APD is used in cis to cleave the initially synthesized Nup98–Nup96 fusion protein (called Nup196), which happens during or shortly after translation (Fontoura et al, 1999). Therefore, the egg extract contains the

already cleaved entities: Nup98 and Nup96, which is a subunit of the Y-complex.

The second APD function is to anchor Nup98 to NPCs—by binding either to the N-terminus of Nup96 (Hodel et al, 2002) or the β-propeller of Nup88 (Griffis et al, 2003; Stuwe et al, 2012; Bley et al, 2022). We have now characterized three nanobodies against the Nup98-APD. One of them (xNup98-Nb1t) stains intact NPCs, permits normal NPC biogenesis, and allows tracking of the assembly process (Figs. 1D, 8, 12, and EV6; Pleiner et al, 2015). By contrast, xhNup98-Nb2i and -Nb3i are inhibitory and cause a severe and early NPC assembly defect—comparable to a block of Nup155 (Figs. 7–9, EV5, and EV6). To understand this difference, we solved their crystal structures in complex with the Nup98-APD (Fig. 11A; Appendix Table S4).

The obtained structures show that the inhibitory nanobodies xhNup98-Nb2i and -Nb3i each recognize overlapping epitopes (Fig. 11A). Both inhibitory epitopes also overlap with the binding sites of the Nup98-APD for Nup96 and Nup88 (Fig. 11B), suggesting that these nanobodies fully block these interactions. Indeed, xhNup98-Nb2i abolishes the binding of Nup98 to both Nup96 and Nup88, as tested in a biochemical assay (Fig. 11C). By contrast, the tracking nanobody xNup98-Nb1t binds to a non-overlapping epitope on the Nup98-APD and is compatible with both interactions (see structural analysis in Fig. 11B and direct interaction assay in Fig. 11C).

So far, it was assumed that Nup98 gets recruited to a pre-assembled NPC scaffold—the argument being that the bulk of Nup98 arrives at NPC assembly sites rather late (Dultz et al, 2008). Our observation that preventing a Nup98–Nup96 interaction causes a much earlier defect, already at scaffold assembly, suggests a different scenario, namely a tight coupling between scaffold assembly and the establishment of the Nup98-dependent permeability barrier of NPCs.

To explore the phenotypes of the inhibitory anti-Nup98 nanobodies in interphase NPC assembly, we inserted *Xenopus* NPCs from egg extract into an intact human NE. Indeed, not only the traditional assembly inhibitors (WGA, BAPTA; Fig. 12A), but also the anti-Nup98 inhibitory nanobodies abolished new NPC insertion (Fig. 12B). Interestingly, however, the xhNup98-Nb2i nanobody did not prevent the recruitment of *Xenopus* Nup98 from the extract to old human NPCs (Fig. 12B). This might appear surprising but can be explained by the fact that the APD is not the only NPC interaction site of Nup98. Indeed, Nup98 is additionally anchored to NPCs by cohesive interactions between Nup98 FG repeats and FG domains of other Nup molecules, as well as by additional linear motifs of Nup98 that bind to Nup205 and Nup155 (Griffis et al, 2002a, 2004; Stuwe et al, 2012; Chatel et al, 2012; Hülsmann et al, 2012; Bley et al, 2022).

The NPC assembly block by the inhibitory anti-Nup98 nanobodies is thus of a different quality than (just) impeding the anchorage of an FG Nup to NPCs. This block occurs very early in the assembly pathway, suppressing the appearance of *Xenopus* Y-complex-positive spots at the NE (Figs. 8C and EV6). The occasionally observed weakly Y-complex-positive structures lacked any Nup98 signal (in contrast to old NPCs). We interpret this as the Nup98-APD mediating interactions that are essential for the assembly of the NPC scaffold. Given the inhibitory effect on the early Y-complex structures, we assume that the APD-Nup96 interaction is the relevant one. This requirement applies to both

**A**

**Immunofluorescence imaging with labeled nanobodies on HeLa cells**

| xhNup133-Nb2t | xhNup358-Nb2t | xhNup93-Nb4i | xhNup358-Nb2t & xhNup93-Nb4i |
|---|---|---|---|

Mock

Nup358 knockdown

**B**

xNup93-Nb2t

xhNup93-Nb4i

C-ter

N-ter

xNup93 (168-820)

**C**

Epitope of xNup93-Nb2t

Epitope of xhNup93-Nb4i

xNup93 (168-820)

**D** **The epitope of xhNup93-Nb4i overlaps with a Nup205 interface**

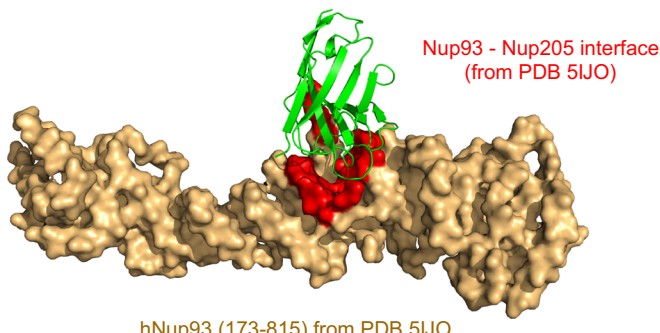

Nup93 - Nup205 interface (from PDB 5IJO)

hNup93 (173-815) from PDB 5IJO

**Figure 10. xhNup93-Nb4i recognizes a conserved epitope in the α-solenoid domain of Nup93 and clashes with Nup205 from a neighboring Nup93·Nup205 dimer.**

(A) The epitope of xhNup93-Nb4i becomes more exposed in NPCs depleted of Nup358. HeLa cells were transfected with siRNAs targeting Nup358, fixed and stained with the indicated tracking nanobodies. Scale bar, 10 μm. (B) Cryo-EM structure of *Xenopus laevis* xNup93(168–820) (gray) in complex with xhNup93-Nb4i (green) and xNup93-Nb2t (blue). The paratope residues of the nanobodies are highlighted in red. (C) A representation of the Nup93 surface with the nanobody epitopes highlighted in green and blue, respectively. (D) xhNup93-Nb4i blocks the interface with a neighboring Nup205 molecule (red). See Methods for details of crystallization, cryo-EM, and structure solving as well as Appendix Tables S2 and S3 for crystallographic and cryo-EM statistics. Source data are available online for this figure.

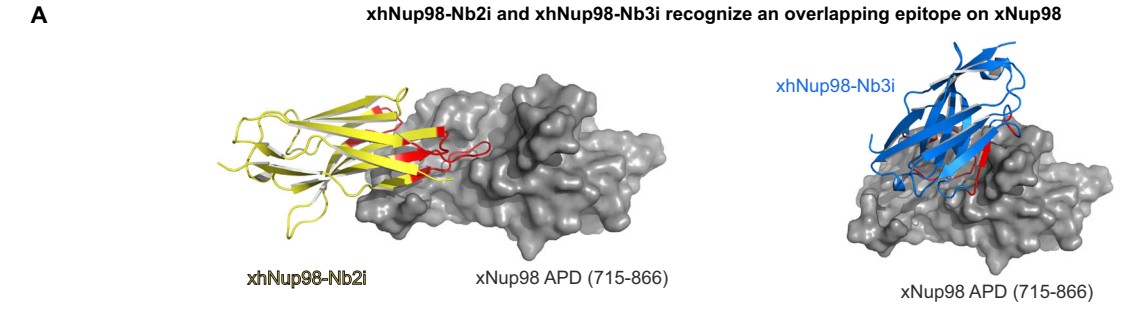

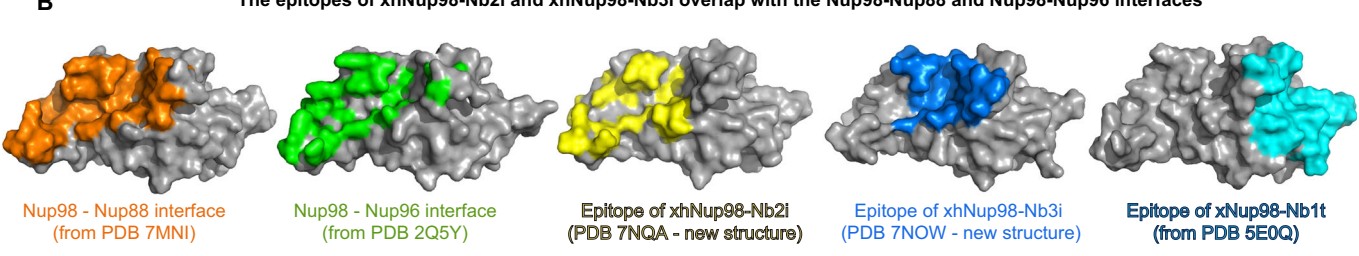

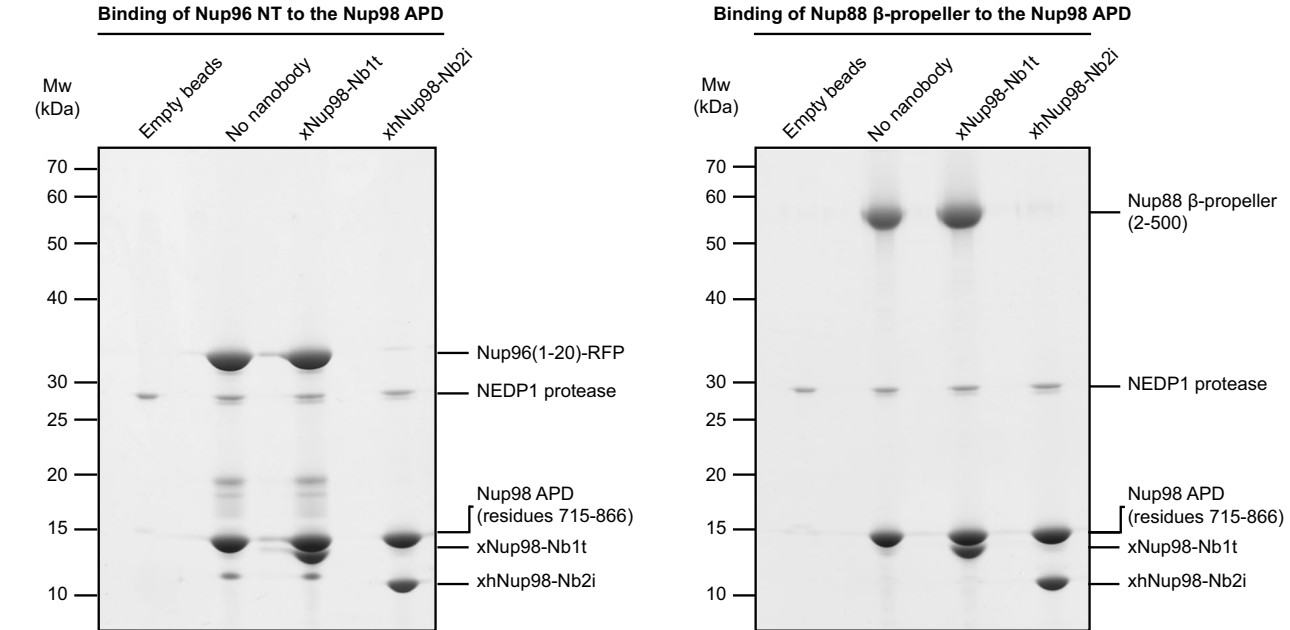

**Figure 11. xhNup98-Nb2i and Nb3i recognize a conserved APD region and block the Nup98–Nup96 and Nup98–Nup88 interactions.**

(A) Crystal structures of the APD of *Xenopus tropicalis* xNup98 (gray) in complex with xhNup98-Nb2i (yellow) and xhNup98-Nb3i (blue). The epitope-contacting residues of the nanobodies are highlighted in red. Both nanobodies recognize overlapping epitopes. Both nanobodies are equally effective in blocking NPC assembly (see Fig. EV5A for a comparison). See Methods for details of crystallization and structure solving and Appendix Table S4 for crystallographic statistics. (B) Surface representations of the APD of Nup98 showing its known interaction interfaces with Nup88 (orange), Nup96 (green), xhNup98-Nb2i (yellow), xhNup98-Nb2i (blue) and xNup98-Nb1t (cyan). (C) The His$_{14}$-NEDD8-tagged APD of xtNup98 (715–866) was immobilized on a Ni$^{2+}$ chelate matrix and incubated with either the xNup98-Nb1t or the xhNup98-Nb2i. Next, either the N-terminus of xNup96 (residues 1–20) fused to RFP (left) or the N-terminal β-propeller of xtNup88 (residues 2–500) (right) were added, and the immobilized APD was eluted along with its binding partners by the tag-cleaving NEDP1 protease (Frey and Görlich, 2014). Eluted fractions were analyzed by SDS-PAGE/Coomassie staining. Source data are available online for this figure.

postmitotic (Figs. 8, 9, and EV6) and interphase NPC assembly (Fig. 12B), suggesting that these two processes are, after all, more similar than currently believed. Given that the final NPC product is the same, this is a plausible scenario.

## Discussion

We present here a toolbox of anti-Nup nanobodies and demonstrate their use as specific trackers or inhibitors of the NPC

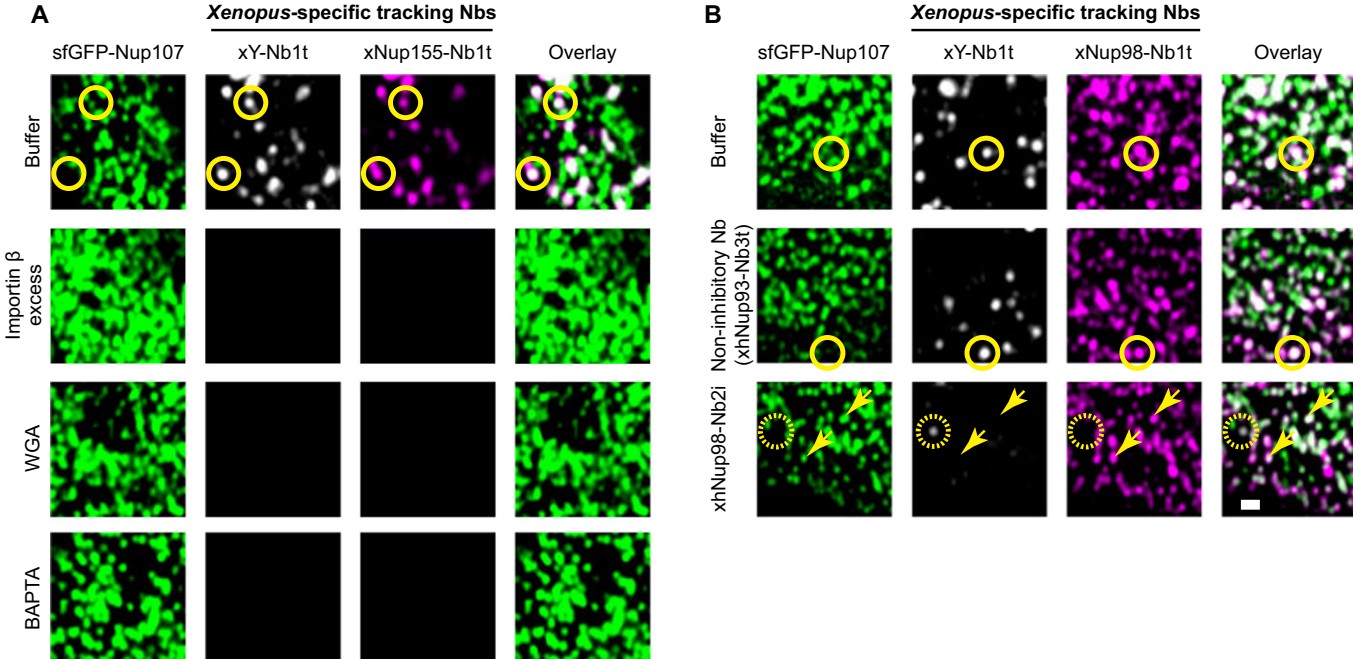

**Figure 12. xNup98-Nb2i blocks interphase assembly of new NPCs without preventing Nup98 from binding pre-existing pores.**

Interphase insertion of new *Xenopus* NPCs into human NEs was performed and tracked with fluorescent nanobodies as in Fig. 3, in the absence or presence of the indicated inhibitors. After 2 h at RT, cells were washed, fixed, and imaged as in Fig. 4. Newly assembled NPCs are marked by circles with solid lines, incomplete pore-like structures by circles with dashed lines, and pre-existing NPCs by arrows. Scale bar, 0.5 µm. (A) An excess of importin β (2 µM), WGA (2.5 µM) and BAPTA (5 mM) completely abolished the insertion of new pores. (B) xhNup98-Nb2i (2 µM) also blocked new NPC insertion but still allowed pre-existing human NPCs to recruit *Xenopus* Nup98. The control nanobody xNup93-Nb3t (2 µM) had no effect on NPC insertion. Source data are available online for this figure.

assembly process. To track postmitotic assembly, we use the well-established *Xenopus* egg extract system. The key challenge in tracking interphase assembly is to distinguish newly assembled from pre-existing NPCs. We overcame this problem by using human NEs as templates and inserting chemically distinct frog NPCs from an activated egg extract into them. The described experimental setup opens new avenues to study NPC assembly during interphase and to biochemically dissect this pathway.

Through AddGene, all nanobody plasmids will be made available to the community. Earlier, we reported the first generation of nanobodies directed against frog nucleoporins (Pleiner et al, 2015). We now demonstrate their utility for tracking NPC assembly from *Xenopus* egg extracts, which represents a major simplification of the assay. In addition, we present a new set of tracking nanobodies with broad cross-reactivity that stain *Xenopus* as well as human NPCs. These include nanobodies against Nup358, Nup133, Nup93, Nup35, and the Nup214·88·62 complex. They are suitable for super-resolution microscopy and allow to resolve NPC ring structures.

We also report the first NPC assembly-inhibiting nanobodies against three targets: Nup93, Nup155, and Nup98. All of them bind to epitopes that are initially accessible in soluble Nup complexes but become buried in protein–protein interaction interfaces during NPC assembly. The nanobodies thus block critical protein–protein interactions. Inhibitory nanobodies perturb nuclear pore formation by introducing a very small, yet specific change and are therefore excellent tools to complement the structural data of fully assembled NPCs (Fontana et al, 2022; Bley et al, 2022; Mosalaganti et al, 2022)

with insights into the function and assembly mechanisms of the different Nup components. We obtained them by a direct cross-species (frog-human) immunization and phage display strategy for binders of conserved and, thus, functionally relevant epitopes. This strategy is generally applicable and can be used to generate highly selective, protein-based inhibitors of also other molecular machines. We are currently extending the anti-NPC toolbox to all relevant subunits, also for capturing in-pathway intermediates, such as the immediately pre-membrane-fusion state of NPCs.

The inhibitory anti-Nup93 nanobody imposed a late block that impairs assembly of the cytoplasmic double-ring, as well as the cytoplasmic appendices. The resulting NPCs lack Nup358, the Nup214·88·62 complex as well as ~1/3 of their Nup98-APD-binding sites. They thus suffer a substantial loss of FG mass, causing defects in the permeability barrier and consequently in active import and export. Yet, this inhibitory nanobody still allowed rudimentarily functional NPCs.

By contrast, the inhibitory anti-Nup155 nanobodies prevented scaffold assembly, resulting in nuclei without nuclear import activity and a NE containing only a few off-pathway NPC assembly intermediates with derailed Nup compositions. This was not unexpected as Nup155 is a critical architectural element of NPCs. Surprisingly, however, the same scaffold phenotype was caused by anti-Nup98 nanobodies targeting its autoproteolytic domain and blocking interactions with Nup96 and the Nup88 β-propeller. If these anti-APD inhibitory nanobodies only interfered with the anchoring of Nup98, one would expect that the NPC scaffold (including the Y-complex, Nup93, Nup155, and Nup35) and other

FG Nups would still assemble correctly. Instead, however, they impose an obvious block at early stages of assembly (Figs. 7–9 and EV6). This suggests that Nup98 is not just a "late addition" to a pre-assembled NPC scaffold but is already involved in the earliest NPC biogenesis steps.

This scenario further suggests the existence of multiple Nup98 populations that are recruited at different assembly stages and through different qualities of Nup–Nup interactions. Some copies of Nup98 are likely recruited at the early stages of assembly through obligatory contacts of the Nup98-APD with Nup96, engage in additional contacts with other scaffold Nups, and thereby trigger the aforementioned assembly checkpoint for the NPC scaffold. We assume that these Nup98 molecules remain very stably attached.

Indeed, one candidate for such early inter-complex interactions can be prominently detected by affinity chromatography from nuclear assembly extracts (Appendix Fig. S5), where a Nup98[395–866] fragment (lacking 394 residues of its FG domain) retrieves not only the Y-complex (through the APD-Nup96 interaction) but also the Nup188·Nup93 complex with similar efficiency. The interaction with the Nup188·Nup93 complex was lost when the remaining 100 residues of the FG domain were also deleted (in Nup98[486–866]). This is consistent with previous reports of yeast Nup188 homologs interacting with GLFG repeats (Andersen et al, 2013; Onischenko et al, 2017). In our data, however, this interaction appears to be more sequence-specific and to favor an interaction with Nup188 over the paralogous Nup205. In addition, Nup98 is known to interact through its R3 motif with Nup155 (Lin et al, 2016; Bley et al, 2022; Mosalaganti et al, 2022).

This multitude of interactions renders the APD actually dispensable for steady-state Nup98 anchorage, which explains why the inhibitory anti-APD nanobodies still allow efficient Nup98 incorporation into pre-existing NPCs (Fig. 12B). Considering, however, the essential role of the APD during the first steps of NPC biogenesis, it is tempting to speculate that early Nup98–Nup96 interactions are special: they might serve as an assembly checkpoint, allowing scaffold assembly and pore formation to proceed only after a sufficient number of barrier-forming Nup98 molecules have been recruited, which in turn connect (at least) three key architectural elements of the NPC scaffold: the Y-complex, Nup155 and the Nup188·Nup93 dimer.

The bulk Nup98 population may arrive later in assembly to complete the "sealing" of the NPC central channel with its permeability barrier. This population is likely recruited through fewer contacts and thus weaker interactions—at least this is suggested by the reported mobility of Nup98 in photobleaching experiments (Griffis et al, 2002; Rabut et al, 2004) and by the ease with which they are exchanged in the presence of egg extract (Fig. 4B). We would assume that the later-arriving population is still heterogeneous, anchored by interactions between the Nup98 unstructured regions and Nup155, Nup188 and/or Nup205, cohesive interactions between FG domains, and binding of the Nup98-APD to still vacant Nup96 sites and to the later-recruited Nup88 (Lin et al, 2016; Bley et al, 2022; Mosalaganti et al, 2022).

It can be estimated that a barrier-free NPC with a channel diameter of 60 nm would cause 1 000 times more leakage of a GFP-sized molecule than a functional NPC (for derivation, see Ribbeck and Görlich, 2001). For larger molecules, this 'leakage ratio' would be even higher. Non-selective leakage is dominant, implying that already a small fraction (a few percent) of barrier-free NPCs would severely degrade the performance of the entire nuclear transport system. This would include a deterioration of the transport-driving RanGTP gradient, impaired nuclear exclusion of cytoplasmic components, and leakage of already imported soluble factors from the nucleus. The Nup98 checkpoint sketched here could be crucial in preventing such a situation, as it would avoid NEs with non-selectively open pores and ensure that only NPCs with selective permeability assemble.

# Methods

## Expression and purification of recombinant proteins in *E. coli*

Recombinant Nups were expressed in NEB Express *E. coli* cells (New England Biolabs, C2523). Nanobodies were expressed in *E. coli* SHuffle cells (New England Biolabs, C3026J) for disulfide bond formation and higher stability (Lobstein et al, 2016). Protein expression was induced by 30–100 μM isopropyl-β-D-thiogalactoside (IPTG) at 18–30 °C for 4–20 h. Next, cells were harvested, resuspended in resuspension buffer (50 mM Tris/HCl pH 7.4, 300 mM NaCl, 20 mM imidazole, 2 mM DTT), lysed by sonication, and the lysates cleared by ultracentrifugation. His-tagged proteins were immobilized onto pre-equilibrated $Ni^{2+}$ chelate matrixes, washed with resuspension buffer, and eluted by the addition of resuspension buffer supplemented with either 400 mM imidazole or the appropriate protease (500 nM bdNEDP1, 50 nM bdSENP1, or 50 nM SENP$^{EuB}$; Frey and Görlich, 2014; Vera-Rodriguez et al, 2019).

## Alpaca immunizations

Immunogens were expressed recombinantly in *E. coli* as $H_{14}$-NEDD8, His$_{14}$-scSUMO or His$_{14}$-SUMO$^{Eu1}$ fusions, purified by Ni-chelate affinity chromatography with protease elution and subsequent size-exclusion chromatography (SEC). Antigens were buffer-exchanged to a physiological, isotonic, isoosmolar buffer (150 mM NaCl, 20 mM Tris/HCl pH 7.4, 250 mM Sorbitol), mixed with either the commercial adjuvant Fama (GERBU Biotechnik GmbH, Cat. No.: 3030) or a squalene-based oil-in-water emulsion, and injected subcutaneously.

Three female alpacas, kept at the Max Planck Institute for Multidisciplinary Sciences, were immunized with 0.5–1.0 mg of the human and *Xenopus* Nups at 3–4 week intervals. Nups that had already been immunized by Pleiner et al (2015) were immunized two more times, whereas Nups that were used for the first time as immunization antigens were injected 3–4 times. Blood samples were taken 4–7 days after the last immunization. These alpaca projects have been approved by the animal welfare authority LAVES (reference numbers: 33.9-42502-05-13A351, 33.9-42502-05-17A220, and 33.19-42502-04-22-00210).

## Immune library constructions and phage display

The generation of phage-displayed nanobody immune libraries from lymphocyte RNA and selection of anti-Nup Nbs with biotinylated baits was performed as previously described (Pleiner et al, 2015, 2018).

The bait concentration was gradually decreased in subsequent panning rounds to a concentration below 1 nM. Specific nanobody inserts were cloned into $H_{14}$-NEDD8, $His_{14}$-scSUMO, or $His_{14}$-SUMO$^{Eu1}$ expression vectors and transformed into *E. coli* cells. Individual clones were sequenced, and representative nanobody sequences were chosen for expression and further characterization.

## Nanobody labeling with maleimide dyes

Fluorophore maleimides were conjugated to nanobodies through two or three engineered surface cysteines (including N- and C-terminal ones), as previously described (Pleiner et al, 2015, 2018). Briefly, purified nanobodies were supplemented with 15 mM dithiothreitol (DTT) for 15 min on ice. Next, nanobodies were buffer-exchanged to (de-gassed) maleimide-labeling buffer (300 mM NaCl, 50 mM potassium phosphate pH 6.8) and rapidly mixed with a 1.2-fold molar excess (over the ectopic cysteines) of the respective maleimide fluorescent dye. Labeling reactions proceeded for 40 min at 4 °C and the excess dye was removed using NAP-5 desalting columns (17-0853-01, GE Healthcare). Nanobodies to be used for confocal microscopy were labeled with Alexa Fluor 488 (Thermo Fisher, A10254), Alexa Fluor 568 (Thermo Fisher, A20341) or Alexa Fluor 647 (Thermo Fisher, A20347), and nanobodies to be used for STED microscopy were labeled with the STED fluorophores Abberior STAR 580 (Abberior, ST580-0003) or Abberior STAR 635P (Abberior, ST635P-0003) (Wurm et al, 2012). Labeling efficiencies (degree of labeling; DOL) were determined by UV–Vis spectroscopy (at 280 nm and the absorbance peak of the respective fluorophore) and validated by size shifts on SDS-PAGE.

## Postmitotic nuclear assembly from *Xenopus* egg extracts (Figs. 1, 7–9, EV5, and EV6)

*Xenopus* egg extracts were prepared as described in Hülsmann et al (2012). To initiate an assembly reaction, 10 µl egg cytosol were pre-incubated with energy mix (10 mM creatine phosphate, 0.5 mM ATP, 0.5 mM GTP, 50 µg/ml creatine kinase) and sperm chromatin for 15 min at 20 °C to allow for chromatin decondensation. As indicated, inhibitory nanobodies were present already during this step. Next, egg membranes (isolated by high-speed centrifugation during the egg extract preparation and further purified by flotation through a Nycodenz gradient) were added.

To follow NPC assembly in real time (Fig. 1C,D), 50 nM of the corresponding tracking nanobody was added, and images were acquired at the indicated time points using an LSM880 FAST Airyscan confocal microscope (Zeiss, Germany). To screen nanobodies for inhibition of NPC assembly (Figs. 7–9, EV5, and EV6), 2 µM of each nanobody was added to the egg cytosol fraction prior to chromatin and membrane addition or at the indicated time point. Alternatively, 2.5 µM WGA, 2 µM importin β, or 5 mM BAPTA was added to the egg cytosol fraction to inhibit NPC assembly. After adding purified egg membranes, nuclear assembly proceeded for 1 h at 20 °C, without shaking.

The formation of NPCs with an intact permeability barrier (Figs. 7 and EV5A) was tested by adding 1 µM IBB-MBP-GFP, 5 µg/ml DAPI, and either active (0.5 µM NES-GFP) or passive (0.5 µM MBP-mCherry) exclusion cargoes. Transport cargoes were incubated for another 30 min at 20 °C, and the resulting nuclei were analyzed

using a Leica SP8 confocal microscope (Leica, Germany). Note that the strength of the inhibition might vary depending on the precise time point in which the inhibitory nanobodies are added. The laid eggs are arrested in metaphase II of meiosis and activated by the addition of calcium to shift them back to interphase, whereby the assembly-inhibitory mitotic phosphorylations are removed then only slowly. For a complete inhibitory effect, nanobodies must be added at a very early time point and before the first interactions between Nups and Nup subcomplexes occur.

To resolve single pores on acquired images of assembled nuclei by confocal microscopy (Figs. 8, 9, and EV6), it is critical to ensure maximum epitope occupancy and use an imaging mounting medium that minimizes bleaching and matches the diffraction index of the objective immersion oil. Thus, a slightly modified staining protocol was used: 15 µl assembled nuclei were fixed with 25 µl 2.4% paraformaldehyde (PFA) for 5 min at RT, layered on top of 250 µl gradients (10% glycerol, 11% sucrose, 100 mM NaCl, 50 mM Tris/HCl (pH 7.4)) and centrifuged onto polylysine-coated coverslips (swingout rotor at $1400 \times g$ for 4 mins). Purified nuclei were washed with phosphate-buffered saline (PBS), permeabilized with 0.3% (v/v) Triton X-100 for 3 min at RT, and blocked with 0.5% bovine serum albumin (BSA). Next, nuclei were incubated with three tracking nanobodies recognizing different Nups and coupled to compatible fluorophores (i.e., Alexa Fluor 488, Alexa Fluor 568, and Alexa Fluor 647) for 30 min on ice. After washing off the excess nanobodies, SlowFade Gold Antifade Mountant (Thermo Fisher) mounting medium was applied. Images were acquired using an LSM880 and deconvolved by Airyscan Processing (Huff, 2015).

The utility of the here reported tracking nanobodies in following postmitotic NPC assembly was reproduced in >200 biological samples, which included experiments that addressed a range of questions—as detailed throughout the manuscript. The conclusion that the nanobodies xhNup93-Nb4i, xhNup98-Nb2i, xhNup98-Nb3i, xhNup155-Nb2i, and xhNup155-Nb3i impede NPC assembly is based on a total of >100 individual biological samples, analyzed in complementary ways, such as by documenting nuclear import and barrier defects, or failure in NPC assembly or changes in the composition of NPCs or NPC-like structures.

## Image analysis of postmitotically assembled nuclei (Figs. 7–9)

To measure the volume of the reconstituted nuclei (Fig. 7D), DAPI and IBB-MBP-GFP were added onto unfixed nuclei assembly reactions. Next, 2-channel z-stacks were acquired using a Leica SP8 microscope with a $\Delta z$ of 0.2 µm, and the acquired z-stacks were thresholded and segmented using the software KNIME version 4.1.0 (Berthold et al, 2009). Segmented images were used to measure the nuclei volume using the "Particle Analyzer" tool of the MorpholibJ Plugin (Legland et al, 2016) from FiJi/ ImageJ (Schindelin et al, 2012). Finally, the average and SD of at least 20 nuclei per sample from three independent experiments were calculated.

To quantify the active import, active exclusion, and passive exclusion (Fig. 7D), acquired images were filtered using FiJi/ ImageJ (Schindelin et al, 2012), thresholded using the Otsu algorithm (Otsu, 1979) and segmented. Segmented images were then used as a mask to measure the mean pixel intensity inside and outside the nuclei using the "Analyze particles" function from FiJi. Partition

coefficients were calculated as the ratio between the mean intensity inside and outside each nucleus. The mean and SD of at least 10 nuclei per sample from 4 independent experiments were calculated. To calculate the total import, the partition coefficient was multiplied by the volume of each nucleus.

To count the number of assembled NPCs (or Nup-positive structures) (Fig. 8C), the "Analyze particles" function from FiJi was used on segmented 3-channel images and the spots detected on each channel were counted. For each tracking nanobody, the average and SD of at least five images from three different experiments were measured.

To analyze the correlation between the signal intensities of different Nups (Fig. 9), nuclear pores and pore-like structures were first detected in the xY-Nb1t channel, and the signal intensity of each detected spot was measured by integration. The Y-complex-positive spots were then also used as a reference mask to integrate the signals in the other Nup-channels and thus to quantify their co-colocalization. Figure 9 illustrates these data as scatter plots, where each point represents one detected NPC or NPC-like structure. The point coordinates are the normalized signal intensities of the Y-complex, and the indicated second Nup. On each plot, the signal intensity distribution of the indicated inhibition condition is overlaid with that of the buffer control. The FiJi scripts mentioned above are included with the source data accompanying this manuscript.

## Semi-permeabilized cell assay for interphase NPC assembly (Figs. 3–5 and 12)

To label pre-existing (old) NPCs, HeLa P2 cells (Gunkel et al, 2021) were CRISPR/Cas9-edited to express Nup107 with sfGFP at its N-terminus (homozygous genomic tagging). The full characterization of this cell line will be reported elsewhere (PG and Volker Cordes, forthcoming). sfGFP-Nup107 cells were grown at 37 °C in high-glucose Dulbecco's modified Eagle's medium (DMEM) supplemented with 10% fetal bovine serum (FBS). One day before use, cells were seeded in a 10-well glass slide (543078, Greiner Bio One) at an appropriate density to reach 70–80% confluence at the time of the experiment.

sfGFP-Nup107 HeLa cells were semi-permeabilized by the addition of transport buffer (TRB; 110 mM potassium acetate, 3.5 mM magnesium acetate, 20 mM HEPES/KOH (pH 7.5), 1 mM EGTA, 250 mM sucrose) supplemented with 30 μg/ml digitonin for 3 min at RT with gentle shaking. This treatment perforates the plasma membrane for entry of Nups and nanobodies while leaving the NE intact. After washing off the digitonin, a *Xenopus* egg extract mixture was added to the cells. These mixtures contained 15 μl of the soluble fraction of *Xenopus* egg extracts, 1.8 μl energy mix (supplying final concentrations of 10 mM creatine phosphate, and each 0.5 mM ATP and GTP), two tracking nanobodies (recognizing different Nups and conjugated to compatible fluorophores) at a 35 nM final concentration each, and TRB to a final volume of 75 μl. To arrest interphase NPC assembly, final concentrations of 2 μM inhibitory nanobodies, 2.5 mM WGA, 2 μM importin β, or 5 mM BAPTA were included. The *Xenopus* egg extract mixture was incubated with the HeLa cells for 2 h at RT, with gentle shaking. The cells were washed three times with TRB and either imaged directly in transport buffer or fixed with 2.4% PFA for 5 min at RT.

For an optimal quality of the acquired microscopy images, fixed cells were washed with TRB + 0.3% (v/v) Triton X-100 for 3 min at RT, blocked with PBS + 1% BSA and subsequently stained with 35 nM of the same tracking nanobodies for 30 min at RT, with gentle shaking. After washing, SlowFade Gold Antifade Mountant was added to the cells to protect the dyes from photobleaching.

For Figs. 3, 4 and 12, an LSM880 FAST Airyscan confocal microscope was used. Three-channel images of nuclear cross-sections or 3-channel z-stacks along the bottom NE with a Δz of 0.2 μm were acquired. Acquired images were then Airyscan processed. Z-stack projections are shown. For super-resolution microscopy (Fig. 5; S1), cells were imaged using a Nikon inverted microscope (Ti-2) mounted with a STEDYCON system (Abberior Instruments, Germany) and deconvolved using the Huygens Professional software (version 19.10; Scientific Volume Imaging, The Netherlands) (Schoonderwoert et al, 2013).

The observation that incubation of digitonin-permeabilized HeLa cells with *Xenopus* egg extract leads to the insertion of frog NPCs into the human NE was reproduced in >100 biological samples.

## Immunofluorescence of human NPCs (Fig. 6A,B; Appendix Fig. S3)

sfGFP-Nup107 HeLa cells were grown on 10-well glass slides as above. For staining, they were either left unfixed and semi-permeabilized with 30 μg/ml digitonin as in Appendix Fig. S3. Alternatively, cells were fixed with 2.4% PFA for 3 min at RT and permeabilized with 0.3% (v/v) Triton X-100 (Fig. 6A). In both cases, cells were then blocked with 1% BSA for 30 min at RT and stained with 35 nM of the indicated Alexa Fluor 647-labeled tracking nanobodies for 30 min at RT. Finally, the excess of tracking nanobodies was washed off, and cells were imaged using a Leica SP8 confocal microscope (Leica, Germany). For STED microscopy (Fig. 6B), cells were imaged in SlowFade Gold Antifade Mountant. Images were acquired with the above-mentioned STEDYCON microscope and deconvoluted using the Huygens Professional software.

## RNA interference in HeLa cells (Fig. 10A)

HeLa cells were seeded at a low density on 10-well glass slides (543078, Greiner Bio One). The next day, cells were transfected with 10 nM of either control or Nup358 Silencer Select siRNAs (4390846 and s11774, respectively; Ambion, USA) using the HiPerFect transfection reagent according to the manufacturer's instructions (301704, QIAGEN, Germany) in serum-free medium (OptiMEM). RNA silencing proceeded for 72 h at 37 °C, and cells were then fixed, blocked, and stained with combinations of the indicated tracking nanobodies. Stained cells were imaged using a confocal SP8 microscope.

## Native purification of endogenous Nups using nanobodies (Fig. 6D)

Biotinylated Nups containing a SUMO[Eu1] cleavage site were immobilized to a Sepharose-streptavidin matrix (53113, Thermo Scientific) and incubated with the soluble fraction of either *Xenopus* egg extracts or HeLa lysates for 40 min at 4 °C. Next, unspecific

binders were washed off and the Nup-nanobody complexes were eluted by the addition of 50 nM SENP$^{EuB}$ (Vera-Rodriguez et al, 2019). The eluted proteins were analyzed by SDS-PAGE followed by Coomassie staining.

## Isolation of Nup98-interacting Nups from *Xenopus* egg extract (Appendix Fig. S5)

*Xenopus* Nup98 fragments were expressed as $H_{14}$-Avi-SumoSTAR fusions in *E. coli*, purified with intact tags, enzymatically biotinylated with BirA (Beckett et al, 1999), immobilized on streptavidin-agarose and incubated with the soluble fraction from *Xenopus* egg extract. After washing, formed Nup98·Nup complexes were eluted by tag-cleavage with 100 nM SumoSTAR protease (Peroutka et al, 2008).

## Binding assays with recombinant Nups and nanobodies (Fig. 11D; Appendix Fig. S4A, B)

Purified nanobodies carrying an N-terminal His-tag and a protease cleavage site (NEDD8, see also Frey and Görlich, 2014) were immobilized onto pre-equilibrated $Ni^{2+}$ chelate beads for 45 min at 4 °C. Next, excess nanobody was washed off with washing buffer and equimolar amounts of the untagged Nup target were added for 30 min at 4 °C. Excess Nups were then washed off and nanobodies were eluted along with their bound Nup targets by the addition of the appropriate protease (NEDP1). The eluted protein fractions were analyzed by SDS-PAGE followed by Coomassie staining.

## Biolayer interferometry (BLI) affinity measurements (Fig. 2A and Appendix Fig. S2)

BLI experiments were performed using an Octet RED96e instrument (ForteBio/Sartorius, Germany) and High Precision Streptavidin biosensors at 25 °C and using PBS (pH 7.4), 0.02% (w/v) Tween-20 and 0.1% BSA as assay buffer. $K_D$s listed in Fig. 2A were determined by modifying nanobodies (containing two ectopic cysteines) with Biotin-[PEG]$_3$-maleimide (Iris Biotech) and immobilizing them to a binding signal of about 1 nm on High Precision Streptavidin sensors. Target proteins (*Xenopus* or human) at 30–100 nM were allowed to bind for 600–1200 s and to dissociate for 600–1200 s, as indicated in the letterings. Apparent $K_D$s were fitted using the Octet Analysis HT 12.0 software.

## Crystallization and structure determination of the *h*Nup35·xhNup35-Nb1t complex (Fig. EV4)

Purified untagged *h*Nup35 (residues 173–248, corresponding to the RRM domain) was incubated with a molar excess of His$_{14}$-NEDD8-xhNup35-Nb1t. The formed complex was immobilized onto a $Ni^{2+}$ chelate matrix, eluted by 0.5 µM NEDP1 protease and further purified by SEC on a Hi-Load Superdex 75 16/60 column eluted in 100 mM NaCl, 10 mM Tris/HCl (pH 7.5), and concentrated to 10 mg/ml.

The optimal crystallization condition was 0.05 M HEPES (pH 6.5); 25% PEG400 (v/v); 0.05 M NaCl, 0.01 M MgCl$_2$. The crystals were cryoprotected by soaking in reservoir solution with increasing amounts of PEG400 up to 40% for 30 min, directly harvested from the robotic plate, mounted in loops, and flash-frozen in liquid nitrogen. Datasets were collected at EMBL beamline P14, PETRA

III storage ring (DESY, Hamburg, Germany) using a EIGER X 16 M detector (DECTRIS). The obtained data were processed with XDS (Kabsch, 2009), *XSCALE* (Diederichs, 2006), and *POINTLESS/ AIMLESS* (Evans and Murshudov, 2013). Initial phases were obtained by molecular replacement in PHASER (McCoy, 2006) using the model coordinates of PDB 4LIR and 2X1O as references. Automatic model building was done using warp (Langer et al, 2008) followed by iterative manual building in *Coot* (Emsley et al, 2010). The structure of the complex was refined using REFMAC5 (Murshudov et al, 2011). Data collection and final refinement statistics are summarized in Appendix Table S1.

## Crystallization and structure determination of the Nup93 nanobodies (Fig. 10)

Homology modeling (including AlphaFold) did not yield structures of xhNup93-Nb4i and xNup93-Nb2t that allowed a plausible docking into the cryo-EM map of the Nup93-nanobody complex (see below). Therefore, we first solved the structures of the unliganded nanobodies by X-ray crystallography. The two nanobodies were expressed as His$_{14}$-bdSUMO fusions, purified by $Ni^{2+}$ chelate chromatography and on-column protease cleavage, followed by SEC on Superdex 75 in 100 mM NaCl, 20 mM Tris/HCl (pH 7.4), and concentrated to 10 mg/ml.

Crystallization conditions were 100 mM imidazole (pH 8.0), 30% PEG8000, 200 mM NaCl for xhNup93-Nb4i (PDB 8CDS), and 100 mM CAPS (pH 10.5), 1.2 M sodium phosphate, 0.8 M potassium phosphate, 200 mM lithium sulfate for xNup93-Nb2t (PDB 8CDT) at 20 °C. Crystals were cryoprotected in the precipitant solution containing 15 or 25% glycerol (v/v), respectively, and plunge-frozen in liquid nitrogen.

Diffraction data were collected at the PXII-X10SA beamline at the Swiss Light Source (SLS, Paul Scherrer Institute, Villigen, Switzerland) at 100 K, using an EIGER2 X 16 M detector (DECTRIS). The datasets were processed in XDS (Kabsch, 2009) and the Phenix Package (Liebschner et al, 2019) was used throughout structure solving. The structures were solved by molecular replacement in Phaser (McCoy, 2006). The search model was generated in Sculptor (Bunkóczi and Read, 2011) with the nanobody Re5D06 (PDB 7ON5) as a structural template (Güttler et al, 2021). Iterative cycles of model building/adjustment in Coot (Emsley et al, 2010) and refinement in phenix.refine (Afonine et al, 2018; Liebschner et al, 2019) were performed until convergence to the final models (Appendix Table S2). Water molecules were built manually in Coot, where the 2Fo−Fc and Fo−Fc maps (contoured at 1σ or 3σ, respectively) showed a peak within a 2.2–3.5 Å distance to oxygen or nitrogen atoms of the protein molecules or other water molecules/ligands. Secondary structure restraints (with isotropic B-factor refinement) were used in early cycles, and restraints were relaxed in later iterations, for which B-factors of all non-water atoms were refined anisotropically.

## Cryo-EM structure of the Nup93·xhNup93-Nb4i·xNup93-Nb2t complex (Fig. 10)

Purified untagged xNup93 (α-solenoid domain, corresponding to residues 168–820) was incubated with a molar excess of purified untagged xNup93-Nb2t and His$_{14}$-bdNEDD8-xhNup93-Nb4i. After immobilization to a $Ni^{2+}$ chelate matrix and washing off

excess components, the equimolar trimeric xNup93·xhNup93-Nb4i·xNup93-Nb2t complex was eluted by NEDP1 protease and further purified by size-exclusion chromatography using a Hi-Load 26/60 Superdex 200 column in 50 mM NaCl, 20 mM Tris/HCl (pH 7.5), 2 mM DTT.

For cryo-EM analysis, the purified complex was diluted to 1.7 mg/ml with 25 mM Tris/HCl (pH 8.0), 50 mM NaCl, 1 mM DTT. Two µl of the diluted complex were applied to freshly glow-discharged R 2/2 holey carbon grids (Quantifoil), which were blotted with a force of 6 for 5 s and immediately vitrified by plunge-freezing in liquid ethane using a Vitrobot Mark IV (Thermo Fisher) operating at 4 °C and 100% humidity. The samples were initially screened using a Glacios cryo-electron microscope (Thermo Fisher) operated at 200 kV with a Falcon-III direct electron detector (Thermo Fisher). Next, a Titan Krios transmission electron microscope (Thermo Fisher) operating at 300 kV with a K3 Summit direct detector (GATAN) and a GIF Quantum Filter was used to obtain the final cryo-EM data. Data were acquired using SerialEM (Mastronarde, 2005) in EFTEM mode with an energy filter slit width of 20 eV and recorded as counting image stacks of 40 movie frames, with a total electron dosage of 54.41 $e^-$/Å² at a magnification of 105,000×, equivalent to a calibrated pixel size of 0.834 Å. 13 001 movie stacks were imaged with a defocus range of −1.5 to −2.5 µm. Warp (Tegunov and Cramer, 2019) was used for motion correction, dose weighting, contrast-transfer function estimation, and particle selecting on the fly. RELION 3.1. (Zivanov et al, 2020) was used for the rest of the image processing. Particles were extracted in a 128-pixel box, binned by 2 at 1.668 Å per pixel, and subjected to reference-free two-dimensional (2D) classifications. A de novo model was created from the good 2D class averages using the 3D Initial Model function, and it was then lowpass filtered and utilized as a low-resolution reference for two rounds of three-dimensional classification, yielding 202,725 particles. These particles were re-extracted in a box of 256 pixels without binning and subjected to global 3D auto-refinement, yielding a 4.8 Å map. The set of particles was subject to one round of CTF refinement and Bayesian polishing. The final global 3D refinement was carried out using the polished particles, which resulted in a 4.4 Å reconstruction.

To build an initial model for the trimeric complex, an AlphaFold2 model (Jumper et al, 2021) of the Nup93 structure (AF-Q7ZX96) and the crystal structures of xNup93-Nb2t and xhNup93-Nb4i were rigid-body fitted in the 4.4 Å map using UCSF Chimera (Pettersen et al, 2004). The fitted models were subsequently subjected to several rounds of MDFF flexible fitting in Namdinator (Kidmose et al, 2019). The final model was modified using Coot (Casañal et al, 2020). PHENIX (Afonine et al, 2018) was used to refine the model in real space and MolProbity (Williams et al, 2018) was used to assess it. Details for data collection and refinement statistics are listed in Appendix Table S3.

### Crystallization and structure determination of the xNup98-APD·xhNup98-Nb2i and xNup98-APD·xhNup98-Nb3i complexes (Fig. 11)

Purified untagged xNup98-APD (residues 714–864) was incubated with a molar excess of either His$_{14}$-NEDD8- xhNup98-Nb2i or His$_{14}$-NEDD8-xhNup98-Nb3i. Next, the formed complexes were immobilized onto a Ni$^{2+}$ chelate matrix and eluted by the addition of 0.5 µM NEDP1 protease. The obtained complexes were further purified by size-exclusion chromatography on a Hi-Load Superdex

75 16/60 column ÄKTA column, equilibrated in 100 mM NaCl, 10 mM Tris/HCl (pH 7.5), concentrated to 10 mg/ml, and screened for crystallization.

The optimal crystallization condition was 2.5% PEG6000, 25% (v/v) PEG MME500, 100 mM Tris/HCl (pH 9). Crystals were flash-frozen in liquid nitrogen without additional cryoprotection. Diffraction data was collected remotely from the beamline PXII at the Swiss Light Source (SLS; Paul Scherrer Institute, Switzerland) and the structure was solved by molecular replacement using the previously published structure of the xNup98-APD·xNup98-Nb1t complex (PDB: 5E0Q) as a search model. Data collection and final refinement statistics are detailed in Appendix Table S4.

### Software used for the preparation of figures

Adobe Illustrator was used for assembling figures, GraphPad Prism (version 9) for plotting data, ImageJ2 (version 2.3.0)/Fiji for analyzing and quantifying microscopic images, Photoshop for the proportional adjustment of brightness of SDS-PAGE gels (original images are included with the source data), and PyMOL for visualization of protein structures.

## Data availability

The coordinates and structure factors of the here reported structures have been deposited in the PDB database (www.rcsb.org) with the following accession codes: 8OZB (xhNup35-Nb1t complexed to the human Nup35 RRM homodimer); 7NQA (xNup98-APD·xhNup98-Nb2i complex); and 7NOW (xNup98-APD·xhNup98-Nb3i complex).); 8CDS (xhNup93-Nb4i); 8CD7 (xNup93-Nb2t); 7ZOX (xNup93·xhNup93-Nb4i·xNup93-Nb2t complex). EMD-14849 is the corresponding accession code for the Nup93·xhNup93-Nb4i·xNup93-Nb2t complex in the EMDB (www.ebi.ac.uk/emdb/).

## Peer review information

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

## Acknowledgements

The authors thank Ulrike Teichmann and her team for alpaca care, immunizations, and blood samplings, Jens Krull and Bianka Mußil for helping in the preparation of the nanobody libraries, Renate Rees for providing the Nup133 and Mamta Jaiswal for providing the Nup214 complex antigens. The authors also thank Jens Krull, Bastian Hülsmann, and Lareen Gräser for the preparation of *Xenopus* egg extracts and Henning Urlaub and team for mass spectrometric protein identification. The authors thank Volker Cordes for *Xenopus*-specific anti-Nup153 antibodies and for supporting the sfGFP-Nup107 cell line development, as well as Antonio Politi (Imaging facility of the institute) for assistance with image analysis. The authors are grateful to the crystallization and cryo-EM facilities of our institute for their excellent support. The authors thank Paloma Tarrío Alves for the scientific illustrations. This work received funding from the Max Planck Society, the Boehringer Ingelheim Fonds (BIF Ph.D. fellowship to MSC), and an ERC Synergy grant (StuDy SARCOMERE to DG).

## Author contributions

**Mireia Solà Colom**: Conceptualization; Resources; Data curation; Formal analysis; Validation; Investigation; Visualization; Methodology; Writing—original draft; Writing—review and editing. **Zhenglin Fu**: Validation; Investigation. **Philip Gunkel**: Resources; Methodology; Writing—review and editing. **Thomas Güttler**: Validation; Investigation. **Sergei Trakhanov**: Validation; Investigation. **Vasundara Srinivasan**: Validation; Investigation. **Kathrin Gregor**: Data curation; Investigation; Visualization. **Tino Pleiner**: Resources; Supervision; Investigation; Methodology. **Dirk Görlich**: Conceptualization; Resources; Formal analysis; Supervision; Funding acquisition; Methodology; Writing—original draft; Project administration; Writing—review and editing.

## Funding

## Disclosure and competing interests statement

MSC, KG, and DG are listed as inventors on a European patent application related to the use of anti-human NPC nanobodies. The remaining authors declare no competing interests.

# Expanded View Figures

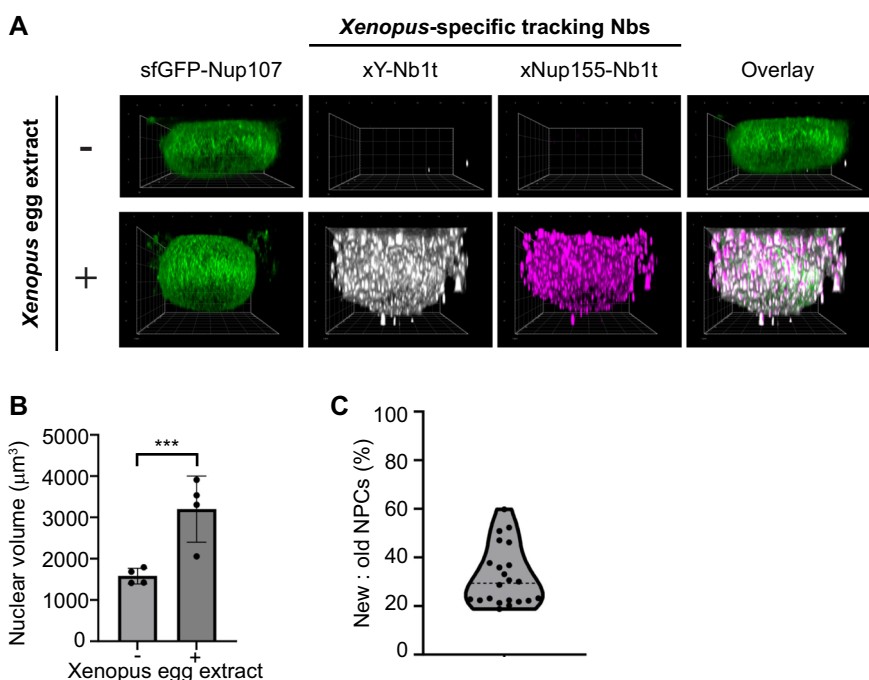

**Figure EV1.    HeLa cell nuclei grow in volume in the presence of *Xenopus* egg extracts.**

(**A**) Interphase insertion of *Xenopus* NPCs from egg extract into human nuclei was performed as in Fig. 3. 3D reconstructions of nuclei were obtained from the acquired 3-channel z-stacks using the Arivis Vision4D software (version 3.1.3; VisionVR, 2020). (**B**) Volumes of 4 individual nuclei per condition from two independent experiments were integrated in the GFP channel using FiJi and plotted. Incubation with egg extract increased the average nuclear volume from 1500 to 3000 μm$^3$. *** significant difference with a *P* value of 0.008 (unpaired *t* test). (**C**) Quantification of new NPC insertion during interphase assembly. NPCs were detected and quantified using a FiJi script. Numbers are normalized to the number of pre-existing human NPCs ( = 100%). Each dot represents one quantified nucleus. Source data are available online for this figure.

**A**

**Arg103 at the CDR3 of xNup98-Nb1t is shape-complementary to xNup98 but clashes with a Lys residue on hNup98**

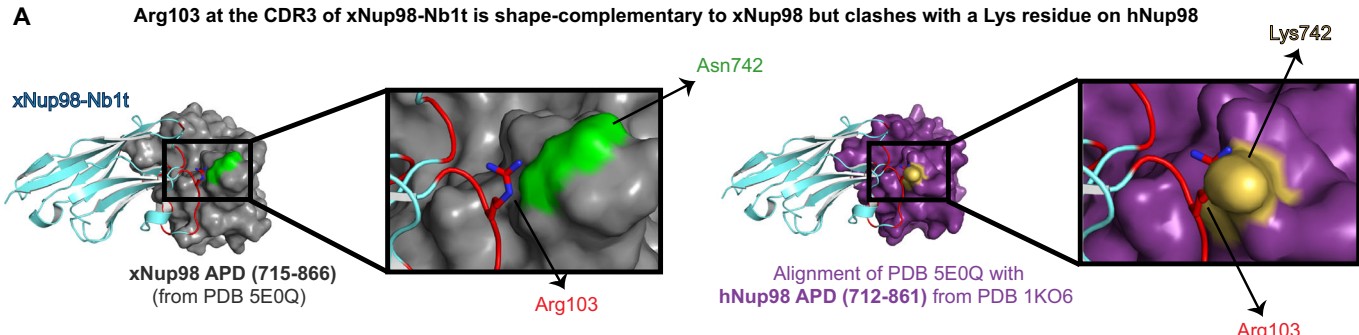

**B**

**xNup93-Nb2t is shape-complementary to xNup93 but clashes with the protein backbone of hNup93**

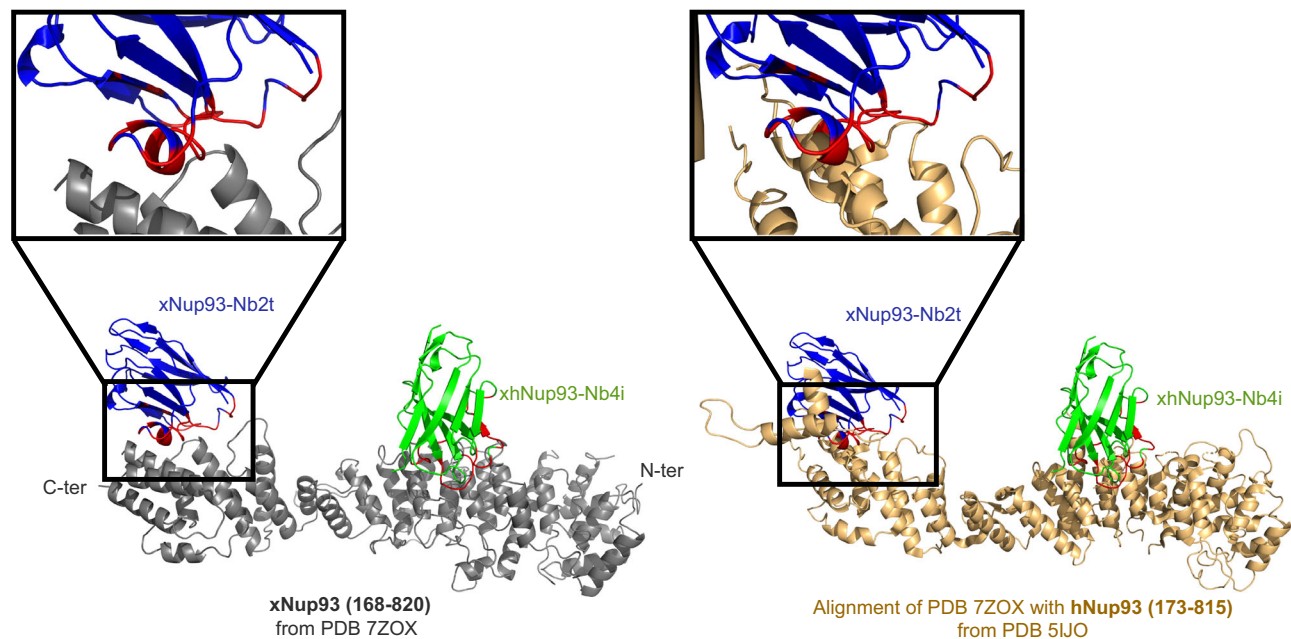

**Figure EV2. *Xenopus*-specific anti-Nup nanobodies are incompatible with the corresponding human Nup targets.**

(**A**) The crystal structure of the xNup98–xNup98-Nb1t complex shows that an arginine residue at the nanobody's CDR3 interacts with an asparagine residue at the APD of *Xenopus* Nup98 (left) (PDB 5E0Q; Pleiner et al, 2015). However, this arginine would clash with a lysine at the same position in hNup98, preventing its binding (right). (**B**) xNup93-Nb2t fits well to a pocket at the C-terminus of *Xenopus* Nup93 (PDB 7ZOX, Fig. 10), but would clash with the backbone of human Nup93 at the same protein region.

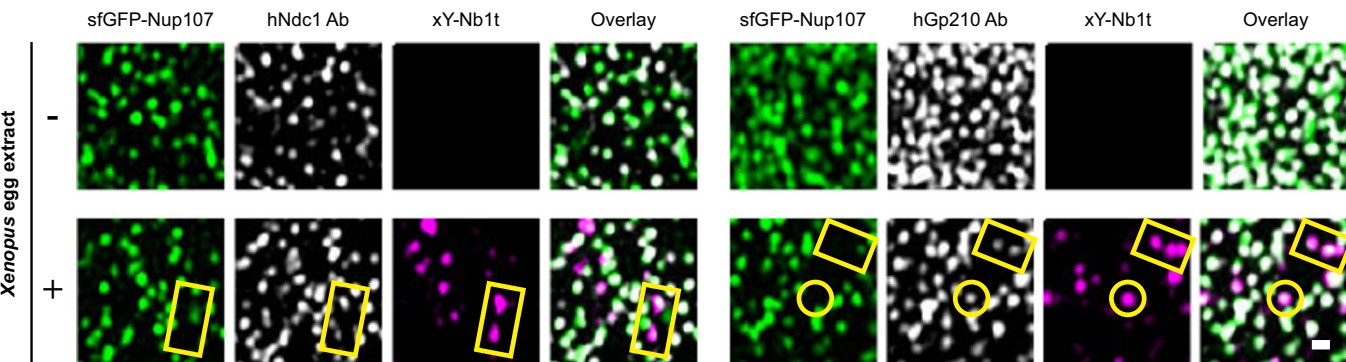

**Figure EV3.  Human membrane Nups get assembled into *Xenopus* NPCs.**

Cells were stained with *Xenopus*-specific x-Ycmplx-Nb1t coupled to Alexa Fluor 647 and antibodies against human Ndc1 and human Gp210 (Stavru et al, 2006a, 2006b). Images were acquired as in as in Fig. 4. Circles or rectangles mark newly inserted pores. Scale bar, 0.25 µm. Source data are available online for this figure.

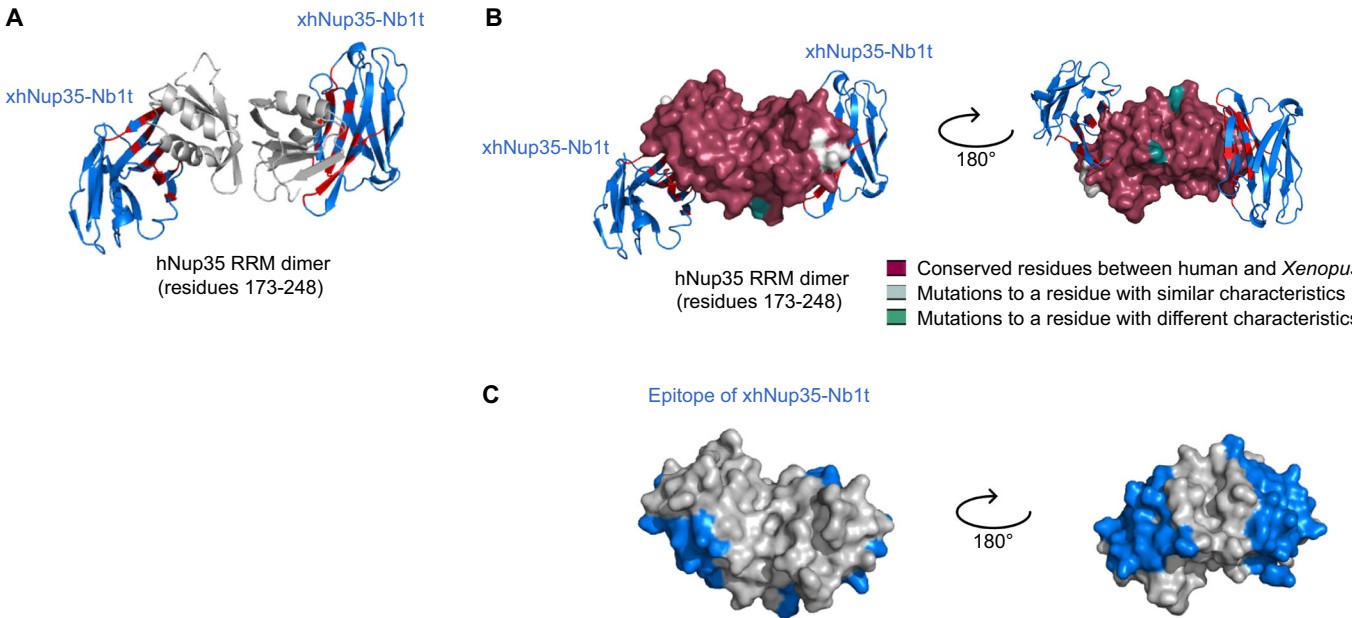

**Figure EV4. xhNup35-Nb1t recognizes a conserved epitope at the RRM domain of Nup35.**

(A) Crystal structure of the homodimeric RRM domain of *Homo sapiens* (h)Nup35 (gray) in complex with xhNup35-Nb1t (blue). The nanobody paratope is highlighted in red. See Appendix Table S1 for crystallographic statistics. (B) Surface representations of the Nup35 RRM domain color-coded according to amino acid conservation between human and *Xenopus* Nup35. The RRM domain and the nanobody epitope are highly conserved, as also indicated by the nanobody's cross-reaction between human and *Xenopus* Nup35 (see Figs. 4 and 6A, B). (C) The xhNup35-Nb1t epitope is highlighted in blue to allow for a straightforward comparison with (B). Although the Nup35 RRM domain mediates Nup35 homodimerization, which is necessary for the assembly of functional NPCs (Vollmer et al, 2012), xhNup35-Nb1t recognizes an epitope distant from the dimerization interface and is compatible with the Nup35 dimerization, explaining why it does not interfere with the formation of functional NPCs. In addition, the Nup35 dimer associates with membranes directly or through Ndc1 at very early assembly steps (Vollmer et al, 2012; Eisenhardt et al, 2014), and it links Nup93 to the β-propeller of Nup155, which is essential for the organization of the NPC inner ring and the assembly of the NPC scaffold (Hawryluk-Gara et al, 2008; De Magistris et al, 2018). However, these critical interactions occur through short linear motifs located at the Nup35 disordered termini (Vollmer et al, 2012; Eisenhardt et al, 2014; Mosalaganti et al, 2022), again explaining why xhNup35-Nb1t does not impede NPC assembly. Indeed, the lack of a phenotype is consistent with homodimerization being the only essential function of this RRM domain.

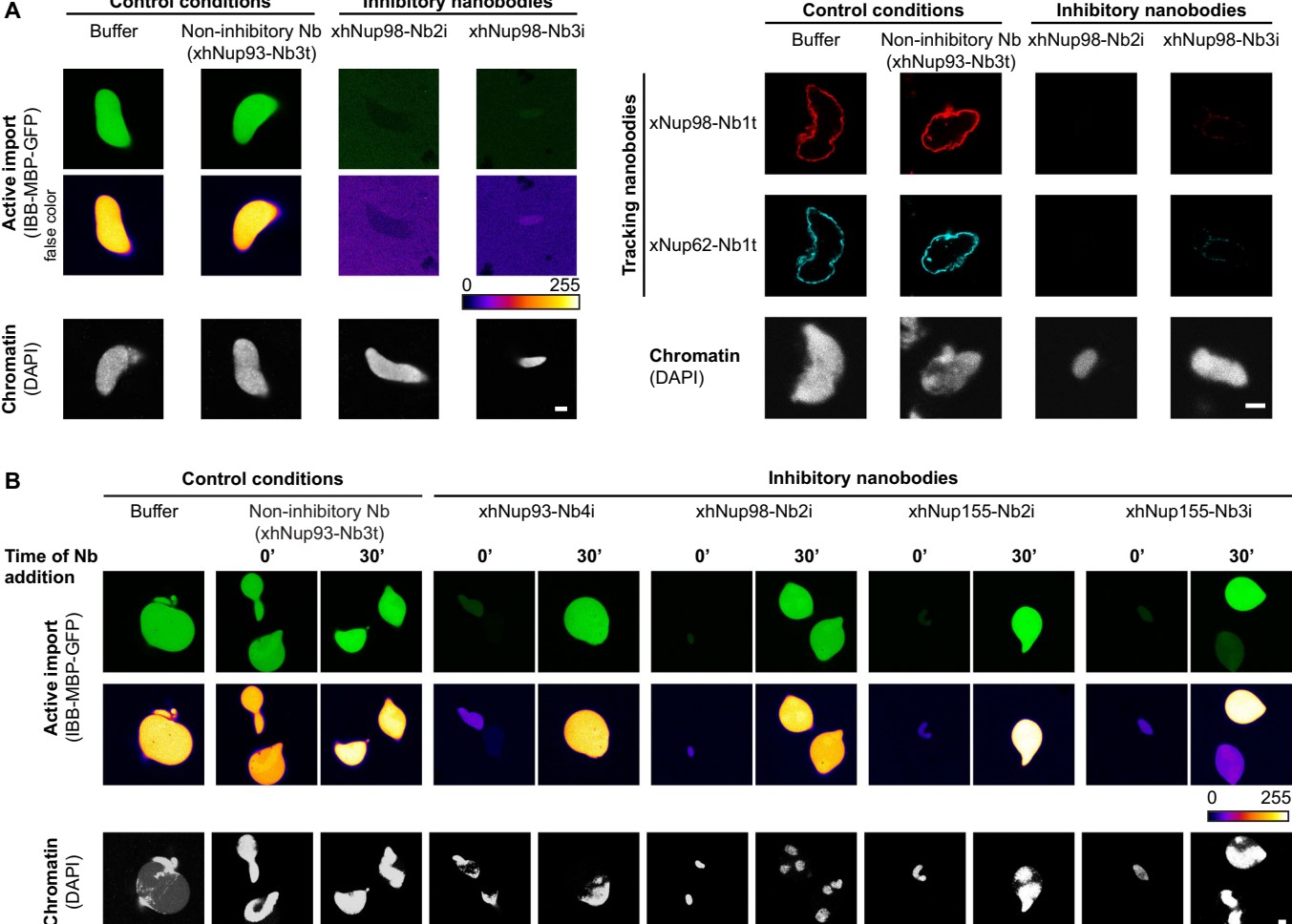

**Figure EV5.  Inhibitory anti-Nup nanobodies disrupt NPC assembly but do not block nuclear transport directly.**

(A) 2 µM nanobodies were added to postmitotic NPC assembly reactions (as in Figs. 7 and 8). xhNup93-Nb3t had no deleterious effect. The presence of xhNup98-Nb2i and xhNup98-Nb3i, however, resulted in pseudonuclei that failed in active nuclear import of the IBB-MBP-GFP fusion (left) and in Nup recruitment to the NE (right). The coherent phenotype of the two inhibitory anti-Nup98 nanobodies is a stringent specificity control to rule out off-target effects as a cause, since the two belong to different classes and recognize different, though overlapping, epitopes (Fig. 11). They probably act by preventing the interaction of Nup98 with the Y-complex component Nup96 and with Nup88 (Fig. 11). Scale bar, 5 µm. (B) Nuclei were assembled as in Figs. 7 and 8, and inhibitory nanobodies were added either prior to chromatin addition and assembly initiation (0') or thereafter (30 min after membrane addition). Next, IBB-MBP-GFP and DAPI were added, and nuclei were imaged 90 min later. Note that the late addition of inhibitory nanobodies allowed for very efficient active import, ruling out that the nanobodies interfere directly with import through functional NPCs. Scale bar, 5 µm. Source data are available online for this figure.

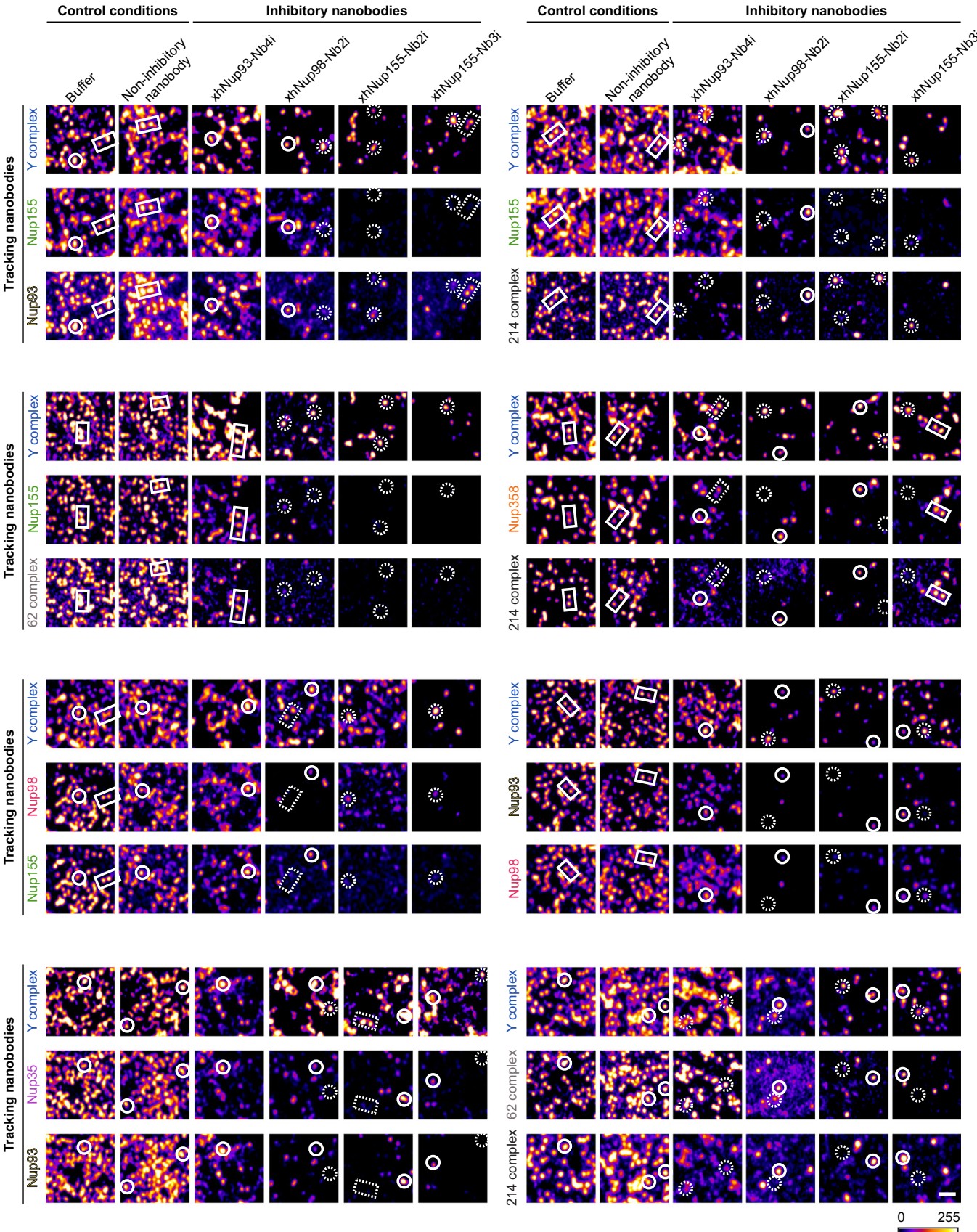

**Figure EV6.  The assembly-inhibited 'NPCs' are decreased in number and altered in composition.**

Postmitotic nuclear assembly in the presence of inhibitory nanobodies was as in Figs. 7 and 8, but 3-channel close-up views with single NPC resolution are shown. The following tracking nanobodies were used for staining: xY-Nb1t, xNup155-Nb1t, xNup93-Nb1t, xhNup35-Nb1t, xNup98-Nb1t, xNup62-Nb1t, xhNup214-Nb1t or xNup358-Nb1t. The non-inhibitory nanobodies used as controls were xhNup93-Nb3t, xNup93-Nb1t, xhNup35-Nb1t, xNup155-Nb1t, xNup358-Nb1t or xNup62-Nb1t. Pore-like structures, in which all three simultaneously probed Nup components were detected, are highlighted with white solid lines. Arrested structures with missing Nup components are highlighted with white dashed lines. A false-color representation (LUT) is shown to facilitate a comparison between the different images. Scale bar, 1 µm.

