## [Peer Review File · The EMBO Journal]

A checkpoint function for Nup98 in nuclear pore formation suggested by novel inhibitory nanobodies

Dirk Görlich, Mireia Solà Colom, Zhenglin Fu, Philip Gunkel, Thomas Güttler, Sergei Trakhanov, Vasundara Srinivasan, Kathrin Gregor, and Tino Pleiner

Corresponding author(s): Dirk Görlich (goerlich@mpibpc.mpg.de)

Review Timeline:

Submission Date:	12th Sep 23
Editorial Decision:	6th Oct 23
Revision Received:	13th Feb 24
Editorial Decision:	5th Mar 24
Revision Received:	11th Mar 24
Accepted:	13th Mar 24

Editor: Hartmut Vodermaier

Transaction Report:

Dr. Dirk Görlich
MPI for Multidisciplinary Sciences, Göttingen
Cellular Logistics
Am Fassberg 11
Göttingen, Germany 37077
Germany

6th Oct 2023

Re: EMBOJ-2023-115591
Nucleoporin-binding nanobodies that either track or inhibit nuclear pore complex assembly

Dear Dirk,

Thank you again for submitting your manuscript on new nucleoporin-tracking/inhibiting nanobodies and their cross-species application to study NPC assembly. I have now received the below-copied comments from three expert referees, in light of whose positive feedback we shall be happy to pursue this work further for EMBO Journal publication. As you will see, the referees do not ask for major additional experiments, but their key data concern is a request for more quantitation of several currently qualitative statements. Their other points are mainly related to discussion, interpretation and presentation, and would in principle appear straightforward to address. I am therefore inviting you to prepare a revised version incorporating the specific suggestions of the three referees.

Perhaps unsurprisingly, the referees have diverging opinions on the overall format and length of the study, with referee 2 preferring a more concise presentation, while referee 1 (during our pre-decision cross-commenting session) indicated reading somewhat longer manuscript appearing justified in order to properly elaborate on initial concepts and strategies. I personally also don't share referee 2's concerns regarding the title, as I think it is quite apt and does not unnecessarily under-sell the study's insights; likewise, the abstract does in my view do an excellent job in outlining concepts and results of the work in a clear and concise fashion.

Regarding figures, we would have an attractive option to include an additional level between the main and the Appendix (supplemental) figures currently included - Expanded View figures. These would be one level below the main figures, but would still be typeset and produced like main figures, and -importantly- be directly accessible in the online (HTML) version of the paper, as well as forming part of a downloadable "PDF+". While EV figures are normally limited to 5, we could in this case exceptionally include 6-7 if needed. Therefore, you may choose some of the currently 12 main figures (or certain panels of them - e.g. 3C and 6 mentioned by ref 2) to become EV figures; maybe also Fig 9 could become an EV figure, in light of referee 3's comments on this way of presentation (which are again not fully shared by referee1). Similarly, you could choose some of the current Figures S1-S8 to be promoted into EV figures - the remaining "supplemental" figures would then move into a separate "Appendix" PDF file together with their legend (please refer to our detailed Guide to Authors and the sections below on how to prepare/format revised manuscripts). Finally, I feel that current Figure 2A could easily be converted into a black-and-white table in the main text, given that it contains only textual information and the included colors are not further referred to.

Should you have any questions or would like to discuss any specifics on how to respond to the referee comments and to prepare this revision, please do not hesitate to contact me anytime. Thank you again for the opportunity to consider this work for The EMBO Journal, and I look forward to your revision in due time.

With kind regards,

Hartmut

1) Every manuscript requires a Data Availability section (even if only stating that no deposited datasets are included). Primary datasets or computer code produced in the current study have to be deposited in appropriate public repositories prior to

resubmission, and reviewer access details provided in case that public access is not yet allowed. Further information: embopress.org/page/journal/14602075/authorguide#dataavailability

9) Digital image enhancement is acceptable practice, as long as it accurately represents the original data and conforms to community standards. If a figure has been subjected to significant electronic manipulation, this must be clearly noted in the figure legend and/or the 'Materials and Methods' section. The editors reserve the right to request original versions of figures and the original images that were used to assemble the figure. Finally, we generally encourage uploading of numerical as well as gel/blot image source data; for details see: embopress.org/page/journal/14602075/authorguide#sourcedata

At EMBO Press, we ask authors to provide source data for the main manuscript figures. Our source data coordinator will contact you to discuss which figure panels we would need source data for and will also provide you with helpful tips on how to upload and organize the files.

In the interest of ensuring the conceptual advance provided by the work, we recommend submitting a revision within 3 months (4th Jan 2024). Please discuss the revision progress ahead of this time with the editor if you require more time to complete the revisions. Use the link below to submit your revision:

Link Not Available

Referee #1:

Colom et al. describe in their manuscript "Nucleoporin-binding nanobodies that either track or inhibit nuclear pore complex assembly" nanobodies against nucleoporins as a versatile tool to investigate the mechanisms of NPC assembly. For this they selected and characterized two sets of antibodies. The first one is specific for the respective *Xenopus* nucleoporins and does not cross-react with the human proteins. They can be used in "classical" nuclear/NPC assembly assays using *Xenopus* egg extracts and a novel developed assays using *Xenopus* egg extracts and tissue culture derived human nuclei to follow *Xenopus* NPC

integration and assembly into the nuclear envelope of the human nucleus. With these tracking antibodies the sequential assembly of NPCs can be followed. The second set constitutes nanobodies reacting with both *Xenopus* and human nucleoporins and some of these can be used to arrest NPC assembly at specific stages. Based on the here solved structures of the nanobody-target complexes specific hypotheses on their mode of inhibitory action are generated, which shed light on which nucleoporin-nucleoporin interactions are specifically needed at certain NPC assembly steps and are nicely reinforced by biochemical nucleoporin-nucleoporin interaction studies.

Overall, this study is very thoroughly conducted, includes a wealth of experiments and provides new insights into how NPC assemble.

specific major concerns essential to be addressed to support the conclusions:

Often, a single representative nucleus/experiment is shown to support the conclusions, e.g. Fig 3B. the authors should provide some quantitation, e.g. for Fig 3B the number of colocalizing Nup155 and Y-complex spots in comparison to single labeled spots from several nuclei. It appears that the nucleus presented shows less Y-complex spots as compared to Nup155 spots. Does this say something about the assembly order?

Also for the experiments presented in Fig 4 and 12, quantitation would support the statements.

minor concerns that should be addressed:

Page 8: "the initially flat HeLa nuclei became rounder; they grew in volume, suggesting that a massive import of *Xenopus* nuclear proteins had occurred, inflating nuclei through their osmotic-colloidal pressure (Fig 3C)." Nuclear volume measurements would support this statement.

Page 9 "The latter would be consistent with the reported mobility of these Nups (Griffis et al., 2002a, 2004)." Rabut et al could be also cited here (DOI: 10.1038/ncb1184)

Page 9: "Finally, by staining newly-assembled pores with antihuman Ndc1 and -Gp210 antibodies, we confirmed that human membrane Nups present in the HeLa cells support the assembly of the new *Xenopus* NPCs.": What is meant by support the assembly? This might be rephrased given the controversial role of Gp201 and Ndc1 in (postmitotic) NPC assembly.

Page 10: "To confirm the insertion of properly assembled NPCs, we acquired higher-resolution images of the HeLa NE containing newly-assembled pores, using two-colour STED (stimulation emission depletion) microscopy." The staining shows only y-complex structure. How do the authors know that those structures are fully assembled and functional NPCs?

Page 13: "As expected, nanobodies recognizing Nups of the NPC outer rings (e.g., Nup358, Nup133) labeled rings with larger diameters than nanobodies against inner ring components (e.g., Nup93, Nup35) (Fig 6B)." A robust quantitative analysis of the diameters would strengthen this point.

Page 15: " Compared to the control, all assembly-arrested nuclei contained a moderately (anti-Nup93 arrested) to dramatically (anti-Nup98- and 155- arrested) reduced number of pore-like structures as stained by x-Ycmplx-Nb1t (Fig S5)." Again, quantitation from several nuclei would strengthen these points.

Page15: "Since the Y-complex is the first NPC component to be recruited to assembly sites (Harel et al., 2003b; Otsuka et al., 2018), this indicates that all inhibitory nanobodies reduced the number of initiated assembly events." The two references concern postmitotic NPC assembly. As I understand the assay employed here reflects the interphase assembly mode.

Page 17 "Nup98 has a conserved domain structure: the unusually conserved N-terminal FG domain, interrupted by a Gle2-binding (GLEBS) domain, makes a key contribution to the NPC permeability barrier (Bailer et al., 1998; Hülsmann et al., 2012)." Also shown by Laurell et al. (doi: 10.1016/j.cell.2011.01.012.).

Page 21: "This implies that non-selective leakage is dominant and that even a very small fraction of barrier-free NPCs would already deteriorate the performance of the entire nuclear transport system." Can the authors provide an estimate about how large this "very small fraction" is?

Fig. 3A: Arrow NE surface points to the cytoplasm. I assume a cell is depicted.

Fig. 4: I am surprised that no exchange of Nup62 between old and new NPCs is observed, which I would have expected based on Rabut et al. Can the authors comment on this?

Fig. 7B: the nucleus assembled in the presence of xhNup93-Nb4i is strange locking. On this the passive import of MBP-mCherry is hard to evaluate and compare to the other inhibitory reactions.

Fig. 9: Info is lacking on how many nuclei were analyzed, how many independent experiments.

Page 42: "One day prior to use, HeLa cells were seeded in a 10-well glass slide at a density..." I guess it is also 543078, Greiner Bio One as stated on page 43.

Referee #2:

The manuscript entitled "Nucleoporin-binding nanobodies that either track or inhibit nuclear pore complex assembly" by Mireia Solà Colom et al., is a carefully planned and executed study, that contributes novel and significant advancement to a very difficult field in cell biology. My overall assessment is that this manuscript should be published and that no additional experiments should be required from the authors. The title might be somewhat misleading since this study does not simply present a new toolbox. The anti-nucleoporin tracking and inhibitory nanobodies are extremely powerful new tools that are rigorously analyzed in this study and are further used to obtain significant new findings regarding the two modes (postmitotic and interphase) of nuclear pore complex assembly. The authors also present an elegant and novel assay for tracking interphase NPC assembly. This hybrid frog-human experimental assay can distinguish between newly inserted and pre-existing NPCs in

the intact nuclear envelope of interphase nuclei. Finally, using a selected set of inhibitory nanobodies, the authors were able to obtain new insights on the assembly process, although specific structural ("architectural") intermediates were not defined. Realistically, and considering several decades of research in this vein, I think this study represents a remarkable and commendable achievement and it will clearly form the basis for future work. The authors rightly point out (page 19, just before the Discussion) that their findings suggest much greater mechanistic similarity between postmitotic and interphase NPC assembly than currently believed in the field. I think this is important and can be further elaborated upon in the Discussion. Another conclusion is a suggested assembly checkpoint linking pore formation to permeability barrier assembly and mentioned in several places in the text. As the authors state that this only "sketched here" (bottom of page 21), I believe this is a legitimate interpretation of the data. On another positive note, the authors state (page 19 and to be completed in table S.5) that all the nanobody plasmids will be made available to the community and this is to be commended.

The greatest weakness of the submitted manuscript is its meticulous and exhaustive nature. The huge number of details associated with all parts of the study makes it very challenging to read through the whole paper. At this age of short attention spans (even among scientists) it is at risk of eliciting the common response: "tl;dr". Therefore, my most important recommendation for the authors is to try to streamline the text, make the manuscript more "readable" and move more of the data to supplementary figures and tables. Some specific suggestions are detailed below.

[REFEREE 1 CROSS-COMMENTS: Regarding the length of the manuscript. As the work includes a huge amount of data I am not declined to read a longer manuscript. I personally like that the authors elaborate on their initial concepts and strategies as here as it helps to follow even a bit longer manuscripts, but this is certainly a matter of taste.]

In addition, I fully agree with the statement (top of page 11) that capturing assembly intermediates has remained the main bottleneck in the field. However, the current study only makes limited progress on this specific front and the sections on pages 15-16 that deal with "NPC-like structures", "rudimentary structures" and "derailed Nup compositions" - with specific emphasis on Figure 9 - are only poorly established in my eyes. Since I cannot think of an experimental strategy that would complement the experiments in Figure 9 and accurately reveal stoichiometry or Nup compositions, I think they need to tone down their statements. The idea that the observed structures represent "off-pathway intermediates that are heterogeneously assembled" is an interesting interpretation of the data, but is not proven by it.

Specific suggestions for improving the manuscript:

- Experts in the field will have no difficulty in following the text and switching between the different experimental systems in the figures. However, in the interest of other potential readers and since the story goes back and forth between systems, I think it is important to clearly spell out these transitions. The main experimental systems are in vitro assembly in *Xenopus* egg extracts, simple immunofluorescence labeling of HeLa cells and the hybrid system of digitonin-permeabilized HeLa cells incubated with *Xenopus* egg extracts. Both the description in the text (perhaps some sub-headings) and the figure-legend titles, or the first relevant section of the legend, should clarify this. This might seem trivial, but I believe it will make the manuscript more accessible to less experienced readers.
- As mentioned, the manuscript text is extremely long and detailed. At a minimum, I would suggest moving Figure 3C and the whole of Figure 6 to the Supplementary section and minimizing their description in the text. Since this also depends on the comments of additional reviewers, I will refrain from suggesting further shortening.
- Bottom of page 4: "(1B)" should be "(Fig 1B)".
- On page 8, the term "a high-speed extract of activated *Xenopus* eggs" will not be easily understood by the average reader. Rather than further lengthening the text, I would move extra explanations to the methods section or rely on previous cited work.
- Replace phrases like "objects that stained bright for" (top of page 16) with "stained brightly for" or "brightly stained by".
- In Figure 8 AB, double check nanobody labels in the 3rd and 4th rows since colors appear to be inverted between left and right (A and B) parts.

Referee #3:

Interphase assembly of nuclear pore complexes, where NPCs are assembled into an intact nuclear envelope, is notoriously difficult to study since this is a very rare event in vivo and there are no good systems to recapitulate this process in vitro. The current study addresses the lack of suitable experimental methods by evolving the classical *Xenopus* egg extract system, typically used to study post-mitotic NPC assembly, into a new tool to investigate interphase assembly. To do so, the authors incubate digitonin-permeabilized human tissue culture cells with *Xenopus* egg extract. Surprisingly, they observe that *Xenopus* NPCs can assemble into a human nuclear envelope, suggesting a remarkable evolutionary conservation of NPC assembly between the two species. They validate this method using previously developed *Xenopus*-specific anti-Nup nanobodies (tracking nanobodies), which shows a burst of newly assembling *Xenopus* NPCs that don't overlap with Nup107-GFP tagged NPCs of the HeLa nucleus.

Another key challenge for the investigation of NPC assembly is the difficulty to halt the process at specific steps. To address this, the authors try to find nanobodies that upon incorporation block the assembly process (inhibitory nanobodies). They use some elegant logic, immunizing alpacas simultaneously with both human and *Xenopus* epitopes followed by multiple rounds of phage display alternating between human and *Xenopus* orthologues. This setup favors the creation of nanobodies against conserved epitopes, which are more likely to lie in interaction surfaces between Nups. They find nanobodies for Nup93, Nup98 and Nup155 which upon incubation with the *Xenopus* egg extract prevent the assembly of functional NPCs as monitored by transport assays and the lack of Nup staining using various tracking nanobodies. To gain further mechanistic insights on the

NPC assembly block, they solve the structure for several of their target-nanobody complexes. From this data it becomes clear that the epitopes of the inhibiting antibodies lie in known interaction surfaces of Nups in the mature complex, while the epitopes of tracking nanobodies map to surfaces that are exposed in the mature NPC.

This study presents a stunning amount of very high-quality data using a variety of different techniques. The method the authors develop to study the interphase assembly of *Xenopus* NPCs in isolated human nuclei as well as the generated nanobodies will be very valuable tools for future research on NPC assembly. All experiments are designed logically, and the key findings are validated by orthogonal approaches. This is fantastic. However, with that in mind, this reviewer feels that the clarity of the manuscript should be improved and the data presented in a more accessible manner so that it's easier to understand.

Major points

1. The panel labeling of the figures should be carefully revised. At the moment, the text often references whole figures instead of specific panels. Since the figures are quite large and contain a lot of data it takes considerable effort to find the exact data that the text is referring to. This is especially true for the last part of the manuscript, where the effect of the inhibitory antibodies is presented (e.g. p15. [...] The two inhibitory nanobodies blocked the incorporation of Nup155 into Y-complex-positive structures in the NE (Fig 8, 9, S5) and prevented the assembly of transport-competent NPCs (Fig 7).[...]). Also, it could make sense to reverse the logic of the panels in figure 9, so that each inhibitory nanobody is an individual panel instead of the tracking nanobodies, this would allow easier referencing in the text.

2. Adding some more quantifications of the presented micrographs would strengthen the data considerably. Also, for the cases where numbers are provided, they are in a tabular format, which is harder to understand compared to when it would be shown in a graph. In some cases, the quantifications are also lacking altogether. This is especially noticeable when quantitative statements are made in the text, but the numbers are found nowhere in the figure. Some of these quantitative statements could be removed, but others are important and should be backed up by data (b, c):

- a. P 8: "During extract incubation, the initially flat HeLa nuclei became rounder; they grew in volume, suggesting that a massive import of *Xenopus* nuclear proteins had occurred, inflating nuclei through their osmotic-colloidal pressure (Fig 3C)." Volume increase cannot really be appreciated from the shown figure. Please show quantification plot of analyzed cells as well.
- b. P. 10, "[...] *Xenopus* NPCs represented between 25 and 40% of the pre-existing (i.e., human) NPCs [...]" and "Longer incubations did not increase the number of assembled *Xenopus* NPCs, suggesting that the system was already saturated after two hours, perhaps because membrane Nups (in particular, the essential membrane Nup Ndc1) had become limiting." => a time course progression and quantification should be presented
- c. p9: "[...] newly assembled (GFP-free) *Xenopus* pores as spots of similar size and brightness (Fig 4C) [...]" => quantification (e.g. histograms) should be shown
- d. p 10: "[...] allowed to resolve ring-like structures of ~100-120 nm in diameter [...]"
- e. p. 13: "[...] Nups of the NPC outer rings (e.g., Nup358, Nup133) labeled rings with larger diameters than nanobodies against inner ring components [...]"
- f. p16: "The Y-complex, Nup35, Nup93, Nup155, the Nup62 complex, and Nup98 still co-localized at approximately wild-type ratios." Please show quantification and refer to figure.

3. Since the figure panels are very complex, it is very difficult for the reader to process the information and conceptualize it. In particular, Figure 9 - although presenting detailed and quantitative data - is hard to understand. It would be very helpful to convert the data presented in this figure into a different format that could then potentially also provide the basis for a model. A possibility for such a representation would be to present a bar graph that annotates for each Y-complex positive spot whether this spot is positive also for the other Nup and possibly whether the intensity of the other Nup in the spot is comparable to normal NPCs. E.g. For Nup93-Nb4i, the graph could show that e.g. 80% of Y complex foci are positive for Nup155, but only 50% have Nup214 and 5% have Nup358. Or in xhNup155-Nb2i: 100% have Nup98, but only at 50% of the normal intensity. The current figure 9 could instead be moved to the supplement. It is unclear, what the axes in figure 9 show. What is the normalization?

[REFEREE 1 CROSS-COMMENTS: Regarding the presentation of Figure 9: this is a novel way presenting the data showing individual data points, which I found very helpful. Once thought about the concept I found the data easy to understand and importantly easy to compare between the panels. I cannot think of another easy way to present this.]

Minor points

1. Page 14: "xhNup98-Nb3i, xhNup155-Nb2i, or xhNup155-Nb3i were present during the nuclear assembly reaction. Nuclear import of IBB-MBPGFP was drastically reduced - 100-fold or even to the extent of nuclear exclusion (Figs. 7B, D)." Ref to figure S7 is missing.
2. Quantification for WGA and BAPTA is missing in figure 7C-D.
3. Page 15: "[...] Compared to the control, all assembly-arrested nuclei contained a moderately (anti-Nup93-arrested) to dramatically (anti-Nup98- and 155- arrested) reduced number of pore-like structures as stained by x-Ycmplx-Nb1t (Fig S5). Since the Y-complex is the first NPC component to be recruited to assembly sites (Harel et al., 2003b; Otsuka et al., 2018), this indicates that all inhibitory nanobodies reduced the number of initiated assembly events.[...]" It is hard to make such a strong statement here. If the recruitment of the Y-complex to chromatin indeed is the first assembly step, this should not be directly affected by the nanobodies. The reduced number of spots at the analyzed timepoint could be a secondary consequence of the

- lack of chromatin decondensation and nuclear growth that prevents the formation of additional NPCs. Also, it should be clarified if the authors refer to total NPC number or density (i.e. NPC number/area) here. The quantification is shown in a table in figure 8C and should be referred to. This table has one value not colored in the correct shade (xhNup155-Nb3i/xNup98-Nb1t).
4. Page 16: "[...] For example, we observed objects that stained bright for Nup358, the Y and Nup214 complexes but lacked Nup155 [...]". What is the frequency of such objects? A quantitative summary representation as suggested in major point 2 would be helpful. This applies also to other places where the text talks about off-pathway intermediates. The evidence presented to conclude that off-pathway intermediates are observed is insufficient.
 5. Page 16: "Nup35, Nup93, Nup155, the Nup62 complex, and Nup98 still co-localized at approximately wild-type ratios. In contrast, the Nup214·Nup88·Nup62 complex and Nup358 were detected only in some pore-like structures, with clearly reduced intensities (Fig 8, 9, S5)." This is not clear.
 6. p.5: "also visualized by live cell and electron microscopy (Goldberg et al., 1997; Dultz et al., 2008; Otsuka et al., 2016)." The citation should be Dultz et al., 2010
 7. p8:" Thus, when new NPCs were assembled, they also contained a functional permeability barrier that kept the imported material inside the nuclei." This is not shown. Either show import with NLS substrate or remove statement.
 8. p. 9: "The latter would be consistent with the reported mobility of these Nups (Griffis et al., 2002a, 2004)." Add citation Rabut et al. 2004
 9. p. 9: "Similarly, our newly solved cryo-EM structure shows that xNup93-Nb2t fits well into a pocket at the C-terminal of Xenopus Nup93,[...]" should be "C-terminus"
 10. A previously established assay for the analysis of interphase NPC assembly (D'Angelo et al., 2006) should be cited.
 11. Figure 3C gives the impression that there are a large number of fluorescent foci around the nucleus in the nanobody channels. How can the detection of those be prevented in the analysis?
 12. In figure 4A in the bottom row the rectangular highlighting box is not correctly positioned in the Nup107-GFP channel. Also, in Figure 5 A and 5B the circles are not positioned precisely in the same spot on the image, even though in this case the difference is small.
 13. P. 13/14: "All nanobodies that brightly stained NPCs of HeLa cells also permitted the formation of import-competent nuclei and, thus, of functional nuclear pores (Fig 7A),[...]" only one nanobody is shown in the figure. Show the others in a supplemental figure.
 14. Figure S5: Can the authors comment on the variability of the results? The staining shown for the Y complex antibody seems to vary quite strongly in some of the panels and that's also true for some of the other antibodies that were used multiple times. Quantification of the data would help to assess this globally and not just restricted to the small field of view shown.
 15. P. 41: "Notably, the strength of the inhibition might vary depending on the precise time point in which the inhibitory nanobodies are added. Egg extracts are arrested to interphase and activated by the addition of calcium. For a complete inhibitory effect, nanobodies need to be added at a very early time point and before the first interactions between Nups and Nup sub-complexes occur." This is not very precise. Please clearly state when the nanobodies were added. Before activation with calcium? Also, the sentence starting "Egg extracts" does not seem to make sense and should be checked for language and factual correctness.
 16. Figure 10 D: legend refers to interface as orange, but it is shown in red.

Max Planck Institute for Multidisciplinary Sciences

Dr. Dirk Görlich · Director Department of Cellular Logistics

Am Fassberg 11 · D-37077 Göttingen · E-mail: goerlich@mpinat.mpg.de · Tel.: ++49 551 2012401

To the Editors of the EMBO Journal
Dr. Hartmut Vodermaier

Göttingen, 12th February 2024

Manuscript EMBOJ-2023-115591

Dear Hartmut,

Thank you very much for the very positive evaluation of our manuscript and the valuable input for further improvements.

The reviewers did not identify any major problems. However, in the revision, we had to find a balance between the (not unjustified) comment of reviewer 2 who found our study already too complex for the average reader and the numerous requests to add further data and discuss more details. In fact, most of the 'missing' data was already in the original submission, so we focused on making this more accessible. This is explained in our point-by-point reply on the next pages. With that, we hope that our manuscript can now be accepted for publication.

With best regards,

Dirk Görlich

Changes to the manuscript

- As suggested by reviewer 2, we changed the title to emphasize the perhaps most impactful finding.
- We shortened the abstract to less than 175 words.
- We extensively optimized the text for clarity, simplicity, and readability.
- We included additional quantifications, namely for nuclear growth as a consequence of nuclear protein import during incubation with egg extract (Figure EV1B), for new NPC insertion during interphase (EV1C), comparing diameters of new and old NPCs (Figure 5C), and diameters of NPC rings decorated by tracking nanobodies (Figure 6B).
- We added Figure S2 to accommodate the biolayer interferometry (BLI) traces for the affinity measurements listed in Figure 2.
- We moved three main figure panels and three supplemental figures to the EV figures. There are now 12 main figures, 6 EV figures, 5 supplemental Figures, and 5 supplemental Tables.
- We compiled a large source data set, including raw images for Figures 1C-D, 3, 4, 5, 6A-B, D, 7A-C, 8A-B, 10A, 11C, 12A-B, EV1, EV5B, as well as numerical data for Figures 6C, 9 and S5.
- We also implemented all changes listed in the reply below.

Answers to the Reviewers' queries

(for clarity, we repeat the Referees' feedback in blue in front of each of our answers)

Referee #1:

Colom et al. describe in their manuscript "Nucleoporin-binding nanobodies that either track or inhibit nuclear pore complex assembly" nanobodies against nucleoporins as a versatile tool to investigate the mechanisms of NPC assembly. For this they selected and characterized two sets of antibodies. The first one is specific for the respective *Xenopus* nucleoporins and does not cross-react with the human proteins. They can be used in "classical" nuclear/NPC assembly assays using *Xenopus* egg extracts and a novel developed assays using *Xenopus* egg extracts and tissue culture derived human nuclei to follow *Xenopus* NPC integration and assembly into the nuclear envelope of the human nucleus. With these tracking antibodies the sequential assembly of NPCs can be followed. The second set constitutes nanobodies reacting with both *Xenopus* and human nucleoporins and some of these can be used to arrest NPC assembly at specific stages. Based on the here solved structures of the nanobody-target complexes specific hypotheses on their mode of inhibitory action are generated, which shed light on which nucleoporin-nucleoporin interactions are specifically needed at certain NPC assembly steps and are nicely reinforced by biochemical nucleoporin-nucleoporin interaction studies.

Overall, this study is very thoroughly conducted, includes a wealth of experiments and provides new insights into how NPC assemble.

Thank you!

specific major concerns essential to be addressed to support the conclusions:

Often, a single representative nucleus/experiment is shown to support the conclusions, e.g. Fig 3B. the authors should provide some quantitation, e.g. for Fig 3B the number of colocalizing Nup155 and Y-complex spots in comparison to single labeled spots form several nuclei.

Figure 3B is an overview of the interphase assembly assay showing a complete nucleus. Figure 4 shows well resolved colocalizations. In Reply Figure 1 (below), we again show colocalizations for newly inserted NPCs - as in Figure 4A - but without the ‘old NPC signal’. This also includes replicates from different nuclei and with a (politically incorrect) color scheme that makes it easier to immediately grasp colocalizing and non-colocalizing signals. The co-localizations between the frog Y-complex, Nup155, Nup94, Nup62, and Nup358 are essentially perfect. As these are “wild-type” assembly conditions, this is not really surprising. There are only very few spots with Y-complex signal but without Nup155 signal (and vice versa); and these are spots of only low signal intensity, probably representing intermediate structures prior to assembly of a complete Y-complex + inner ring scaffold.

It appears that the nucleus presented shows less Y-complex spots as compared to Nup155 spots. Does this say something about the assembly order?

This low-resolution overview image of Figure 3 does not allow for such conclusion. There, we tried to give an impression for a complete nucleus, which implies that not all NPCs are exactly in the same plane. Chromatic aberration might thus contribute to the variability in the intensity ratios. There could also be variability in decorating a given epitope with a fluorescent nanobody or a real variability in NPC composition. As illustrated Reply Figure 1, there are actually only very few spots that are Nup155 positive and Y-complex negative. Please note that the intensity ratios between different NPC markers also vary in the post-mitotic assembly mode, as can be seen in Figure 9, where we quantify hundreds of NPCs.

Also for the experiments presented in Fig 4 and 12, quantitation would support the statements.

Quantifications are shown below. They fully support our conclusions.

minor concerns that should be addressed:

Page 8: "the initially flat HeLa nuclei became rounder; they grew in volume, suggesting that a massive import of *Xenopus* nuclear proteins had occurred, inflating nuclei through their osmotic-colloidal pressure (Fig 3C)." Nuclear volume measurements would support this statement.

This is the quantification. The nuclear volume of the permeabilized cells at least doubles during incubation with egg extract. The data is now shown as Figure EV1B.

Page 9 "The latter would be consistent with the reported mobility of these Nups (Griffis et al., 2002a, 2004)." Rabut et al could be also cited here (DOI: 10.1038/ncb1184)

We added the Rabut et al. citation as suggested.

Page 9: "Finally, by staining newly-assembled pores with antihuman Ndc1 and -Gp210 antibodies, we confirmed that human membrane Nups present in the HeLa cells support the assembly of the new *Xenopus* NPCs.": What is meant by support the assembly? This might be rephrased given the controversial role of Gp201 and Ndc1 in (postmitotic) NPC assembly.

We are not aware that their roles are controversial. Ndc1 is clearly required for NPC assembly in vertebrates, gp210 is clearly not required. Fibroblasts, for example, lack gp210 and thus assemble NPCs in the absence of gp210.

Yet, this statement was not intended to discuss requirements, although we take it for granted that membrane-integral Nups are required for NPC assembly. Since these membrane Nups are not supplied with the (membrane-free) high-speed egg extract, they must be present in the NE template and be compatible with frog Nups. Figure EV3 documents the integration of human NDC1 and gp210 into the newly assembled frog NPCs. Sure, in the text, we could dive deeper into this topic. Given, however, that the text is already rather complex, we prefer the present, simpler statement.

Page 10: "To confirm the insertion of properly assembled NPCs, we acquired higher-resolution images of the HeLa NE containing newly-assembled pores, using two-colour STED (stimulation emission depletion) microscopy." The staining shows only y-complex structure. How do the authors know that those structures are fully assembled and functional NPCs?

We observed several features suggesting that the Y-complex positive structures include fully assembled NPCs. First of all, we observed ring-like assemblies of Y-complex protomers (Figure 5), which is a key criterion for correct NPC assembly. Second, we established that the Y-complex co-localizes in new NPCs with Nup155, Nup93, Nup358, Nup98, Nup153, Nup35, the Nup214 complex, the Nup62 complex, NDC1, and gp210 (Figure 4 and EV3). This is already a rather large dataset to support our conclusions. However, it is technically not yet possible to test symmetry, composition (with respect to all ~30 Nups), and transport proficiency at the level of individual NPCs seen in a microscopic image. If this was the question, then the reviewer may be asking for too much.

On the other hand, we probably see not only fully assembled NPCs but also smaller intermediates that have not yet reached the stage of a ring-like assembly. This is particularly obvious when comparing old NPCs (Fig. 5A) and new NPCs (Fig. 5B) in the higher resolved STED channel 1. Since this should not be fully ignored, we added the following note to the text: "*These STED images (in particular the higher resolved channel 1) appear to visualize not only fully assembled NPCs. Possibly they also captured smaller *Xenopus* intermediates that might not yet have reached the ring-like assembly stage (Fig. 5B). This is particularly obvious in comparison to the imaged old human NPCs that are surrounded by only very few intermediates (Fig. 5A, left).*"

Page 13: "As expected, nanobodies recognizing Nups of the NPC outer rings (e.g., Nup358, Nup133) labeled rings with larger diameters than nanobodies against inner ring components (e.g., Nup93, Nup35) (Fig 6B)." A robust quantitative analysis of the diameters would strengthen this point.

The diameters of the rings labeled by the different nanobodies have been measured. A plot is now included in Figure 6C and the raw data of the measurements is provided with the source data files.

Page 15: " Compared to the control, all assembly-arrested nuclei contained a moderately (anti-Nup93 arrested) to dramatically (anti-Nup98- and 155- arrested) reduced number of pore-like structures as stained by x-Ycplx-Nb1t (Fig S5)." Again, quantitation from several nuclei would strengthen these points.

The quantifications for that are shown in Figure 8C. Figure 8C is now referenced in the text.

Page15: "Since the Y-complex is the first NPC component to be recruited to assembly sites (Harel et al., 2003b; Otsuka et al., 2018), this indicates that all inhibitory nanobodies reduced the number of initiated assembly events." The two references concern postmitotic NPC assembly.

Yes, this is how the citations were meant, and indeed, the corresponding paragraph was about postmitotic assembly phenotypes. However, following the suggestions of reviewer 3, we removed the statement altogether, because the order of recruitment is not that clear.

As I understand the assay employed here reflects the interphase assembly mode.

No, in the current manuscript, we have focused on the effects of the inhibitory nanobodies on the postmitotic mode. The reviewer may be confusing this with our pre-submission, where we had two separate manuscripts, one on postmitotic and one on interphase NPC assembly, and the reviewers and editor insisted on merging the two. The inhibitory nanobody effects on interphase assembly are, however, essentially identical to the postmitotic assay. Most of this data fell victim to the merger and the need to reduce the complexity of the whole story. We have left only the inhibitory anti-Nup98 nanobody effect on interphase NPC assembly in the manuscript, because there, we can differentiate between the assembly phenotype and NPC-anchorage of Nup98 in a single experiment.

Page 17 "Nup98 has a conserved domain structure: the unusually conserved N-terminal FG domain, interrupted by a Gle2-binding (GLEBS) domain, makes a key contribution to the NPC permeability barrier (Bailer et al., 1998; Hülsmann et al., 2012)." Also shown by Laurell et al. (doi: 10.1016/j.cell.2011.01.012.).

We added the Laurell reference as suggested.

Page 21: "This implies that non-selective leakage is dominant and that even a very small fraction of barrier-free NPCs would already deteriorate the performance of the entire nuclear transport system." Can the authors provide an estimate about how large this "very small fraction" is?

Well, if only 1% of NPCs had no permeability barrier and if this fraction leaked 1000 times more of a GFP-sized macromolecule, then the total leakage would increase by a factor of 10. This should already have a detrimental effect on the RanGTP gradient and thus on active transport.

Fig. 3A: Arrow NE surface points to the cytoplasm. I assume a cell is depicted.

Corrected, so the arrow points now to the nucleus.

Fig. 4: I am surprised that no exchange of Nup62 between old and new NPCs is observed, which I would have expected based on Rabut et al. Can the authors comment on this?

Rabut et al., (2004) reported a residence time of 13 hours for Nup62, which is not exactly high mobility. In our case, we incubated for up to 2 h at 20°C, and this lower temperature is likely to reduce the exchange rate further. So, we would not conclude that there is a big difference. Keep also in mind that Nup62 is present in two different complexes (Nup62·Nup54·Nup58 and Nup62·Nup88·Nup214) and that these may have different residence times.

Nevertheless, the very stable NPC anchorage of the Nup62·Nup54·Nup58 complex is puzzling, given that it binds the Nup93 N-terminus with only μM affinity. This suggests that additional interactions keep it in place.

Fig. 7B: the nucleus assembled in the presence of xhNup93-Nb4i is strange locking. On this the passive import of MBP-mCherry is hard to evaluate and compare to the other inhibitory reactions.

Weird morphologies with growth defects are part of the phenotype observed in nuclei arrested by the xhNup93-Nb4i nanobody. The nuclei are unable to exclude MBP-mCherry, which is consistent with the defect in outer ring assembly that is described in the manuscript. To better illustrate the described phenotype, three additional image examples are included in the source data.

Fig. 9: Info is lacking on how many nuclei were analyzed, how many independent experiments.

The information has been added to the figure legend of figure 9. In addition, these numerical data are now provided with the source data files.

Page 42: "One day prior to use, HeLa cells were seeded in a 10-well glass slide at a density..." I guess it is also 543078, Greiner Bio One as stated on page 43.

Yes, it is. This is now added to the text.

Referee #2:

The manuscript entitled "Nucleoporin-binding nanobodies that either track or inhibit nuclear pore complex assembly" by Mireia Solà Colom et al., is a carefully planned and executed study, that contributes novel and significant advancement to a very difficult field in cell biology. My overall assessment is that this manuscript should be published and that no additional experiments should be required from the authors. The title might be somewhat misleading since this study does not simply present a new toolbox.

This is a valid point, made also by other colleagues. We therefore changed the title to now highlight the conceptually most relevant conclusion: *'A checkpoint function for Nup98 in nuclear pore formation suggested by inhibitory nanobodies'*

The anti-nucleoporin tracking and inhibitory nanobodies are extremely powerful new tools that are rigorously analyzed in this study and are further used to obtain significant new findings regarding the two modes (postmitotic and interphase) of nuclear pore complex assembly. The authors also present an elegant and novel assay for tracking interphase NPC assembly. This hybrid frog-human experimental assay can distinguish between newly inserted and pre-existing NPCs in the intact nuclear envelope of interphase nuclei. Finally, using a selected set of inhibitory nanobodies, the authors were able to obtain new insights on the assembly process, although specific structural ("architectural") intermediates were not defined. Realistically, and considering several decades of research in this vein, I think this study represents a remarkable and commendable achievement and it will clearly form the basis for future work. The authors rightly point out (page 19, just before the Discussion) that their findings suggest much greater mechanistic similarity between postmitotic and interphase NPC assembly than currently believed in the field. I think this is important and can be further elaborated upon in the Discussion. Another conclusion is a suggested assembly checkpoint linking pore formation to permeability barrier assembly and mentioned in several places in the text. As the authors state that this only "sketched here" (bottom of page 21), I believe this is a legitimate interpretation of the data. On another positive note, the authors state (page 19 and to be completed in table S.5) that all the nanobody plasmids will be made available to the community and this is to be commended.

Thankyou!

The greatest weakness of the submitted manuscript is its meticulous and exhaustive nature. The huge number of details associated with all parts of the study makes it very challenging to read through the whole paper. At this age of short attention spans (even among scientists) it is at risk of eliciting the common response: "tl;dr".

I (DG) had to google "tl;dr", but what can we say? We are not surprised by this comment. Originally, we had two back-to-back manuscripts, which made the story much easier to digest. The Editor and the two reviewers, who looked at our pre-submission, insisted on merging the two. This is now the result when following the Editor's advice.

Therefore, my most important recommendation for the authors is to try to streamline the text, make the manuscript more "readable" and move more of the data to supplementary figures and tables. Some specific suggestions are detailed below.

We introduced many small changes to the text to improve readability and to make the concepts more visible amidst the details. However, the reviewers had also requested to add more data and quantifications, which moved the manuscript again towards more complexity. We hope that we have found a good compromise.

[REFEREE 1 CROSS-COMMENTS: Regarding the length of the manuscript. As the work includes a huge amount of data I am not declined to read a longer manuscript. I personally like that the authors elaborate on their initial concepts and strategies as here as it helps to follow even a bit longer manuscripts, but this is certainly a matter of taste.]

In addition, I fully agree with the statement (top of page 11) that capturing assembly intermediates has remained the main bottleneck in the field. However, the current study only makes limited progress on this specific front and the sections on pages 15-16 that deal with "NPC-like structures", "rudimentary structures" and "derailed Nup compositions" - with specific emphasis on Figure 9 - are only poorly established in my eyes. Since I cannot think of an experimental strategy that would complement the experiments in Figure 9 and accurately reveal stoichiometry or Nup compositions, I think they need to tone down their statements. The idea that the observed structures represent "off-pathway intermediates that are heterogeneously assembled" is an interesting interpretation of the data, but is not proven by it.

The two issues may be related. Perhaps it is so difficult to trap defined assembly intermediates (at least with methods with inhibitory nanobodies) because the assembly pathway can diverge into heterogeneous states – depending on which components are available. NPC-like structures of aberrant composition are an observation and not just an interpretation (See Figure 9). Whether these structures can be chased into proper NPCs when the nanobody block is relieved (if this were possible) is indeed an open question. However, with the inhibitory nanobodies bound, these are clearly dead-end (= off-pathway) products.

Specific suggestions for improving the manuscript:

- Experts in the field will have no difficulty in following the text and switching between the different experimental systems in the figures. However, in the interest of other potential readers and since the story goes back and forth between systems, I think it is important to clearly spell out these transitions. The main experimental systems are in vitro assembly in *Xenopus* egg extracts, simple immunofluorescence labeling of HeLa cells and the hybrid system of digitonin-permabilized HeLa cells incubated with *Xenopus* egg extracts. Both the description in the text (perhaps some sub-headings) and the figure-legend titles, or the first relevant section of the legend, should clarify this. This might seem trivial, but I believe it will make the manuscript more accessible to less experienced readers.

All switches between the experimental systems are now clearly indicated, and we also periodically remind the reader of which system is currently described/ discussed.

- As mentioned, the manuscript text is extremely long and detailed. At a minimum, I would suggest moving Figure 3C and the whole of Figure 6 to the Supplementary section and minimizing their description in the text. Since this also depends on the comments of additional reviewers, I will refrain from suggesting further shortening.

Figures 3C, 4D, and 7E have been moved to the Expanded View (EV) as suggested by the Editor.

Figure 6A and 6C, however, are really crucial. They illustrate a central concept, namely that inhibitory epitopes are buried in the assembled state but accessible in the pre-assembly state.

- Bottom of page 4: "(1B)" should be "(Fig 1B)".

Corrected.

- On page 8, the term "a high-speed extract of activated *Xenopus* eggs" will not be easily understood by the average reader. Rather than further lengthening the text, I would move extra explanations to the methods section or rely on previous cited work.

High-speed extract is synonymous with a membrane-depleted extract. This is actually a detail the reader needs to know for fully understanding the interphase NPC assembly assay. This is now re-phrased accordingly and in a more concise manner.

- Replace phrases like "objects that stained bright for" (top of page 16) with "stained brightly for" or "brightly stained by".

“Stain bright” is correct English. One also writes “the sun shines bright”. The alternative phrasing “the sun shines brightly” comes up mostly in German translations. The Google count is 98 million to 13 million in favor of "bright".

- In Figure 8 AB, double check nanobody labels in the 3rd and 4th rows since colors appear to be inverted between left and right (A and B) parts.

Corrected.

Referee #3:

Interphase assembly of nuclear pore complexes, where NPCs are assembled into an intact nuclear envelope, is notoriously difficult to study since this is a very rare event in vivo and there are no good systems to recapitulate this process in vitro. The current study addresses the lack of suitable experimental methods by evolving the classical *Xenopus* egg extract system, typically used to study post-mitotic NPC assembly, into a new tool to investigate interphase assembly. To do so, the authors incubate digitonin-permeabilized human tissue culture cells with *Xenopus* egg extract. Surprisingly, they observe that *Xenopus* NPCs can assemble into a human nuclear envelope, suggesting a remarkable evolutionary conservation of NPC assembly between the two species. They validate this method using previously developed *Xenopus*-specific anti-Nup nanobodies (tracking nanobodies), which shows a burst of newly assembling *Xenopus* NPCs that don't overlap with Nup107-GFP tagged NPCs of the HeLa nucleus.

Another key challenge for the investigation of NPC assembly is the difficulty to halt the process at specific steps. To address this, the authors try to find nanobodies that upon incorporation block the assembly process (inhibitory nanobodies). They use some elegant logic, immunizing alpacas simultaneously with both human and *Xenopus* epitopes followed by multiple rounds of phage display alternating between human and *Xenopus* orthologues. This setup favors the creation of nanobodies against conserved epitopes, which are more likely to lie in interaction surfaces between Nups. They find nanobodies for Nup93, Nup98 and Nup155 which upon incubation with the *Xenopus* egg extract prevent the assembly of functional NPCs as monitored by transport assays and the lack of Nup staining using various tracking nanobodies. To gain further mechanistic insights on the NPC assembly block, they solve the structure for several of their target-nanobody complexes. From this data it becomes clear that the epitopes of the inhibiting antibodies lie in known interaction surfaces of Nups in the mature complex, while the epitopes of tracking nanobodies map to surfaces that are exposed in the mature NPC.

This study presents a stunning amount of very high-quality data using a variety of different techniques. The method the authors develop to study the interphase assembly of *Xenopus* NPCs in isolated human nuclei as well as the generated nanobodies will be very valuable tools for future research on NPC assembly. All experiments are designed logically, and the key findings are validated by orthogonal approaches. This is fantastic.

Thank you!

However, with that in mind, this reviewer feels that the clarity of the manuscript should be improved and the data presented in a more accessible manner so that it's easier to understand.

Major points

1. The panel labeling of the figures should be carefully revised. At the moment, the text often references whole figures instead of specific panels. Since the figures are quite large and contain a lot of data it takes considerable effort to find the exact data that the text is referring to. This is especially true for the last part of the manuscript, where the effect of the inhibitory antibodies is presented (e.g. p15. [...] The two inhibitory nanobodies blocked the incorporation of Nup155 into Y-complex-positive structures in the NE (Fig 8, 9, S5) and prevented the assembly of transport-competent NPCs (Fig 7).[...]).

Agreed. We now refer to individual panels wherever this is appropriate.

Also, it could make sense to reverse the logic of the panels in figure 9, so that each inhibitory nanobody is an individual panel instead of the tracking nanobodies, this would allow easier referencing in the text.

Figure 9 can be read as a table with rows and columns. Each column represents one inhibitory nanobody, each row represents one tracking nanobody, and each tracking-inhibitory nanobody combination is illustrated by one scatter plot. Perhaps the panel letters (A-E) in the original submission obscured this logic. We have now removed them for more clarity.

2. Adding some more quantifications of the presented micrographs would strengthen the data considerably. Also, for the cases where numbers are provided, they are in a tabular format, which is harder to understand compared to when it would be shown in a graph. In some cases, the quantifications are also lacking altogether. This is especially noticeable when quantitative statements are made in the text, but the numbers are found nowhere in the figure. Some of these quantitative statements could be removed, but others are important and should be backed up by data (b, c):

Figure 9 shows a lot of the quantifications in an intuitively accessible, graphical format referring to the data shown in Figures 8 and S5 (now EV6). Please note that the quantifications in Figure 9 are based on much larger datasets than the individual microscopic images shown in the other figures.

Nevertheless, and as already detailed in our reply to reviewer 1, additional quantifications have now been added for figures 3C (now EV1), 5, and 6 (see new panels 5C, 6C, EV1B, and EV1C).

Regarding the suggestion to change the tabular to the graph format for figures 7D and 8C, we believe that the tabular format is more appropriate. The differences on the described phenotypes can be easily appreciated, and we color-coded the table cells to make it more obvious. Graphs would also take more space.

a. P 8: "During extract incubation, the initially flat HeLa nuclei became rounder; they grew in volume, suggesting that a massive import of Xenopus nuclear proteins had occurred, inflating nuclei through their osmotic-colloidal pressure (Fig 3C)." Volume increase cannot really be appreciated from the shown figure. Please show quantification plot of analyzed cells as well.

Quantifications are added in Fig. 3C, which is now Figure EV1. The nuclear volume roughly double upon incubation with egg extract. See also our answer to reviewer 1.

b. P. 10, "[...] Xenopus NPCs represented between 25 and 40% of the pre-existing (i.e., human) NPCs [...]" and "Longer incubations did not increase the number of assembled Xenopus NPCs, suggesting that the system was already saturated after two hours, perhaps because membrane Nups (in particular, the essential membrane Nup Ndc1) had become limiting." => a time course progression and quantification should be presented

Sure, we could do that. However, given that the manuscript is already overloaded with data (Reviewer 2 has really a point here) we are hesitant to add even more. We had performed a series of pretrials to establish robust conditions for the assay, and this is the bottom line. When looking into details, there is variability between different egg extracts. For very active ones, the endpoint is reached already after one hour. For others, this takes two hours. When the endpoint is reached also depends on the extract volume in the assay. We think that this is sufficient information for reproducing the assay.

As already detailed in our answer to reviewer 1, a quantification of new NPC insertion is now shown as Figure EV1C.

c. p9: "[...] newly assembled (GFP-free) Xenopus pores as spots of similar size and brightness (Fig 4C) [...]" => quantification (e.g. histograms) should be shown

This is obvious from the images, in particular, the human-frog cross-reacting anti-Nup35 and anti-Nup214 complex nanobodies in figure 4C allow a direct comparison. Even if we would include such histograms, this would require many additional panels. Thank you for the suggestion, but we prefer not to increase the complexity of the data even further.

d. p 10: [...] allowed to resolve ring-like structures of ~100-120 nm in diameter [...]

We added plots with diameter measurements as the panel 5C. The corresponding raw data are included in the source data files.

e. p. 13: [...] Nups of the NPC outer rings (e.g., Nup358, Nup133) labeled rings with larger diameters than nanobodies against inner ring components [...]

The diameters of the rings labeled by the different nanobodies have been measured and a plot is now included as Fig. 6C.

f. p16: "The Y-complex, Nup35, Nup93, Nup155, the Nup62 complex, and Nup98 still co-localized at approximately wild-type ratios." Please show quantification and refer to figure.

Quantifications are shown in Figure 9. We have added another reference to the figure. The raw data of the measurements is included with the source data.

3. Since the figure panels are very complex, it is very difficult for the reader to process the information and conceptualize it. In particular, Figure 9 - although presenting detailed and quantitative data - is hard to understand. It would be very helpful to convert the data presented in this figure into a different format that could then potentially also provide the basis for a model. A possibility for such a representation would be to present a bar graph that annotates for each Y-complex positive spot whether this spot is positive also for the other Nup and possibly whether the intensity of the other Nup in the spot is comparable to normal NPCs. E.g. For Nup93-Nb4i, the graph could show that e.g. 80% of Y complex foci are positive for Nup155, but only 50% have Nup214 and 5 % have Nup358. Or in xhNup155-Nb2i: 100% have Nup98, but only at 50% of the normal intensity. The current figure 9 could instead be moved to the supplement. It is unclear, what the axes in figure 9 show. What is the normalization?

[REFEREE 1 CROSS-COMMENTS: Regarding the presentation of Figure 9: this is a novel way presenting the data showing individual data points, which I found very helpful. Once thought about the concept I found the data easy to understand and importantly easy to compare between the panels. I cannot think of another easy way to present this.]

Indeed, we had considered many ways of presenting the large dataset of combinatorial quantifications, and Figure 9 was the most intuitive. In fact, it is organized like a matrix/table, one reads columns with inhibitory and rows with tracking nanobodies. Reviewer 2 suggested to rotate the matrix, but we think that the problem was confusion by the panel letters (A-E), which have now been removed for clarity. We further optimized the figure design to make the matrix logic clearer.

Minor points

1. Page 14: "xhNup98-Nb3i, xhNup155-Nb2i, or xhNup155-Nb3i were present during the nuclear assembly reaction. Nuclear import of IBB-MBPGFP was drastically reduced - 100-fold or even to the extent of nuclear exclusion (Figs. 7B, D)." Ref to figure S7 is missing.

Added as suggested.

2. Quantification for WGA and BAPTA is missing in figure 7C-D.

The inhibitory effects of WGA and BAPTA are standard controls in the NPC assembly field, and we feel that this does not need further validation, also because the phenotypes are fully penetrant. Adding WGA and BAPTA quantifications to Figure 9, for example, would make the figure even more complex and no longer fit onto one page. And we have already received the "tl;dr" comment from reviewer 2.

3. Page 15: "[...] Compared to the control, all assembly-arrested nuclei contained a moderately (anti-Nup93-arrested) to dramatically (anti-Nup98- and 155- arrested) reduced number of pore-like structures as stained by x-Ycmplx-Nb1t (Fig S5). Since the Y-complex is the first NPC component to be recruited to assembly sites (Harel et al., 2003b; Otsuka et al., 2018), this indicates that all inhibitory nanobodies reduced the number of initiated assembly events.[...]" It is hard to make such a strong statement here. If the recruitment of the Y-complex to chromatin indeed is the first assembly step, this should not be directly affected by the nanobodies.

We take the point. The nanobodies block the interaction between NPC subcomplexes, and this is probably not the first event in NPC assembly. We have removed the statement.

The reduced number of spots at the analyzed timepoint could be a secondary consequence of the lack of chromatin decondensation and nuclear growth that prevents the formation of additional NPCs.

Indeed, this could explain why the inhibitory anti-Nup93 nanobody reduced the number of NPCs (to one third), even though it only blocks the outer ring assembly. We have changed the text accordingly. However, the block by the inhibitory anti-Nup155 and anti-Nup98 nanobodies appears to be much tighter, directly preventing the assembly of the NPC scaffold.

Also, it should be clarified if the authors refer to total NPC number or density (i.e. NPC number/area) here. The quantification is shown in a table in figure 8C and should be referred to. This table has one value not colored in the correct shade (xhNup155-Nb3i/xNup98-Nb1t).

It refers to the total NPC number and this should be clear from the figure heading "*Number of Nup-positive spots*" and the figure legend "*All inhibitory nanobodies result in the formation of nuclei with a reduced number of NPCs or arrested structures. The number of NPCs and arrested structures was counted from acquired images of the nuclear surface*".

4. Page 16: "[...] For example, we observed objects that stained bright for Nup358, the Y and Nup214 complexes but lacked Nup155 [...]". What is the frequency of such objects? A quantitative summary representation as suggested in major point 2 would be helpful. This applies also to other places where the text talks about off-pathway intermediates. The evidence presented to conclude that off-pathway intermediates are observed is insufficient.

Their frequencies can be seen in Figure 9 (in respect to the Y-complex signal), comparing the first with the second row. As far as we can tell, these residual structures are dead-end products (= off-pathway intermediates) because the nuclei that contain them remain fully inactive in nuclear import.

5. Page 16: "Nup35, Nup93, Nup155, the Nup62 complex, and Nup98 still co-localized at approximately wild-type ratios. In contrast, the Nup214·Nup88·Nup62 complex and Nup358 were detected only in some pore-like structures, with clearly reduced intensities (Fig 8, 9, S5)." This is not clear.

Figure 9 shows the quantifications, and there the effects are very clear. One needs to compare the coloured with the grey dots. Figure S5 (now EV6) shows representative images, and also here, the general reduction in co-localization as well as a small number of still co-localizing dots can be seen.

6. p.5: "also visualized by live cell and electron microscopy (Goldberg et al., 1997; Dultz et al., 2008; Otsuka et al., 2016)." The citation should be Dultz et al., 2010

Corrected as suggested.

7. p.8: " Thus, when new NPCs were assembled, they also contained a functional permeability barrier that kept the imported material inside the nuclei." This is not shown. Either show import with NLS substrate or remove statement.

It is very well established that nuclei of digitonin-permeabilised cells are very active in importing NLS-containing proteins when incubated with an interphase egg extract (see e.g. Figure 2 in Görlich et al., 1994, Cell). This can be shown by following an NLS-tracer. The stronger argument is, however, the increase in volume because this indeed the consequence of bulk protein import. The volume doubles during the incubation with extract. Our statement referred to this later criterion. We re-phrased this sentence to make this even clearer and added the quantification of nuclear volume increase as Figure EV1B. Please see also our answer to reviewer 1.

8. p. 9: "The latter would be consistent with the reported mobility of these Nups (Griffis et al., 2002a, 2004)." Add citation Rabut et al. 2004

Added as suggested.

9. p. 9: "Similarly, our newly solved cryo-EM structure shows that xNup93-Nb2t fits well into a pocket at the C-terminal of Xenopus Nup93,[...]" should be "C-terminus"

Changed as suggested.

10. A previously established assay for the analysis of interphase NPC assembly (D'Angelo et al., 2006) should be cited.

Citation is added in the introduction.

11. Figure 3C gives the impression that there are a large number of fluorescent foci around the nucleus in the nanobody channels. How can the detection of those be prevented in the analysis?

These are probably annulate lamellae and thus nothing unusual. During the analysis, only the NPCs that are on the same plane as the GFP signal (*i.e.*, embedded in the nuclear membrane) have been considered.

12. In figure 4A in the bottom row the rectangular highlighting box is not correctly positioned in the Nup107-GFP channel. Also, in Figure 5 A and 5B the circles are not positioned precisely in the same spot on the image, even though in this case the difference is small.

Amended as suggested.

13. P. 13/14: "All nanobodies that brightly stained NPCs of HeLa cells also permitted the formation of import-competent nuclei and, thus, of functional nuclear pores (Fig 7A),[...]" only one nanobody is shown in the figure. Show the others in a supplemental figure.

This is shown indirectly by all tracking (labelled) nanobodies that were present during these assembly reactions, as well as by the "Non-inhibitory Nb" controls used in Figures 7A, 8A-B, 9, 12B, EV5 A-C and EV6. In each case, the non-inhibitory control was an unlabelled tracking nanobody used at the same concentration as the inhibitory nanobodies. These are now indicated in the corresponding Figure legends and include:

- Figure 7A, Figure 12B, Figure EV5: xhNup93-Nb3t
- Figure 8A-B, Figure 9, and Figure EV6: xhNup93-Nb3t, xNup93-Nb1t, xhNup35-Nb1t, xNup155-Nb1t, xNup358-Nb1t and xNup62-Nb1t.

For technical reasons, these controls had to be different from the labelled tracking nanobodies (otherwise, the labeled ones would be outcompeted and could not stain).

14. Figure S5: Can the authors comment on the variability of the results? The staining shown for the Y complex antibody seems to vary quite strongly in some of the panels and that's also true for some of the other antibodies that were used multiple times. Quantification of the data would help to assess this globally and not just restricted to the small field of view shown.

We agree, and therefore we show exhaustive quantifications in Figure 9. Please see also our discussion on staining variability in our answer to reviewer 1.

15. P. 41: "Notably, the strength of the inhibition might vary depending on the precise time point in which the inhibitory nanobodies are added. Egg extracts are arrested to interphase and activated by the addition of calcium. For a complete inhibitory effect, nanobodies need to be added at a very early time point and before the first interactions between Nups and Nup sub-complexes occur." This is not very precise. Please clearly state when the nanobodies were added. Before activation with calcium? Also, the sentence starting "Egg extracts" does not seem to make sense and should be checked for language and factual correctness.

We used interphase extract from Ca^{2+} activated eggs. For clarity, we changed the order of the sentence: Extracts were prepared from Ca^{2+} activated eggs, stabilized with cycloheximide and thus arrested in interphase.

Such interphase extract initially contains still mitotic phosphorylations, which are only slowly removed. Nanobodies were added immediately after thawing the extract on ice. No inhibitory effect was evident when added 30 minutes after mixing with chromatin, membranes and E-mix (see Figure 7E – now EV4).

16. Figure 10 D: legend refers to interface as orange, but it is shown in red.

Corrected.

Dr. Dirk Görlich
MPI for Multidisciplinary Sciences, Göttingen
Cellular Logistics
Am Fassberg 11
Göttingen, Germany 37077
Germany

5th Mar 2024

Re: EMBOJ-2023-115591R
A checkpoint function for Nup98 in nuclear pore formation suggested by inhibitory nanobodies

Dear Dirk,

Thank you again for submitting your revised manuscript to The EMBO Journal. After careful re-assessment by our editorial office and by referee 3, I am pleased to inform you that we should now be able to swiftly proceed with acceptance and publication of your study, as soon as the following remaining editorial points have been addressed:

- Referee 3 noted a small presentational issue that may still need to be modified.
- Please adjust the order of the manuscript sections: Title page with complete author information, Abstract, Keywords, Introduction, Results, Discussion, Materials & Methods, Data Availability Section, Acknowledgements, Disclosure and Competing Interests Statement, References, Main figure legends, Tables, Expanded Figure Legends. Please remember to include 4-5 general keyword terms to enhance searchability on the abstract page, and to add the required "Disclosure and Competing Interests Statement"
- In the Data Availability section, while direct URLs for the datasets are not essential, URLs pointing to the cited databases themselves should be included. Suggested wording: "The [structural coordinates | microarray | mass spectrometry ...] data from this publication have been deposited to the [name of the database] database [URL] and assigned the identifier(s) [accession | permalink | hashtag ...]". Also, please ensure that datasets become publicly accessible upon acceptance.
- Please remove the DOI information from the reference list (except for pre-published papers that do not have assigned volume and page/locator numbers)
- In the Appendix, please name each figure and table "Appendix Figure/Table S1...", according to their callouts in the text and in the Appendix ToC
- Figure 12A has not been called out in the manuscript, please check (most likely, the first reference to Fig 12B should be Fig 12A-B?).
- We note that some of the images in Figure 4A appear to be also (partly) shown in Figure 3B. Please check/clarify and state this explicitly in the legends to both figures.
- Routine pre-acceptance checks by our data editors have also raised a few queries regarding figures, data, and legends, which I would ask you to address (ideally using track changes and/or commenting were the manuscript text is concerned, to facilitate my re-checking):
 - * A separate 'Data Information' section is required in the legends of figures 4a-c; 5a-b; 7a-c; 8a-b; 12a-b.
 - * Figure 4A-C legends define "rectangles", which are however not present in these figures. This needs to be rectified.
 - * Please note that the white rectangles are not defined in the legend of figure 6b. This needs to be rectified.
 - * Please note that information related to N is missing in the legend of figure 6c.
 - * Please note that the error bars are not defined in the legends of figures 5c; 6c; EV 1b.
 - * Please note that the scale bar needs to be defined for figure EV 5b.
- Please provide suggestions for a short 'blurb' text prefacing and summing up the conceptual aspect of the study in two sentences (max. 250 characters), followed by 3-5 one-sentence 'bullet points' with brief factual statements of key results of the paper; they will form the basis of an editor-written 'Synopsis' accompanying the online version of the article. Please also upload a synopsis image, which can be used as a "visual title" for the synopsis section of your paper. The image should be in PNG or JPG format, and please make sure that it remains in the modest dimensions of (exactly) 550 pixels wide and 300-600 pixels high.
- Finally, I would propose to incorporate the following minor changes to the title and abstract, mainly to make them slightly more

explicit :

Title: "A checkpoint function for Nup98 in nuclear pore formation suggested by NOVEL inhibitory nanobodies" - or even better turned into active voice "New inhibitory nanobodies reveal/indicate a checkpoint function for Nup98 in nuclear pore formation"?
Abstract, 1st sentence: "Nuclear pore complex (NPC) biogenesis is A STILL ENIGMATIC example of protein self-assembly."

I am therefore returning the manuscript to you for a final round of minor revision, solely to allow you to make these modifications and upload any revised files. After that, we shall be ready to proceed with formal acceptance and production of the manuscript.

Yours sincerely,

Hartmut

9) Digital image enhancement is acceptable practice, as long as it accurately represents the original data and conforms to community standards. If a figure has been subjected to significant electronic manipulation, this must be clearly noted in the figure legend and/or the 'Materials and Methods' section. The editors reserve the right to request original versions of figures and the original images that were used to assemble the figure. Finally, we generally encourage uploading of numerical as well as gel/blot image source data; for details see: embopress.org/page/journal/14602075/authorguide#sourcedata

At EMBO Press, we ask authors to provide source data for the main manuscript figures. Our source data coordinator will contact you to discuss which figure panels we would need source data for and will also provide you with helpful tips on how to upload and organize the files.

Further information is available in our Guide For Authors:

In the interest of ensuring the conceptual advance provided by the work, we recommend submitting a revision within 3 months (3rd Jun 2024). Please discuss the revision progress ahead of this time with the editor if you require more time to complete the revisions. Use the link below to submit your revision:

Link Not Available

Referee #3:

Overall the modification made by the authors have improved the manuscript considerably and I recommend publication in the EMBO Journal. As said in my original review, the manuscript describes tools that will be very valuable for the community and characterizes them extensively with novel insight into the interfaces that are crucial for NPC assembly.

I had suggested some modifications to data presentation and quantification. Although I do not agree with the authors in all points on why they do not implement some of these modifications, I consider it their choice and it should not preclude publication. One note in my previous review that the authors may have overlooked: The table in figure 8C has one value not colored in the correct shade (xhNup155-Nb3i/xNup98-Nb1t).

All editorial and formatting issues were resolved by the authors.

Dr. Dirk Görlich
MPI for Multidisciplinary Sciences, Göttingen
Cellular Logistics
Am Fassberg 11
Göttingen, Germany 37077
Germany

13th Mar 2024

Re: EMBOJ-2023-115591R1

A checkpoint function for Nup98 in nuclear pore formation suggested by novel inhibitory nanobodies

Dear Dirk,

Thank you for submitting your final revised manuscript for our consideration. I am pleased to inform you that we have now accepted it for publication in The EMBO Journal.

Yours sincerely,

Hartmut
